# Beam electrons as a source of Hα flare ribbons

Malcolm Druett[1], Eamon Scullion[1], Valentina Zharkova[1], Sarah Matthews[2], Sergei Zharkov[3] & Luc Rouppe Van der Voort[4]

The observations of solar flare onsets show rapid increase of hard and soft X-rays, ultra-violet emission with large Doppler blue shifts associated with plasma upflows, and Hα hydrogen emission with red shifts up to 1–4 Å. Modern radiative hydrodynamic models account well for blue-shifted emission, but struggle to reproduce closely the red-shifted Hα lines. Here we present a joint hydrodynamic and radiative model showing that during the first seconds of beam injection the effects caused by beam electrons can reproduce Hα line profiles with large red-shifts closely matching those observed in a C1.5 flare by the Swedish Solar Telescope. The model also accounts closely for timing and magnitude of upward motion to the corona observed 29 s after the event onset in 171 Å by the Atmospheric Imaging Assembly/Solar Dynamics Observatory.

[1] Northumbria University, Department of Mathematics, Physics and Electrical Engineering, Newcastle upon Tyne NE1 8ST, UK. [2] Mullard Space Science Laboratory, Department of Space & Climate Physics, University College London, Holmbury St Mary, Dorking, Surrey RH5 6NT, UK. [3] Hull University, School of Mathematics & Physical Sciences, Kingston upon Hull, East Yorkshire HU8 5ST, UK. [4] Institute of Theoretical Astrophysics, University of Oslo, P.O. Box 1029, Blindern, Oslo NO-0315, Norway. Correspondence and requests for materials should be addressed to M.D. (email: malcolm.druett@northumbria.ac.uk).

Complex processes of plasma heating in solar flares can be effectively diagnosed from the increase of intensities of hard X-ray emission combined with soft X-ray, extreme ultra-violet and ultra-violet emission with blue-shifts reported from the early space observations[1,2]. Flare dynamics in the lower atmosphere can be derived from observations of optical lines and, in particular, hydrogen Hα line 6563 Å emission with red shifts[3–9]. Further space missions uncovered details of this increase in coronal line intensities with upward velocities reaching 1000 km s$^{-1}$ (refs 3–6,10–14). Often flares show soft X-ray emission of highly ionized ions of FeXXIV or even FeXXV[15,16], which can be only produced by non-thermal processes[17] linked to energetic particles.

Observations of flare emission with blue-shifts in coronal lines and red-shifts in chromospheric lines were interpreted by hydrodynamic (HD) responses of flaring atmospheres to heating by particle beams injected from the top and precipitating to lower atmospheric levels[18–22]. There are three types of hydrodynamic models defined by their initial conditions, from which heating starts: type 1 uses the quiet Sun chromosphere[18,19,20,23] (in Lagrangian coordinates), which is converted by electron beam heating into a flaring atmosphere with its own corona, transition region and chromosphere; type 2 uses a pre-heated flaring atmosphere comprising of semi-empirical model VAL F[24] in the chromosphere and the quiet Sun (QS) corona attached above the transition region in Lagrangian[21] or linear[22,25] coordinates, which is also heated by precipitating beam electrons; type 3 uses an isotropic atmosphere evenly heated over a linear depth by some unspecified agents[26].

There are three types of heating function: H1—by beam electrons in Coulomb collisions with electron density derived from a continuity equation approach (CEA)[18,23,27]; H2—by beam electrons in Coulomb collisions with density derived from a flux conservation equation[20–22,28], which has a serious (infinity) limitation[29,30] at the stopping depths in the chromosphere for electrons with the lower cutoff energy; H3—by unspecified agents with equal energy deposition per volume at any depths[14,26]. Heating by particle beams is considered to be either impulsive of 5–10 s[18,21–23], or prolonged (30–300 s)[14,19,20,25,26] accounting for different types of flaring events.

The cooling in all hydrodynamic models is provided by radiation from the corona and transition regions, calculated in optically thin emission for the solar abundances[31]. The additional cooling by hydrogen line emission in the chromosphere is calculated by solving radiative transfer equations[21,22], or by adding hydrogen radiative losses for relevant beam parameters as arrays to the cooling function[23]. A hydrodynamic timescale (10–100 s)[32,33] is much longer compared to a radiative timescale (0.3 s)[32,34] that supports a consequential use of hydrodynamic and radiative models.

Heating of the QS chromosphere by beam electrons (HD model type 1)[18,19,23] is shown to sweep plasma to lower atmosphere, forming a flaring atmosphere with the new corona, transition region and chromosphere. This sweeping is followed by the plasma evaporation back to the corona combined with formation of a low-temperature condensation in the chromosphere moving as a shock to the photosphere. A hydrodynamic heating in the other two types of models (preheated and isotropic) results in chromospheric plasma evaporation without sweeping, combined with the shock moving downwards to the lower atmosphere with smaller velocities[14,21,22,25,26].

Most HD models[14,18,20,21,22,25,26] account quite well for evaporation (upward) velocities and intensities of extreme ultra-violet emission. However, types 2 and 3 models are less successful in interpreting the red-shifted Hα line profiles[7–9,35]. Earlier calculations of Hα line profiles[36–38] carried out for pre-heated hydrodynamic atmospheres[21], with heating function by Nagai and Emslie[20], showed the simulated profiles with blue-shifts, contrary to the red-shifts observed[7–9,35]. These discrepancies were previously attributed to a complex geometric multi-thread structure of flares[36–38].

The advances in space and ground-based instruments with high spatial and temporal resolution (Interface Region Imaging Spectrograph (IRIS)[39], the Atmospheric Imaging Assembly (AIA) aboard on the Solar Dynamic Observatory (SDO)[40] and notably the CRisp Imaging Spectro-Polarimeter[41] located at the Swedish 1-m Solar Telescope (SST)[42,43]) helped to eliminate some effects of spatial inhomogeneities in flaring regions emitting Hα lines[22,44–47]. Rubio da Costa et al.[47] reported simulated Hα line profiles with a small red-shift (about 15 km s$^{-1}$) at a flare onset and larger blue-shifts at 52 s later. However, this model still cannot explain the Hα line observations[7,8,35] with larger red-shifts taken at the flare onsets.

The radiative models describing hydrogen emission in flares utilize the effects of electron beams in two ways: via heating of the ambient plasma by beam electrons as considered in HD models, and via non-thermal ionization and excitation of hydrogen atoms by beam electrons for a flux conservation approach (FCA)[48] and for a CEA[27,49]. The heating and non-thermal excitation and ionization rates of hydrogen atoms are significantly affected by the approaches used for particle kinetics, producing in FCA smaller electron numbers at chromospheric levels compared to CEA. This occurs because of the electron number truncation in FCA at the upper chromosphere[29,30], before a stopping depth of lower energy electrons, in order to avoid the infinite heating[29]. This, in turn, shifts to the upper chromospheric depths the effect of beam electrons on hydrogen emission in FCA.

The CEA provides very smooth distributions of beam electrons at all precipitation depths, with maximum heating occurring in the chromosphere at the stopping depth of electrons with a lower cutoff energy[27]. This heating leads to formation of hydrodynamic shocks in the middle chromosphere, where the Hα line cores are formed, contrary to the hydrodynamic models type 2 (refs 21,22) using FCA, where this shock is formed in the upper chromosphere. Therefore, current radiative hydrodynamic models of type 2 and 3 cannot explain the red-shifted Hα line profiles observed at flare onsets reported since early 80s.

In this paper we confirm the earlier observations[7,8,35] by presenting Hα line profiles with strong red shifts recorded using SST for a flaring event onset in a C1.5 flare. These profiles are interpreted with a simultaneous hydrodynamic model type 1 (ref. 23) and full non-LTE (NLTE) radiative model for 5 level plus continuum hydrogen atoms (model HYDRO2GEN) by considering hydrogen non-thermal excitation and ionization rates by beam electrons[49].

## Results

**Active region topology and hard X-ray emission.** The C1.5 class flare occurred on 30th June 2013 in the active region (AR) 11778 during the time 09:11–09:27 UT, as per the GOES light curve in the 1.0–8.0 Å channel (Fig. 1a: black line, with peak indicated by the grey horizontal line). The flare originated in a complex configuration of magnetic field with the opposite polarity connected to another active region located in the south east (Fig. 1c,d). The initial flare started at 09:13:54 UT (event 1) in the north-east location of the negative polarity region of the AR11778 (Fig. 1c).

At 09:15:54 UT it continued in the south-west location of the same region (event 2) (Fig. 1c), where the Hα ribbons were formed and the line profiles observed. A few minutes later hard

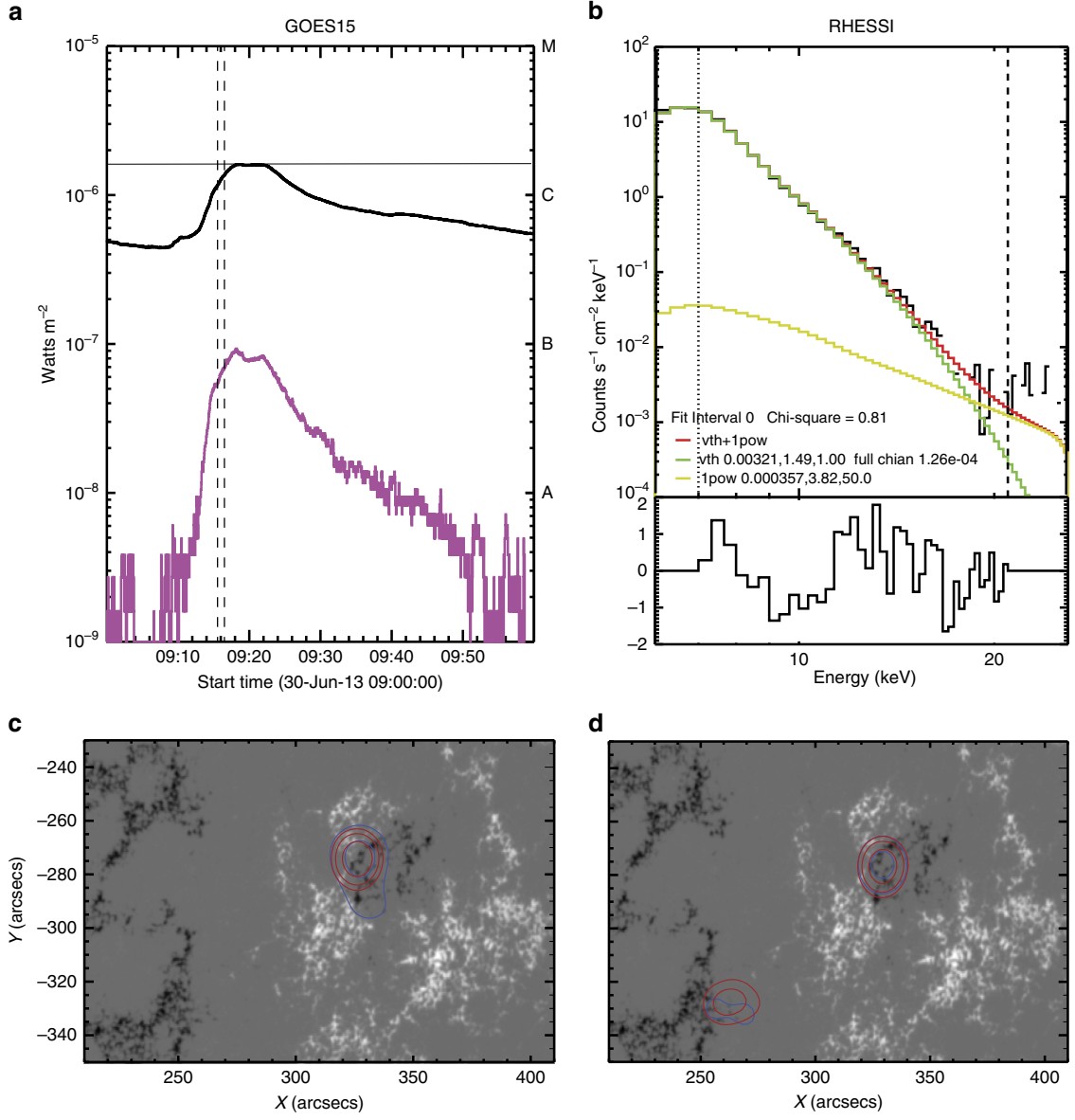

**Figure 1 | The active region topology and hard X-ray emission. (a)** The GOES X-Ray light curves of the flare in the 1–8 Å (black) and 0.5–4.0 Å (magenta) channels. The vertical dashed lines correspond to the time interval of the RHESSI spectrum in event 2. **(b)** RHESSI photon flux spectrum for event 2 with residuals derived with CLEAN in the 20 s interval around the time of Hα emission for thermal (green line) plus single power-law (yellow line) components, giving the total (magenta line). Hard X-ray emission is mostly of thermal nature with a small non-thermal component (see for details the current section and 'Methods section: Reduction of Hα line emission') with the parameters: spectral index about 3.8 and initial energy flux can be a factor (0.7–3) of $F_0 = 10^{10}$ erg cm$^{-2}$ s$^{-1}$. **(c)** The hard X-ray emission contours appearing in event 1 (top) and event 2 (Bottom, blue contour) coinciding with the times of the observations of Hα kernels with red-shifts in the ribbon (09:16 UT). These are overlaid onto the HMI magnetogram. The response in the 5–12 keV channel is shown using red contours, and the response in the 12–25 keV channel with blue. **(d)** Hard X-ray emission overlaid on the HMI magnetogram appearing with the event 3 occurring ~4 min later (09:20 UT), during the maximum in GOES light curve.

X-ray emission appeared at the location of the south east active region (event 3) (Fig. 1d). The GOES light curves include the contributions from all three events of this active region. Event 2 (Fig. 1c) contributes to this light curve at the times indicated between the vertical lines in Fig. 1a. The data from the helioseismic and magnetic imager (HMI) did not detect any sunquakes[50] in these events.

Figure 1b displays the hard X-ray photon spectrum for event 2 measured by RHESSI with detectors 4, 5 and 9. The spectrum was fitted from 09:15:54 to 09:16:14 UT over the energy range of 7–21 keV using object spectral executive and thermal (green line) plus single power-law (yellow line) components, giving the total (magenta line). The background period was 09:38:40 to 09:40:56

UT. The photon spectrum for event 2 can be also fitted by the thermal function only with the similar accuracy ($\chi^2$). This indicates that hard X-ray emission in the vicinity of event 2 has a strong thermal component related to a difference in spatial resolution for hard X-ray and Hα observations (see methods section 'Reduction of Hα line emission'). For this reason, the hard X-ray energy spectrum presented in Fig. 1b is for a demonstration only of a weak non-thermal component with spectral index of 3.8 and a lower cutoff energy of about 7–10 keV.

The hard X-ray contour images in Fig. 1c,d were made using the CLEAN algorithm and detectors 3–8 with 20 s integration times. The contour levels in 6–12 keV (red) and 12–25 keV (blue) are at 30, 50 and 70% of the maximum intensity, covering the

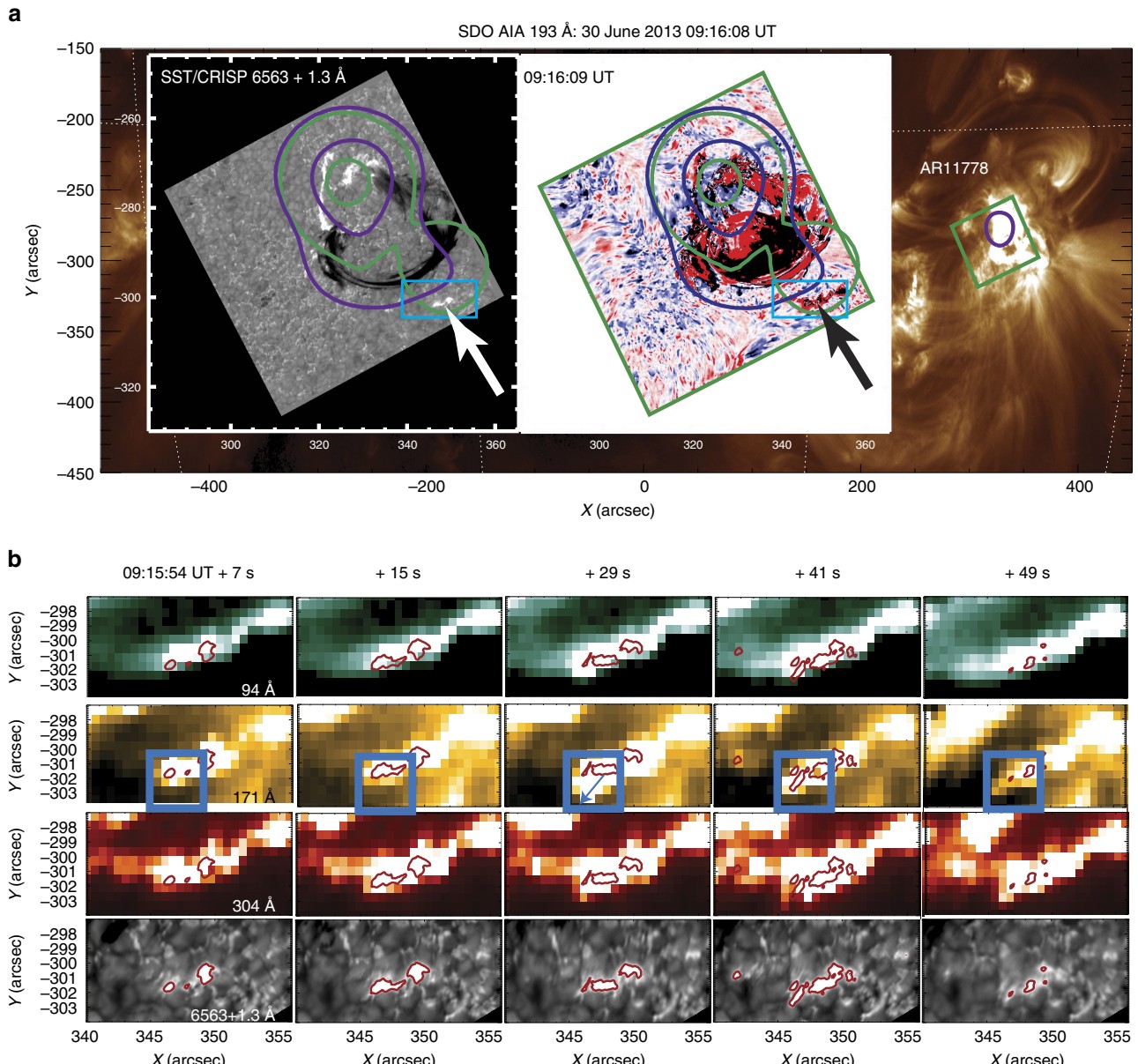

**Figure 2 | Observations with AIA. (a)** A context image for the observation in AIA 193 Å overlaid with the CRISP FOV outlined in green within AR11778. Inset left: The co-temporal (09:16:09 UT) CRISP image in the Hα line far red wing reveals bright flare ribbons point to by the white arrow for event 2 that are co-spatial within RHESSI imaging contours in 6–12 keV (green) and 12–25 keV (purple). Inset right: The Hα dopplergram for the 33 pt. spectral scan per pixel, containing blue/red-shifted motions marked by the relevant colour presented in the range of ± 20 km s$^{-1}$. The blue boxes in the insets (**a**) highlight the section of the ribbon formation in event 2, which is displayed in **b**. (**b**) The image sequence describing evolution of the ribbon in the AIA 94, 171 and 304 Å channels from top to bottom, respectively. These are co-spatial and co-temporal with the bright ribbon features (contoured in red), in the Hα far red-wing images of + 1.3 Å. The 171 Å channel reveals a bright jet-like protrusion (within the blue boxed region) that appears to form between the time frames 09:15:54 UT + 15 s and + 29 s (corresponding to 93 km s$^{-1}$) in the direction of the blue arrow and disappears by the time frame + 49 s.

area 6–8 pixels for the latter. The initial energy flux for event 2 was about $10^{26}$ erg s$^{-1}$. The range of initial energy fluxes $F_0$ for this event 2 is discussed in the Methods section 'Reduction of Hα line emission'.

**Hα line and coronal jet images.** *Hα images.* The Hα line observation sequence occurred from 09:15.54 UT to 10:17:18 UT and was carried out by SST using the CRisp Imaging Spectro-Polarimeter (CRISP)[42]. CRISP is especially suited for spectroscopic imaging of the chromosphere in the popular Hα line (6,562.8 Å), being equipped with three high-speed, low-noise CCD cameras that operate at a frame rate of 36 fps. The C1.5 class

flare under investigation was captured in Hα line within the CRISP Field-of-View (FOV) of $55 \times 55$ centred at heliocentric coordinates (323.4″, − 287.9″) (Fig. 2a). We refer to Methods section 'Reduction of Hα line emission' for a description of the reduction technique used for the CRISP data.

*Coronal jet images.* The images obtained by AIA instrument aboard on the SDO AIASDO were used for the background in Fig. 2 to locate the Hα ribbons. To achieve sub-AIA pixel accuracy in the temporal and spatial co-alignment of CRISP images with AIA, the photospheric bright points common to both FOV were cross-correlated. The AIA images (Fig. 2b) for the hotter channels (that is, transition region—He ii 304 Å;

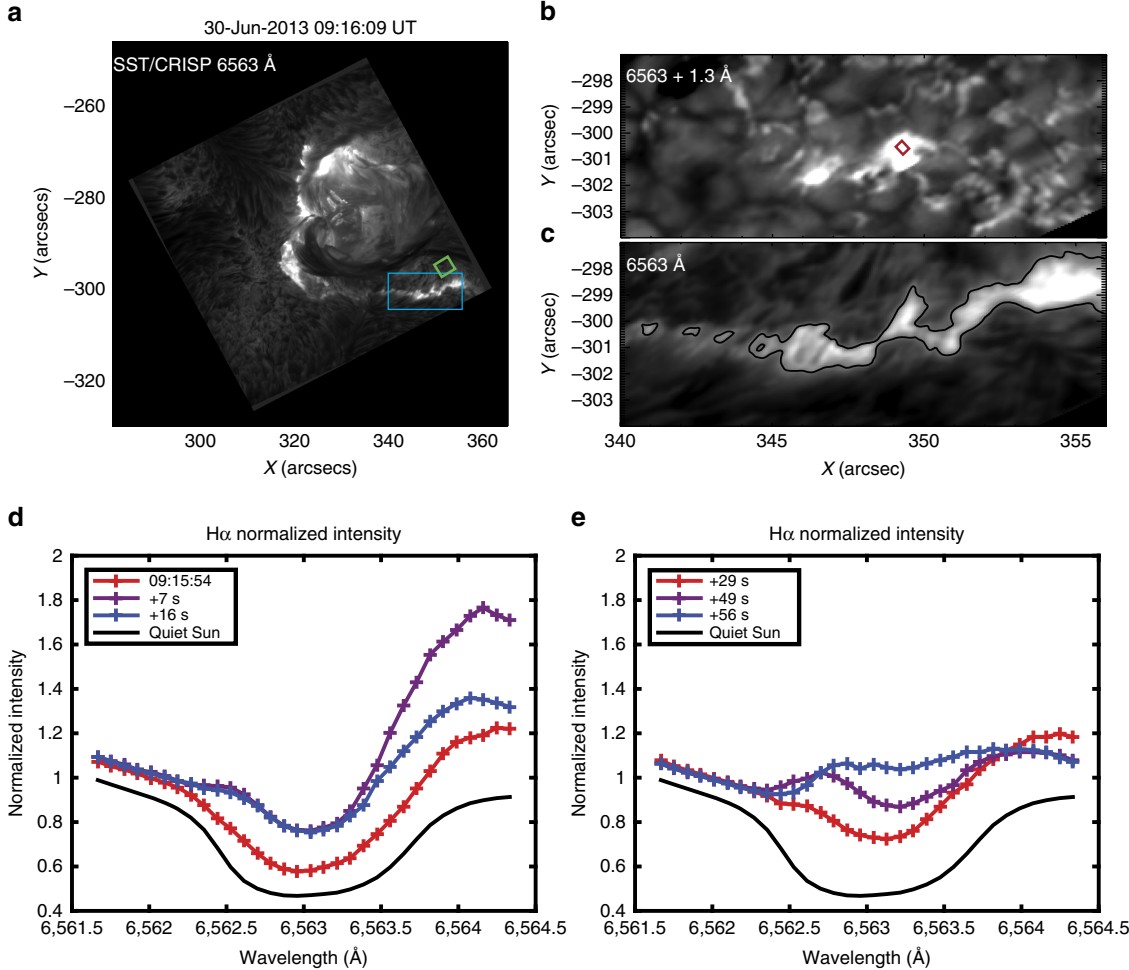

**Figure 3 | Hα line profile observations using SST.** (**a**) The CRISP Hα line core image (6,563 Å) with a blue box outlining the part of the flare ribbon under investigation. The green box corresponds to the pixels selected to construct the average quiet Sun spectral profiles, ie, close to the ribbon formation and free of any activity, within the time interval of the ribbon formation. (**b**) The corresponding FOV for the Hα far red wing intensity at +1.3 Å, with a red box corresponding to the region where the spectral profiles of interest are extracted. (**c**) The contoured ribbons of the Hα line core image for the blue box region is presented. (**d**) The averaged and normalized Hα spectral line profiles, determined from the red box pixels, are presented for time intervals corresponding to the 1st (09:15:54 UT: red solid line), the second (+7 s: purple solid line) and the third (+16 s: blue solid line) time frames. The Hα line profiles display exceptionally strong red-shifts. (**e**) The averaged and normalized Hα spectral line profiles for significantly later time frames corresponding to +29 s (red solid line), +49 s (purple solid line) and +56 s (blue solid line) when there were no longer strong red-shifts but rather core emission with peaks in both blue and red near wings. The black solid lines describes the averaged QS background Hα profile, deduced from the region defined by the green box in **a**. Intensities were normalized against the background levels using the QS intensity of 9,890 counts per pixel at 6561.7 Å; as a reference.

Corona—Fe xii 171 Å; Flaring/hot Corona—Fe xxiii 94 Å) were reduced and aligned to 1,700 Å, via the *aia_prep* routine in SolarSoft Interactive Data Language (SSWIDL). Subsequent images in all AIA channels were de-rotated to the CRISP start time (Fig. 2). The SST telescope turret continually tracked the starting target. Therefore, throughout the observation the CRISP image sequences are excellently co-aligned with AIA and RHESSI images.

**Hα-line profiles.** The CRISP observation of event 2 (see 'Active Region topology and hard X-ray emission' and Fig. 1) began at 09:15:54 UT, just before the peak of flux in the GOES light curve produced by all three events. The Hα line emission was subjected to data reduction and normalization (see Methods section 'Reduction of Hα line emission'). Figure 3a shows the full CRISP field of view image in the Hα line core (6,563 Å) at 09:16:01 UT. The green box shows the $31 \times 31$ pixel square (1,333 km$^2$) used for the QS reference intensity, which had no interference from overlying structures during the relevant observational frames. The blue rectangle in Fig. 3a displays the zoomed field of view

used in panels b and c. In Fig. 3b we see the image taken in the red wing of Hα at 6,564.376 Å and Fig. 3c shows the line core. To assess the feature identified, data was extracted from a $5 \times 5$ pixel square (215 km$^2$), which contained the region of the greatest red wing enhancement in the 09:16:01 UT frame. This kernel area is highlighted by the red square in Fig. 3b.

The resulting Hα line profiles are shown in Fig. 3d,e. The CRISP observation captured the onset of a strong chromospheric downflow in the second ribbon area highlighted by the blue box in Fig. 3a. The red wing enhancement started in the 09:15:54 UT frame (Fig. 3d, red line), increased between 09:15:54 and 09:16:01 UT and peaked at 09:16:01 UT (Fig. 3d, purple line), 7 s after the flare onset. Contrary to the symmetric Hα line profile of the QS, the emission in the red wing exhibited a single-peaked profile (Fig. 3d). This suggests that the peak can be attributed to a strong downflow in the chromosphere with Doppler velocity of 45–50 km s$^{-1}$. This red wing enhancement was reduced 9 s later, while the core emission remained at a slightly raised level, compared to the QS (Fig. 3d, blue line).

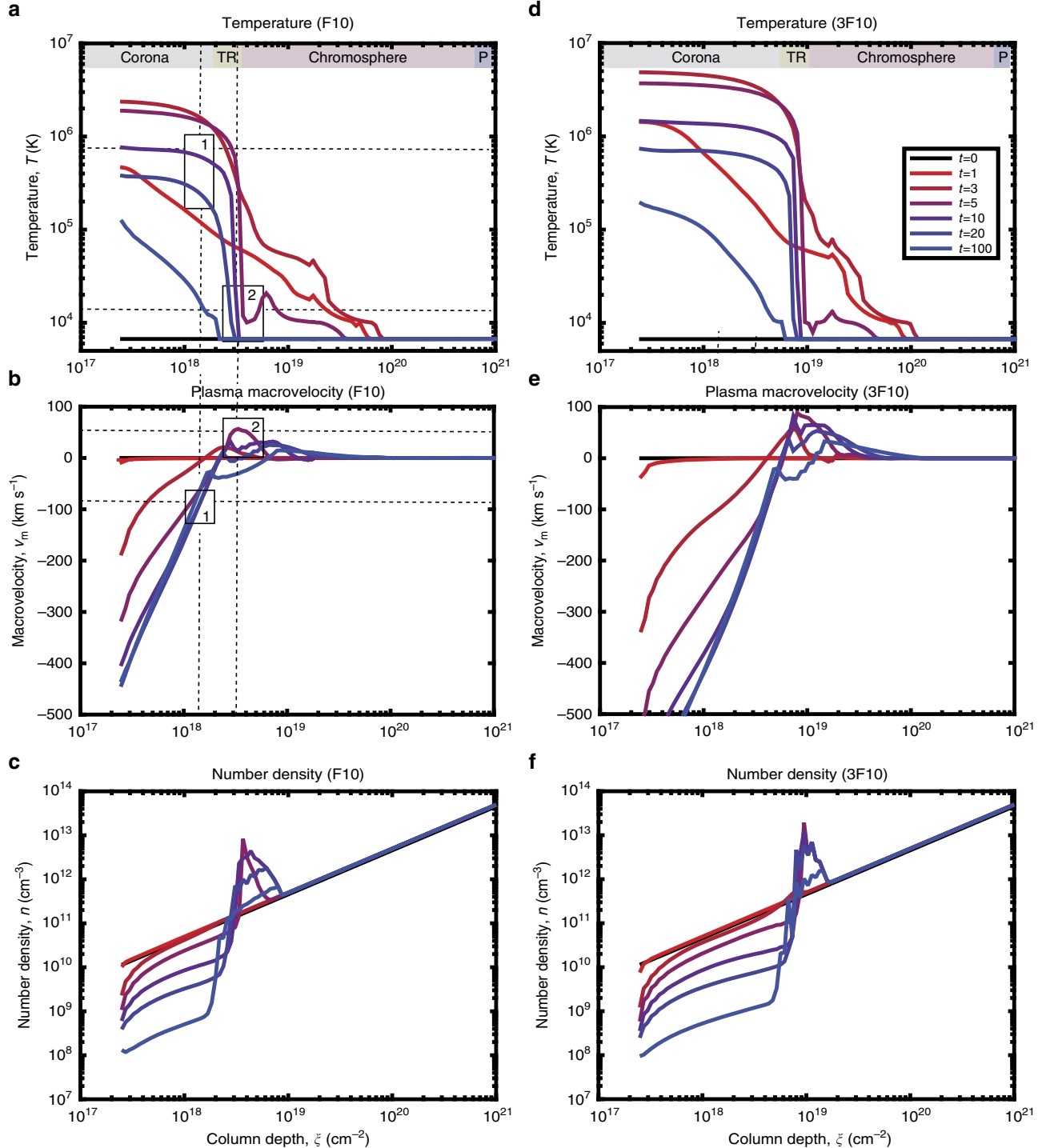

**Figure 4 | Simulated hydrodynamic responses.** The simulated hydrodynamic responses of a flaring atmosphere to injection of a beam with the initial flux of $10^{10}$ erg cm$^{-2}$ s$^{-1}$ (F10 model, left panels) and $3 \times 10^{10}$ erg cm$^{-2}$ s$^{-1}$ (3F10 model, right panels) following Zharkova and Zharkov[23] showing column depth dependencies of: (**a,d**)—the electron kinetic temperature, K, (**b,e**)—the plasma macrovelocity, km s$^{-1}$ and (**c,f**)—the plasma number density, cm$^{-3}$ forming a flaring corona, chromosphere and photosphere (see the text for more details).

Throughout observations the blue wing had only a slightly raised intensity (without peaks) compared to the QS, in agreement with the wing intensity enhancement, or background level increase, appropriate to flares. After 29–56 s, the red wing enhancement was reduced towards the flare background level and the core intensity was increased for the times when the H$\alpha$ line was in emission (Fig. 3e, blue line).

**Hydrodynamic response**. The method for calculation of a hydrodynamic response of the flaring atmosphere to injection of power-law beam electrons is described in the Methods section 'Hydrodynamic response to heating by an electron beam'.

Figure 4 shows plots of electron kinetic temperatures (a and d), macrovelocities (b and e) and plasma number densities (c and f) as functions of column depth calculated as a hydrodynamic response of the ambient plasma to injection of a power-law beam

with the initial flux of $10^{10}$ erg cm$^{-2}$ s$^{-1}$ (F10 model, left panels) and $3 \times 10^{10}$ erg cm$^{-2}$ s$^{-1}$ (3F10 model, right panels). The initial QS chromosphere density is indicated by the straight lines in Fig. 4c,f. The flaring transition region is swept by the beam towards $3 \times 10^{18}$ cm$^{-2}$ (F10 model) or $9 \times 10^{18}$ cm$^{-2}$ (3F10 model), with the flaring chromosphere extending to $8 \times 10^{19}$ cm$^{-2}$ (F10 model) or $2 \times 10^{20}$ cm$^{-2}$ (3F10 model) followed by a flaring photosphere (Fig. 4).

Temperatures in the flaring corona are strongly increased compared to the initial chromospheric temperature, with the magnitude scaled proportionally with the beam initial flux (compare Fig. 4a,d). While the ambient density is significantly reduced from the initial QS chromospheric magnitude ($10^{10}$ cm$^{-3}$) to $10^9$–$10^8$ cm$^{-3}$ to form the new corona of a flaring atmosphere[18] (Fig. 4c). These trends are similar to hydrodynamic models heated by electron beams with the same parameters reported by Fisher et al.[21]. The beams with moderate initial fluxes considered in this study do not heat the flaring corona to 10 MK (Fig. 4a,d) that is fully acceptable according to the statistical analysis of soft X-ray emission in flares[17]. However, our hydrodynamic model heated by beams with the initial energy fluxes of $10^{11}$ erg cm$^{-2}$ s$^{-1}$ or greater is proven to produce coronal temperatures of 10–20 MK (refs 18,23,51).

The upward motion of a flaring plasma is reflected in the macrovelocity plots (Fig. 4b,e) showing evaporation of chromospheric plasma upwards to the newly formed corona at the column depths between $10^{17}$ and $10^{19}$ cm$^{-2}$) (Fig. 4b, area below the box 1). This evaporation lasts, in general, for a few thousand seconds expanding upwards with increasing velocities even after the beam is stopped[18,23]. The evaporation velocities range from a few tens of km s$^{-1}$ (at 1 s) to four hundred km s$^{-1}$ (at 20–100 s). The evaporation will increase the coronal density at later times ($> 3$–5 min) as reported from observations[14,19].

At the same time, the beam energy deposition leads to formation of a low-temperature condensation in the flaring chromosphere seconds after beam injection begins (Fig. 4a,b, box 2) with a slightly increased temperature up to $10^4$ K. This condensation moves as a shock towards the photosphere and interior[51] with velocities from 30 to 35 km s$^{-1}$ (7F9 model) up to 50 km s$^{-1}$ (F10 model) (Fig. 4b) and up to 90 km s$^{-1}$ (3F10 model) (Fig. 4e). The density of this shock is about $10^{13}$ cm$^{-3}$ (Fig. 4c). This is different from the results of HD models of type 2, where the shock is formed at upper atmospheric depths[14,22,25], because of their different initial atmospheres and heating functions (see Introduction section). However, both types of hydrodynamic models (1 and 2), when simulated for a longer time (above 100 s considered in this paper) consistently show chromospheric plasma evaporation to the corona[18,23,51] with similar velocities (up to 1000 km s$^{-1}$ (refs 14,21) or up to 1500 km s$^{-1}$ (refs 18,23,51)).

**Probing hydrodynamic results with the AIA observations.** In the considered HD model plasma evaporation (Fig. 4a,b, box 1) (that can be called 'smooth evaporation'[14,18,21]) starts first from the second of the beam injection and continues for 100 s (and above, not shown here). For F10–3F10 models it reaches velocities of 50–100 km s$^{-1}$ in the lower flaring corona and several hundred km s$^{-1}$ in the upper flaring corona (Fig. 4).

AIA observations of event 2 in 94, 171 and 304 Å channels presented in Fig. 2 have shown rather variable signatures. A bright, transient jet-like protrusion of plasma from the ribbon in the 171 Å AIA channel was detected between 15 and 29 s after the event onset, which appeared linked to the strong downflow regions in Hα emission (red contours) (Fig. 2b). At the same time, there are no jets seen in the 94 or 304 Å emission.

The jet velocity in 171 Å, measured 29 s after event 2 (beam injection) began, was 93 km s$^{-1}$. This was derived from the apparent motion of the jet within the AIA image set in the 171 Å channel. The error in measurements is sensitive to a pixel size (0.6″), reaching about $\pm 30$ km s$^{-1}$ in the time frame of jet propagation. This estimation is accounted for by a height of the box 1 within Fig. 4b, which shows the macrovelocity within a range of 63–123 km s$^{-1}$ centred at 93 km s$^{-1}$. This velocity is close to other upflow observations of 100 km s$^{-1}$ derived for flares with the similar beam parameters[10,11].

Comparison of the models presented in Fig. 4a,b shows that the coronal temperature variations for 3F10 model would not account for the observed jet in the AIA 171 Å emission. However, the temperature profile evolution for F10 model between 5 and 100 s shown in Fig. 4a, box 1, reveals that the plasma can be detectable in the temperature range of $\log T = 5.2$ to $\log T = 6.05$ at the depths of the low flaring corona. The 171 Å channel is the most sensitive to this range, compared to other available AIA channels (see Methods section 'AIA line synthesis'). Moreover, the velocity range derived from the AIA 171 Å channel, averaged at 93 km s$^{-1}$, closely resembles the predictions of F10 model of a hydrodynamic response to plasma heating by an electron beam for a given temperature range, as shown in Fig. 4a.

From the hydrodynamic simulations the response in 94 Å channel is expected to be rather weak. This is because the 94 Å emission is detected at a secondary sensitivity peak, at 1 MK relevant for the flaring corona in this event, and not at the main sensitivity peak of 10 MK (Fig. 6b, green line, Methods section 'AIA line synthesis'). There was slightly increased signal in the 94 Å protrusion (Fig. 3b, first row), which was most evident in 171 Å images (Fig. 3b, second row, blue arrow).

The fact that the jet-like feature was seen only in the 171 Å channel and not in 304 Å or not clearly in 94 Å channel can be explained by a fast (tens of seconds) reduction of the plasma temperature and density in the newly formed flaring corona caused by radiative cooling, thermal conduction and plasma motion[52,53]. Indeed, at the later times (20–30 s), after the beam is off, the coronal temperature was quickly reduced from two million to the sub-million Kelvin range (Fig. 4a, box 1), and the plasma density was also reduced from the chromospheric ($10^{10}$ cm$^{-3}$) to coronal $10^9$ cm$^{-3}$ density (Fig. 4c) (see Methods section 'AIA line synthesis').

The cooling process in the hydrodynamic model can quickly reduce the differential emission measure of a flaring corona, as demonstrated in Fig. 6 of Somov et al.[18], allowing the coronal emissivity to reach the range matching the AIA sensitivity window (see Methods section 'AIA line synthesis'). This made the plasma upflow detectable only in the AIA 171 Å passband at 29 s after the event onset, when the coronal temperature in a flaring corona is dropped to the AIA range. Although, the coronal temperature in a flaring atmosphere at this time remains still too high for the intrusion to be clearly seen in the 304 Å passband, it can be observed later ($> 100$ s) after further cooling.

**Simulated radiative response in the Hα line.** The simulated Hα line profiles were calculated for non-thermal excitation and ionization by an electron beam with the initial fluxes of $10^{10}$ erg cm$^{-2}$ s$^{-1}$ (F10 model), $3 \times 10^{10}$ erg cm$^{-2}$ s$^{-1}$ (3F10 model), representing an upper estimate of the flux and $7 \times 10^9$ erg cm$^{-2}$ s$^{-1}$ (7F9 model), representing a lower estimate with a beam spectral index of four (Fig. 5a,c), as suggested by the RHESSI and tuned by Hα observations (see Methods section 'Reduction of Hα line emission'). For calculation of Hα line profiles the full NLTE problem for five levels plus continuum hydrogen atom was solved for the simulated hydrodynamic models using the approach described in the Methods section

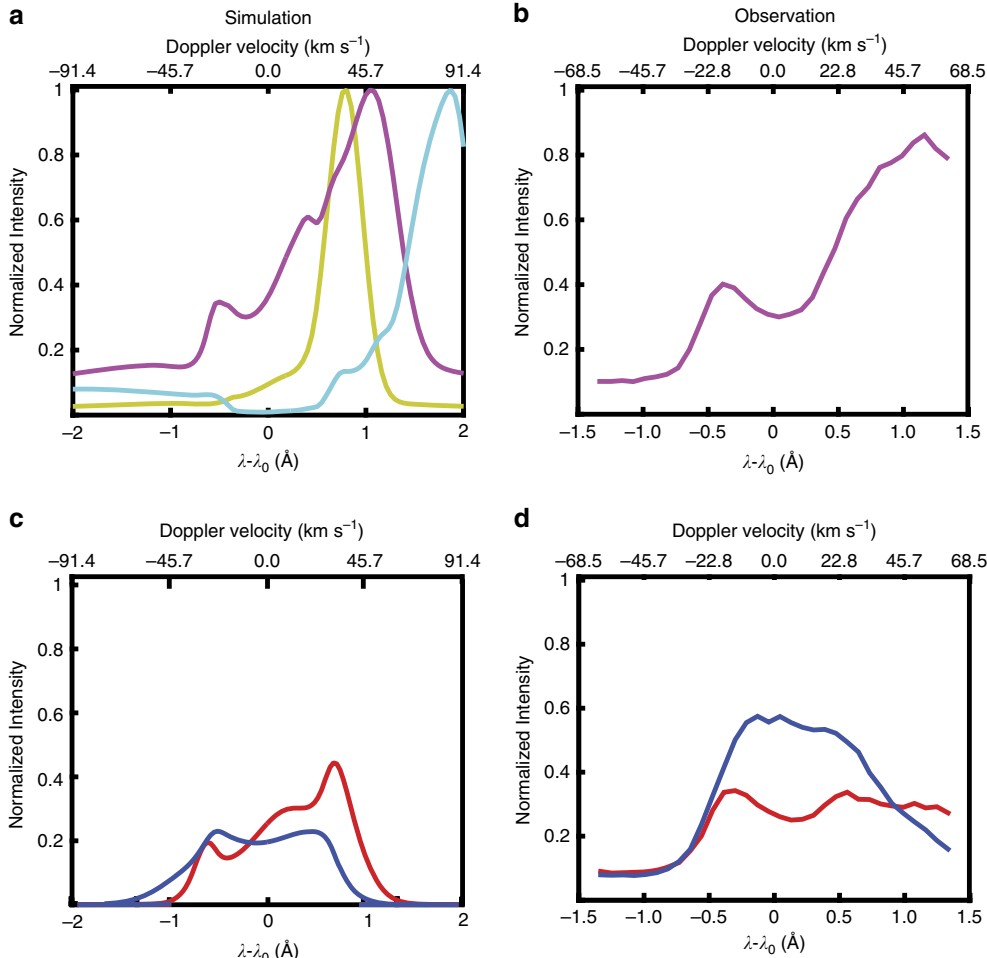

**Figure 5 | Simulated and observed Hα line enhancements.** (**a**) the synthetic Hα line normalized intensity versus a distance ($\lambda - \lambda_0$), in Å, from the Hα line central wavelength, $\lambda_0$ ($\lambda_0 = 6563$ Å) taken from the simulation at $+5$ s after a beam onset for the F10 model (magenta line), the 3F10 model (cyan line) and a model with initial flux $7 \times 10^9$ erg cm$^{-2}$ s$^{-1}$ (7F9 model, yellow line) (**b**) the normalized background-subtracted Hα profile observed $+7$ s after the ribbon onset in the event 2. (**c**) The Hα line normalized intensity simulated for the F10 model at later times after the beam onset: $+30$ s (red solid line) and $+70$ s (blue solid line) and (**d**) the observed Hα profiles at the similar times of $+29$ s (red solid line) and $+56$ s (blue solid line) after the event 2 onset.

'Radiative transfer method'. The simulated profiles are normalized in the similar way to the observed profiles, as described in the Methods section 'Reduction of Hα line emission'.

Non-thermal collisions between beam electrons and hydrogen atoms for all HD type 1 models cause excess excitation to the upper state ($n = 3$) of the Hα line transition, quickly converting the Hα spectral line from absorption into emission. The emission in the near wing wavelengths from the line centre have a lower optical depth and, thus, less absorption, resulting in the small intensity increase in the near wings ($\pm 0.5$ Å) (called 'horns') (Fig. 5). However, the main contribution of energetic beam electrons is the strong ionization of hydrogen atoms in a flaring atmosphere causing increase of their ionization degree by orders of magnitude[49]. This raises the density of the ambient electrons, compared to the density expected from their kinetic temperature. This, in turn, produces a significant increase of Hα line wing intensities owing to Stark's effect.

The radiative simulations clearly show that in the first seconds after the beam onset Hα line profiles are dominated by non-thermal ionization by the beam electrons and the downward motion of the shock (see Fig. 4b, box 2). For this flaring event the beam has a relatively low initial energy flux about $0.7$–$3.0 \times 10^{10}$ erg cm$^{-2}$ s$^{-1}$ resulting in a moderate increase of the Hα wing intensity (Fig. 5a). The horn in the near blue wing, about $-0.5$ Å

from the central line wavelength (Fig. 5a), is in a normal position to be caused by a radiative self-absorption as discussed above. However, the horn in the near red wing reveals a large increase of the intensity caused by a Doppler-shift of the emission wavelength caused by a downward movement of the hydro-dynamic shock (Fig. 4b, box 2) growing from 35 km s$^{-1}$ (7F9 model) up to 50 km s$^{-1}$ (F10 model) or 90 km s$^{-1}$ (for model 3F10) at the times of maximum beam deposition.

When the beam is switched off, thermal heating and slow recombinations of the ambient electrons with hydrogen atoms become the main sources of sustaining hydrogen atoms' excitation and, thus, Hα emission (Fig. 5c,d). One can see a decrease in the total intensity in the line compared to the intensity simulated during the beam injection (compare Fig. 5a,c). There is also a decrease of the red wing intensity over the subsequent 60 s (Fig. 5c). At later times in simulations, after the beam is off, Hα emission profiles become standard thermal profiles, exhibiting after 70 s a small intensity enhancement in the blue horn (Fig. 5c, blue line).

**Comparison with Hα line observations.** The simulated Hα line profiles were compared with the profiles observed by CRISP by averaging the emission over all the pixels in the red box of Fig. 3b (with the QS background intensity subtracted) during the flare

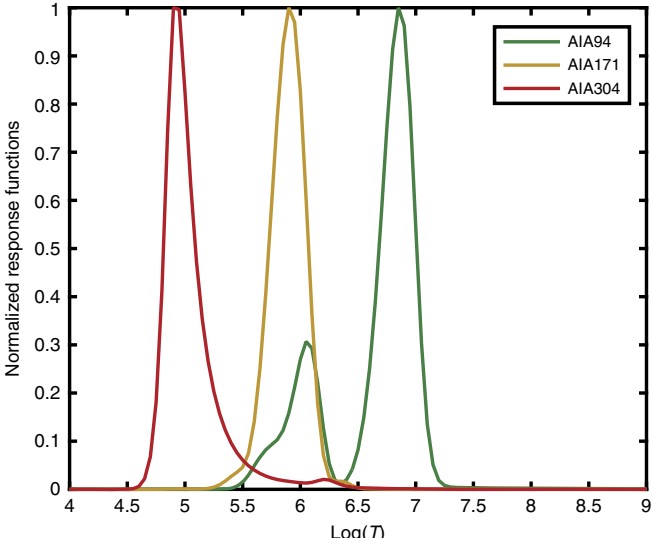

**Figure 6 | The normalized response functions of the AIA channels.** The normalized response functions of the AIA 94 Å (green line), 171 Å (yellow line) and 304 Å (red line) channels plotted against $\log_{10}$ of T (temperature). The AIA 94 Å channel has its largest sensitivity peak close to 10 MK but it is not limited in sensitivity to that specific temperature. It is shown in the green line that the AIA 94 Å channel has a secondary peak with maximum at the temperature of 1–2 MK.

onset (Fig. 5b) and over the next hundred seconds (Fig. 5d). The simulation produces intensities of Hα line emission from a flaring atmosphere within the spectral range ($\pm 3.0$ Å from a central wavelength) that is broader than the observational range ($\pm 1.5$ Å). The simulated profiles are shown to $\pm 2$ Å to demonstrate that the emission profile extended into a far red wing beyond the range (1.5 Å) defined by CRISP's current spectral filter.

The red-shift in the simulated Hα line profile reaches a maximum at (or just after) 5 s of the electron beam onset when the downward velocity in the hydrodynamic model is maximal (Fig. 4b, box 2). Only for the F10 model the shape of simulated Hα line profile and the magnitude of the red shift is closely matched by the Hα line profile observed by CRISP over the similar interval (7 s) after the event onset (Fig. 5d), while the beam with lower or higher energy fluxes produce much smaller or much higher red shifts, than those observed.

The Hα line core in model F10 is formed at depths of a HD shock, whose downward motion makes the line intensity redshifted by around 1 Å from the central wavelength ($\lambda_0 = 6563$ Å), corresponding to a Doppler velocity of 47 km s$^{-1}$ (Fig. 5a). This is very close to the velocities of 45–50 km s$^{-1}$ derived from the observed profile (Fig. 5b). Hence, we present the first successful interpretation of Hα line red-shifted profiles observed at the onset of a flare that has been long overdue for the past three decades[7,8].

This comparison confirms that the observed red shift in Hα line can be only caused by beam electrons with the initial flux close to $10^{10}$ erg cm$^{-2}$ s$^{-1}$. Measuring Doppler shift of the emission outside of the horns of the Hα line profile is an alternative method for determining the parameters of electron beam, allowing us to tune the estimations of initial energy flux derived from the low-resolution RHESSI data (see Methods section 'Reduction of Hα line emission').

## Discussion
In this study we presented multi-wavelength observations of a flaring event onset obtained with the highest temporal and spatial resolution from CRISP/SST, AIA/SDO and RHESSI. The C1.5

class flare observed on 30 June 2013 in AR 11778 produced three flaring events, which contribute to its hard X-ray and soft X-ray light curves. The flaring event 2 produced two Hα ribbons, in one of which Hα line profiles were recorded in $5 \times 5$ pixels using CRISP/SST with the maximum downward velocity of 45–50 km s$^{-1}$. There are also plasma upflows of 93 km s$^{-1}$ observed in the 171 Å AIA channel 29 s after the event onset, occurring just above the Hα line ribbon with the downward motion.

These observations were successfully interpreted with the combined hydrodynamic and full NLTE radiative models (HYDRO2GEN) affected by power-law electron beams. The beam parameters for this event are estimated using the hard X-ray photon spectrum observed by RHESSI and tuned with the high-resolution Hα observations. Our simulations show that for this flaring event heating of flaring atmosphere by beam electrons in the HD model starts from the QS chromosphere, converting it into a flaring atmosphere with its own corona, transition region and chromosphere. Beam electrons quickly sweep the ambient plasma to deeper atmospheric layers causing, in turn, a fast upward motion of the swept plasma back to the corona and downward motion as hydrodynamic shocks[18,23].

The upward motion, which occurs from the first seconds after a beam onset, reflects the chromospheric evaporation caused by a hydrodynamic response of the flaring atmosphere to heating by electron beam. The chromospheric plasma in this upward motion for this flaring event is observed injected into a flaring corona 29 s after the event (or beam) onset that fits very well our hydrodynamic model and the sensitivity windows of AIA in the different channels (94, 171 and 304 Å). The plasma jet becomes only visible in the AIA 171 Å channel, at the times when the temperatures and densities of the flaring corona are reduced to the magnitudes detectable within this AIA passband.

Additional support to the proposed HYDRO2GEN model is provided from fitting the observed Hα-line profiles with large red shifts with the simulated profiles obtained from a full NLTE approach applied to 1D flaring atmospheres being a HD response to electron beam heating. The Hα line in flaring atmospheres is shown to be dominated by: first, an increase in the line wing intensities due to Stark's effect caused by non-thermal ionization of the ambient hydrogen by beam electrons, and second, a hydrodynamic shock motion downward leading to large Doppler-shifts. The combination of these effects for this flaring event produces a big increase of the Hα line intensity in the red wing at about 1 Å from the line central wavelength, corresponding Doppler velocities of 45–50 km s$^{-1}$ derived from the observation. The latter is closely reproduced by the simulations only for the model F10, clearly restricting the initial energy flux of beam electrons capable of accounting for such a red shift. In addition, this close fit highlights a need to extend the spectral windows for observations of Hα line dynamics in flaring atmospheres, which will allow capture of the profiles with large red shifts occurring in the first 100 s of a flaring event.

It should be noted that the ratio of red-to-blue wing intensities of the simulated Hα line profile is slightly higher than in the observed profile, by a factor of 1.2. In addition, the wavelength of the central reversal (with the maximal absorption) in the simulated Hα line profile at 5 s is slightly blue-shifted from the central wavelength, compared with the observations (compare Fig. 5a,b). Such blue shifts of the central reversals in Hα lines could be real as they were also observed by Ichimoto and Kurokawa[7] for the profiles with strong red shifts (see their Fig. 4a at 00:19:59 UT).

It appears that small blue (or red) shifts of the central reversals can reflect the overlying Hα-line emission with strong upward (or downward) motions produced by different layers of a flaring event, so that their superposition could shift the central reversal

emission towards the blue or red wing, accordingly[7]. There is also a possibility that the Hα ribbon emission observed with SST occurred in a much smaller source size than the SST diffraction-limited resolution of 100 km in Hα. This could lead to over-smoothing, or averaging, of the observed Hα line intensity over a larger area than the real emission comes from that causes the differences in the observed and simulated ratios of the red-to-blue horn intensities and the blue-shifted central intensity.

While this scenario is plausible, it still assumes that the observed red-to-blue intensity ratio is perfectly accurate, which may not be the case, given that we do not have infinite spatial resolution in the SST observations. Therefore, this outstanding issue cannot be fully reconciled beyond the limits of the current state-of-the-art SST observations and needs to be progressed with observations by the instruments with higher resolution, such as the Daniel K. Inouye Solar Telescope (DKIST)[54].

This study provides a close interpretation of large red-shifted Hα line observations of solar (and possibly stellar) flares indicating a need for broader spectral windows capable to fully capture the dynamics of flaring events.

## Methods

**Reduction of Hα-line emission.** The Hα observations consisted of equidistant scanning of 33 wavelength points from $-1.38$ Å to $+1.38$ Å about the Hα line centre, resulting in an effective observation cadence of $\sim 7.27$ s. The image quality of the time series data significantly benefited from the correction of atmospheric distortions by the SST adaptive optics system[41]. Post-processing was applied to the data sets with the image restoration technique multi-object multi-frame blind deconvolution (MOMFBD)[55]. Consequently, every image is close to the theoretical diffraction limit for the SST with respect to the observed wavelengths. We followed the standard procedures in the reduction pipeline for the CRISP data[56].

The mean Hα profile intensities were taken in each of the 33 spectral positions of the QS and the flare kernel. Then the data were smoothed, to remove an instrumental spiking effect between adjacent spectral positions, by creating 32 interpolated spectral data points that are the mean of the two adjacent data points. Intensities were normalized against background levels using the QS average of 9,890 counts per pixel at 6561.7 Å as a reference. After 7 s, the kernel produced 10,949 counts per pixel at the 6561.7 Å spectral position, and 17,651 counts per pixel at the peak of the red-shifted intensity (6564.2 Å).

From the CRISP Hα red wing image taken at the time of greatest red wing enhancement (Fig. 3b), a strong, transient enhancement at $6563 + 1.3$ Å can be registered (depending on the emission level) in a range of 266–712 SST pixels with the resolution of 0.0592″. Then a single RHESSI pixel ($2'' \times 2''$) contains $33 \times 33 \approx 1100$ SST pixels. As the RHESSI area was too big (6–8 pixels) and the resolution too low, the areas of Hα flaring kernels for event 2 were used. Taking into account that $1 = 725$ km $= 7.25 \times 10^7$ cm, the area is estimated to vary within $(0.3–1.4) \times 10^{16}$ cm$^2$. This leads to the estimation of initial energy flux of hard X-ray emission for event 2 in the location of Hα ribbon of about $F_0 \approx (0.7 - 3.0) \times 10^{10}$ erg cm$^{-2}$ s$^{-1}$.

In order to derive the observed Hα profile, we used a flaring kernel in $(5 \times 5) = 25$ pixels of the SST event 2. The area covered by other $(1100-25) = 1075$ SST pixels (98% of a single RHESSI pixel) is the neighbouring area of this active region, which is not directly affected by this particular electron beam. This difference in the spatial resolutions of RHESSI and SST data also can explain a good fitting to thermal emission in the RHESSI data (coming from the pixels not associated with the Hα enhancements). Therefore, the RHESSI data should be (and was) only used for estimating the order of magnitude of the beam flux, while the other means confirming the precise beam flux are required, for example, from fitting of the Hα line profiles described in the section 'Simulated radiative response in the Hα line'.

**Hydrodynamic response to heating by an electron beam.** For physical conditions in a flaring atmosphere and with respect to findings from hard X-ray emission in the section 'Active Region topology and hard X-ray emission', we used the first part of HYDRO2GEN code simulating a 1D hydrodynamic response of the ambient plasma heated by a power law electron beam, adopted from Zharkova and Zharkov[23] (see their equations (3–6)) following the model by Somov et al.[18] updated with hydrogen radiative losses[49].

A hydrodynamic response of the ambient plasma in this event can also be caused by a high-energy thermal beam because the hard X-ray flux derived from RHESSI can be equally well fit by the thermal curve (see the section 'Active Region topology and hard X-ray emission'). However, as simulated by Somov et al.[57], the hydrodynamic response of a flaring atmosphere to a thermal beam is similar to that of a power-law beam, while raising the additional problem of thermal conductivity saturation. In order to avoid this problem, in our simulation for event 2, we chose to heat the flaring plasma by a power-law beam instead.

We consider a limited region of the QS chromosphere with column depths $\xi_{min} \leq \xi \leq \xi_{max}$, with $\xi_{min} = 10^{17}$ cm$^{-2}$ corresponding to the QS transition region, followed by the chromosphere and at $\xi_{max} = 10^{22}$ cm$^{-2}$ corresponding to the upper photosphere.

The initial QS chromosphere is assumed to be in hydrostatic equilibrium and the temperature is constant,

$$v(0, \xi) = 0,$$
$$T_e(0, \xi) = T_i(0, \xi) = T_0 = \text{const}, \tag{1}$$

where the initial temperature $T_0$ was derived from the semi-empirical simulations[24] equal to 6,700 K (ref. 18).

The initial plasma density of the QS chromosphere is defined as follows

$$n(0, \xi) = n_{min} + h_0^{-1}(\xi - \xi_{min}), \tag{2}$$

where $n_{min} = 10^{10}$ cm$^{-3}$ and $h_0$ is the height scale in a flaring loop:

$$h_0 = \frac{k[1 + x(T_0)]T_0}{\mu g_s}, \tag{3}$$

where $k$ is Boltzmann's constant, $x$ is the ionization degree, $g_s$ is the solar acceleration of gravity, $\mu = 1.44 m_H$ is the average atom mass of the ambient plasma, $m_H$ is the mass of hydrogen atom. Let us neglect any heating fluxes at the top and bottom boundaries, eg,

$$\frac{\partial T_{e,i}}{\partial \xi}(i, \xi_{min}) = \frac{\partial T_{e,i}}{\partial \xi}(i, \xi_{max}) = 0, \tag{4}$$

Let us also consider the upper boundary at the initial time to be a free surface at the presence of the external pressure $p_{cor}$ described as:

$$p_{cor}(0, \xi_{min}) = n_{min}k[1 + x(T_0)]T_0. \tag{5}$$

The plasma velocity at the lower boundary is defined by the equation

$$\frac{\partial v}{\partial \xi}(t, \xi_{max}) = 0. \tag{6}$$

The plasma is heated by an electron beam precipitating from the top boundary with the heating function derived from continuity equation[27]. Plasma cooling is caused by the viscosity, or motion between electrons and ions[18] and radiative cooling[31] updated with the hydrogen radiative losses[49].

After solving this system of four partial differential equations with the initial and boundary conditions (1–6) for precipitating electron beam with given parameters (initial energy flux $F_0$ and spectral index $\gamma$) we obtain time-dependent distributions of electron $T_e$ and ion $T_i$ temperatures, ambient plasma density $T$ and macrovelocities, $v$.

Heating by electron beam is found to sweep the plasma from the QS chromosphere towards deeper atmospheric levels converting the QS chromosphere into a flaring atmosphere with its own corona, transition region and chromosphere[18,23]. This is different from the other hydrodynamic models (Type 2 and 3 from the Introduction section), which use semi-empirical (pre-heated) flaring chromospheres VAL F[24] with the attached QS corona as the initial condition[21,22]. Thus, they are skipping the phase of conversion of the QS chromosphere into a flaring corona, flaring transition region and flaring chromosphere. The pre-heated hydrodynamic models work perfectly well for flares with pre-flare events, while our model is more applicable for the initial flaring events without prior heating.

The parameters of an injected beam are selected close to the range of parameters of hard X-ray emission derived from RHESSI for event 2 (see section 'Active Region topology and hard X-ray emission'): a single power-law energy spectrum with a spectral index of about 4 based on a comparison of spectral indices using Fokker–Planck approach (see Fig. 11 in ref. 58), a lower cut-off energy of 7–10 keV (from the total range 7–21 keV recorded by RHESSI). The limits of initial energy flux of beam electrons in event 2 was estimated from hard X-ray emission utilizing the areas of Hα kernels because of strong contamination of hard X-ray with thermal emission of the background corona owing to a much lower spatial resolution of the RHESSI (2″) versus SST (0.06″) pixels (Methods section 'Reduction of Hα line emission').

Three hydrodynamic models were produced for heating by beam electrons with the initial energy fluxes covering the upper and lower estimates: $F_0 = 7 \times 10^{10}$ (7F9 model), $F0 = 10^{10}$ erg cm$^{-2}$ s$^{-1}$ (F10 model) and $F_0 = 3 \times 10^{10}$ erg cm$^{-2}$ s$^{-1}$ (3F10 model). The duration of beam injection is chosen as 10 s to match the fast rise in Hα emission. The initial energy flux of a beam varies as a triangular function in time, with maximum at 5 s (ref. 23). The NLTE simulations for these HD atmospheres (called HYDRO2GEN model) enabled us to verify the most applicable initial electron flux for the event 2 by comparing the observed Hα profiles with the simulated ones (see Fig. 5a and Discussion).

**AIA line synthesis.** The HD models of type 1 presented in this paper include a flaring corona that is obtained from a conversion of the QS chromosphere, rather than having initially an inherent corona in HD models of type 2. Although, it should be noted that a large area covered by AIA pixels is the neighbouring corona, because given the difference in spatial resolutions of the AIA (0.6″) and SST (0.06″) pixels, the minimum area covered by a single AIA pixel includes

$10 \times 10 = 100$ SST pixels. Within these, only 25 pixels contain Hα emission while the other 75 pixels are, in fact, the neighbouring corona rather than the flaring event for which our hydrodynamic model is applicable. Therefore, our model does not intend to explain the contributions of any neighbouring coronal pixels to the emission of a flaring corona captured also by AIA, because our model certainly is not intended and does not solve the quiet Sun coronal heating problem.

Each passband in the images from AIA detects plasmas with different emissivity defined by the local atmospheric temperature and density. The normalized instrumental response functions of the AIA channels are shown in Fig. 6, plotted

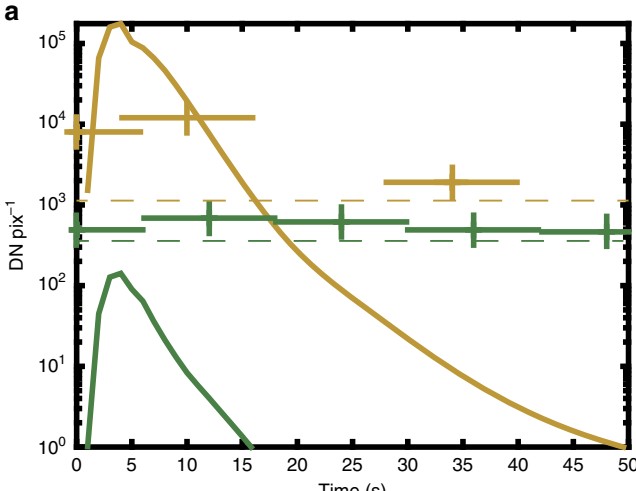

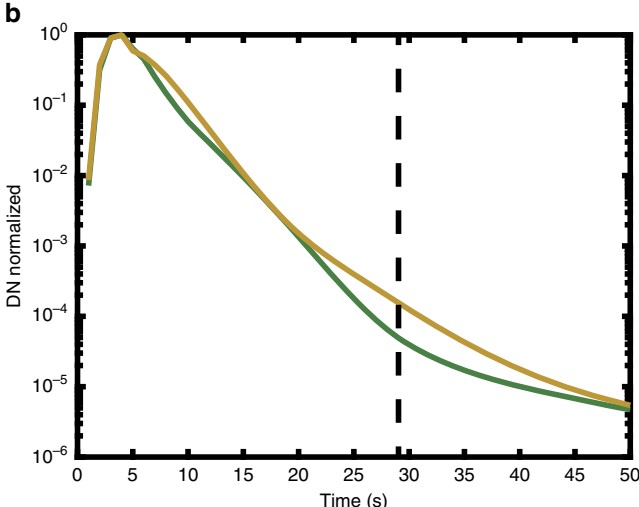

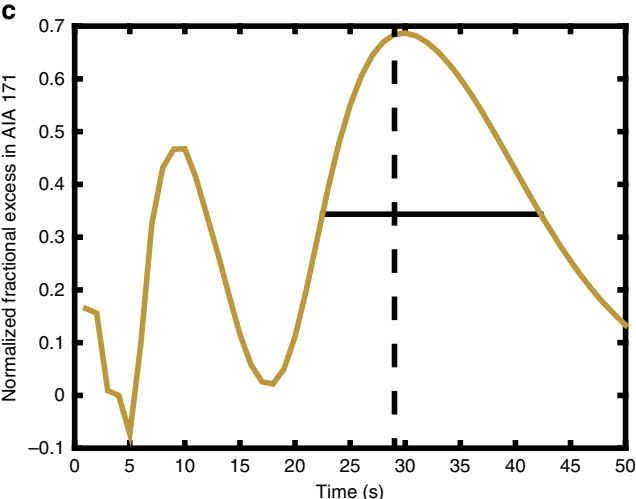

against $\log^{10}$ of T (temperature) for the spectral lines of interest (94, 171 and 304 Å). The AIA 94 Å channel has its largest sensitivity peak close to 10 MK combined with a secondary sensitivity peak close to the temperature of 1 MK (green line in Fig. 6), that is, it is sensitive to the flaring corona temperature peak (1–2 MK, see Fig. 4a) in the flaring event 2 derived from the F10 model. We therefore expect to observe slightly enhanced emission from the source of the beam injection in the 94 Å channel above the observed background level, in addition to a strong enhancement in the 171 Å channel over the background level (Fig. 7a).

The intensity responses, $I_i$ for each of the AIA channels $i$, were simulated following the AIA calibration method, described by Boerner et al.[59],

$$I_i = \int K_i(T(z)) n_e^2(z) \mathrm{d}z, \qquad (7)$$

where $K_i(T)$ is the temperature response function for the AIA channel $i$ (ref. 59), shown in Fig. 6, $n_e$ is the electron density, and the integral is performed over the height $z$, of the model atmosphere.

Figure 7a shows the simulated (green line) and observed (green crosses) light curves for AIA 94 Å channel, and likewise, in yellow, the light curves for the AIA 171 Å channel. There is a small increase above the background levels in the observed AIA 94 Å emission (green crosses) during the first 20–30 s (Fig. 7a). Because of the secondary sensitivity peak in AIA 94 Å, this flaring F10 model is capable of producing the AIA 94 Å enhancements on the order of $1 - 2 \times 10^2$ in intensity (DN units) (Fig. 7a, green line) at the temperature of 1–2 MK, hence, confirming that 94 Å emission is detectable from this model.

Most importantly, the F10 HD model leads to a much greater excess of intensity in the AIA 171 Å channel for some time during and after the beam injection (Fig. 7a, yellow line), as observed (Fig. 7a, yellow crosses). This AIA 171 Å enhancement should remain visible during the outflow (jet) process, and is indeed observed (see Fig. 2b, blue arrow). The simulated jet travels at $\sim 90\,\mathrm{km\,s^{-1}}$ and the observed jet traverses 3 pixels in the image space corresponding to $\sim 1500\,\mathrm{km}$. Hence, the simulated jet would take $\sim 15$–$20\,\mathrm{s}$ to appear 3 pixels from the Hα kernel location in the observations (or later if travelling at some angle out of the plane of observation). So the simulation predicts that we should only expect to see the displacement of the jet after this time, in agreement with observations (Fig. 2b, blue arrow).

The simulated light curves in the two channels were normalized to unity at their peak values (Fig. 7b) and subtracted, in order to analyse the excess of the AIA171 Å enhancement relative to that in AIA 94 Å. This excess is plotted as a fraction of the enhancement in the AIA 171 Å channel in Fig. 7c.

The enhancement in AIA 171 Å peaks during the beam injection phase (0–10 s) and decreases afterwards (Fig. 7a). After 50 s the response in this channel has returned to background level (Fig. 7a). Because the jet is observed away from the Hα kernel location after 20 s and the AIA cadence is 12 s, the jet should only be visible in AIA 171 Å for 1-2 time frames according to the F10 model (as indicated by the FWHM line plotted in Fig. 7b), which is indeed the case (Fig. 2b).

Figure 7c and shows that at 30 s our model predicts a much greater enhancement over the AIA 171 Å background than over the AIA 94 Å background. Therefore, the F10 model predicts the presence of a jet, outflowing from the chromospheric source of beam heating, that is visible in AIA 171 Å and not AIA 94 Å at around 30 s. The fact that a jet is not observed in AIA 94 Å but only in AIA 171 Å (Fig. 2b), adds further evidence to support the value of the flux used in the F10 model, because it places an upper boundary on the temperature, T, of the outflow, ie, T is much $< 10\,\mathrm{MK}$, and limited to 1–2 MK. At the same time, the jet is not observed in AIA 304 Å (Fig. 2b) resulting in similar implications for the lower temperature. One can conclude that the jet must be also much hotter than 100 000 K (the sensitivity peak for the AIA 304 Å channel). This is why this jet is clearly observed in the AIA 171 Å channel.

**Figure 7 | Simulated and observed AIA light curves.** (**a**) The simulated light curves in the AIA 94 Å (green line) and 171 Å (yellow line) channels for contributions from the flaring corona, transition region and chromosphere. The simulation does not include background from the overlying upper corona or neighbouring corona. Observed values for the 94 Å (green crosses) and 171 Å (yellow crosses) channels including this background are shown. (**b**) The simulated profiles of the signals in AIA 94 Å (green line) and 171 Å (yellow line) above background. These profiles have been normalized to 1 at their peak values. The AIA 171 Å channel is particularly bright compared to the AIA 94 Å channel at around 30 s. (**c**) The normalized fractional excess in AIA 171 Å. The normalized light curves in panel b were subtracted to find the relative excess in the 171 Å channel. This excess is plotted as a fraction of the emission in the 171 Å channel at each instant. The full width half maximum (Horizontal black bar) indicates the times at which the jet is particularly bright in AIA 171 Å compared to AIA 94 Å, the vertical bar represents the maximum relative brightness at around 30 s.

**Radiative transfer method.** For the interpretation of hydrogen emission in flaring atmospheres defined by the hydrodynamic models discussed above we use the second part of HYDRO2GEN code utilizing a full non-LTE approach for a five level plus continuum hydrogen model atom. Both radiative and collisional mechanisms of hydrogen activation and deactivation are considered for thermal and beam electrons, external and internal diffusive radiation, and three-body recombinations.

Here we consider hydrogen excitation and ionization by beam electrons in addition to thermal ones, because only the non-thermal rates can explain the rapid (one second or so) increase of Hα line emission in the core and wings. Thermal electrons, even with higher temperatures, cannot account for timing and shape of the observed Hα line emission.

For non-thermal hydrogen excitation and ionization rates by beam electrons the analytical formulae by Zharkova and Kobylinskii[49] were used, with the beam electron densities calculated from the continuity equation[27,60]. Stimulated photo-excitation, de-excitation and ionization rates by external sources were taken from Zharkova and Kobylinskii[61].

Steady state equations are considered for all the transitions in a 5 level plus continuum hydrogen atom model. For the lines and Lyman continuum, which are optically thick, the radiative transfer equations are solved in the integral form, as follows:

$$S(\tau) = \frac{\lambda}{2} \int_0^{\tau_0} K_1(|\tau - t|)S(t)dt + S^*(\tau), \qquad (8)$$

where $S$ is the source function for the line or continuum, $\tau$ is an optical depth in the line centre or the continuum head, $\lambda$ is the survival probability of a scattered photon, and $S^*$ is the initial source function, calculated without diffusive radiation. The integral was calculated over all the optical depths up to its maximum value for each different wavelength, $\tau_0$.

The kernel functions, $K_1$, in the lines and Lyman continuum are given by the following equations:

$$K_1(|\tau|) = F(T) \int_{\nu_{1c}}^{\infty} f_1 \nu^2 \exp\left[-\frac{h(\nu - \nu_{1c})}{kT}\right] E_1(|\tau| f_1)d\nu, \quad \text{Lyman continuum}$$

$$K_1(|\tau|) = A \int_{-\infty}^{\infty} (\alpha(x))^2 E_1(|\tau| \alpha(x))dx. \qquad \text{Lines}$$

$$(9)$$

$F(T)$ is a normalization coefficient of the kernel functions for the Lyman continuum, and $A$ is the one for the lines. $T$ is the kinetic temperature of the plasma, $\nu$ is the frequency of the radiation and $\nu_{1c}$ the frequency in the Lyman continuum head. $E_1(x)$ is the exponential integral of the first kind. The absorption profiles in the lines, $\alpha(x)$, were taken in the form of a Voigt function, where $x$ is a dimensionless wavelength measured in Doppler half widths from the line central wavelength. The effective Doppler half widths of spectral lines, $\Delta\nu_D$, defined by thermal motion of hydrogen atoms are calculated for the temperature profiles from the hydrodynamic model for every instant of beam injection by considering the contribution of the relevant Doppler widths from each layer weighed by the layer thickness. Turbulent velocities were included in the calculation of the Doppler half widths[62], and were around $1 \text{ km s}^{-1}$ in the flaring chromosphere. At all depths this produced only a very small additional contribution to the Doppler half widths due to thermal velocities. For the Balmer lines Stark's effect, due to strong electric fields caused by ionization of the ambient plasma, was also considered.

The absorption profile in the Lyman continuum, $f_1$, is defined as follows:

$$f_1 = \left(\frac{\nu_{1c}}{\nu}\right)^3. \qquad (10)$$

The radiative transfer and statistical equilibrium equations were solved together to define source functions in each atomic transition and ionization degree of hydrogen atoms in the atmosphere at any given instant of a hydrodynamic response. The solutions of the radiative transfer equation (8) were found using the L2 approximation described by Ivanov and Serbin[63]. The system of integro-algebraic, non-linear equations was solved iteratively until all the source functions are converged with the relative accuracy of $10^{-5}$. Then the simulated Hα line intensities are calculated from the relevant source function using Voigt's absorption profiles. For the ease of comparison with observations, the simulated profiles are normalized to the same units as the observed intensity.

**Data availability.** The data sets generated during and/or analysed during the current study are available in figshare with the identifier 10.6084/m9.figshare.4907285.

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

## Acknowledgements

M.D. and V.Z. wish to acknowledge the PhD studentship support from Northumbria University via the University Research Development Fund at Research Councils UK (RCUK). The authors acknowledge IDL support provided by STFC. We are most grateful to the staff of the SST for their invaluable support with the observations. The Swedish 1-m Solar Telescope is operated on the island of La Palma by the Institute for Solar Physics at Stockholm University in the Spanish Observatorio del Roque de los Muchachos of the Instituto de Astrofísica de Canarias.

## Author contributions

M.D.: Calculations of NLTE model and theoretical line profiles and light curves, reduction of observational profiles, comparison of the profiles with simulations, producing figures, writing the paper and its discussion. E.S.: Interpretation and analysis of observations from SDO and SST, editing and discussion of the paper. V.Z.: Formulation of the NLTE and hydrodynamic problems, hydrodynamic and NLTE simulations, writing the paper, editing and discussion. S.M.: Processing of RHESSI data, building the energy spectra for this event and beam parameters, paper editing and discussion. S.Z.: Reduction of HMI magnetic data, producing plots for Figure 1, investigation of HMI helioseismic data for this event, hydrodynamic simulations, paper editing and discussion. L.R.V.d.V.: Observer and provided observational data reduction for SST.

## Additional information

**Competing interests:** The authors declare no competing financial interests.

