## [Peer Review File · Nature Communications]

Reviewer #1 (Remarks to the Author):

This is a very interesting work that reports on high-resolution multiband observations of a solar flare, in particular in the H-alpha line, and explain them in detail with specific hydrodynamic and radiation modeling. The work shows that the modeling is able to explain the detailed line shapes at very high time resolution, and in particular those at early times, with the impact of high speed electron beams onto the thick chromosphere (hammer effect), as never done before.

The modeling and interpretation looks very strong and solid on the side of visible band and chromospheric description, which is also the central goal in this work. However, the work involves also the higher energy bands and the coronal part of the flare and, in this case, the analysis, modeling and interpretation does not look at the same level. One important issue is that in their modeling the density does not appear to increase in the corona as a consequence of the upflows (chromospheric evaporation). This looks inconsistent with the emission peak in the high energy bands (e.g. Fig.1), and with the results of several other models that include electron beam heating, even at moderate beam fluxes, e.g. Fisher et al. (1985), Peres et al. (1987), Allred et al. (2005). In order to obtain a work with a uniform quality, the authors are requested to take care of this issue. More detailed questions are listed in the following.

Sec.2

- Fig.1: two light curves are shown (the caption counts one). What is the gray horizontal line?

- Fig.2: the velocity of 93 km/s is affected by an uncertainty that is probably not small. The authors should quantify and discuss this error.

- Fig.3: please, explain the normalization and provide a typical intensity.

Sec.3.1

- The authors should provide some basic information about the parameters of the 1D hydrodynamic model, initial and boundary conditions, geometry, resolution.

- The coronal response looks puzzling and inconsistent with the light curves in Fig.1a. Upflows from the chromosphere involve that the density increases in the corona, but this does not seem the case in Fig. 4c, where instead it looks like the density decreases for small column depth. 1D hydrodynamic simulations with similar beam parameters typically obtain much more significant evaporation (see above). The authors should clarify.

- "This is too high to be detected in the 17.1 nm AIA passband, but too cool to be detected in the 9.4 nm passband." 1 MK plasma is easily and customarily detected in the 17.1 nm AIA band. The 9.4 nm band has also a secondary sensitivity peak just below 1 MK and may detect such plasma as well. However, the 9.4 nm band is typically much less sensitive than the 17.1 nm one. The authors are requested to be very careful when they interpret the data without a detailed model, and modify the text accordingly.

Sec.3.2.1

- "Thermal Doppler broadening, turbulent motion Doppler broadening and collisional broadening from free electrons are included." Please provide some details about each of the effects, e.g. typical values, reference for turbulent motion, etc.

Sec.3.2.2

- Fig.5 is very important and the authors should describe more explicitly and at once what is the physical origin of the line shapes and of the two peaks (Fig.5a, left) and make a more explicit and detailed correspondence with Fig.4.

- "this is compared to a one order of magnitude change in intensity in the observations." Where can we read this change in Fig.5b? What is the normalization?

Sec.4

- The authors claim that the AIA upflow "fits the hydrodynamic simulations remarkably well". This might be true for the speed, but it does not seem so for the density, as mentioned above. The authors should patch and comment this issue.

- 20 sec is really a very short time for cooling, so short that it looks unphysical. The authors should check and justify this time with estimates from basic physical cooling mechanisms, i.e. conduction and/or radiation cooling.

Reviewer #2 (Remarks to the Author):

This study represents an excellent example how cutting edge observed data combined with state-of-the-art simulations advance our understanding of physical processes on the Sun. The manuscript represents first observations of the pre-flare shock. Shocks observed after the flare onset have been observed for long time, however, pre-flare shocks were never observed. Nevertheless, they have been predicted by hydrodynamic models since 80-s. The problem was that these shocks are short lived, and they manifest themselves as red-wing intensity enhancement combined with a red shift. In moderate to strong flares the intensity enhancements may be so strong that the pre-flare red-wing activation may not be readily detectable. That is where high spatial, spectral and temporal resolution observations are vital for these kind of study.

Earlier observations by Wulser and Marti (1989) did detect red-shifted profiles but their interpretation of the data were limited to qualitative comparison with the then available simulations (e.g., Canfield, Gunkler and Ricchiazzi 1984; Canfield and Gayley 1987), which were not able to reproduce the observed intensity distribution in the H-alpha wings.

Here authors consider radiative transfer in a single plane flaring atmosphere along with non-thermal excitation and ionisation and were able to successfully match the observed line profile. This proves that the used hydrodynamic models correctly reproduces macro-velocities and the NLTE model correctly calculates the distribution of hydrogen excitation and ionisation levels as well as the emergent spectral line and continuum emission.

The confirmation of the hammer effect on chromospheric plasma and validation of the models discussed here is significant on several levels. First, the model can be instrumental for understanding such active phenomena as white light flares, IRIS explosive events and Ellerman bombs. Research on these events has gained importance lately, so the study under discussion is both timely and valuable. Although there is extensive observational data on these highly energetic events, we are still not able to accurately describe and pinpoint the emission mechanism. How does the continuum enhancement occurs in solar flares? Several different approaches are discussed but so far the conclusions are not definite. It seems that the present approach could be tremendously helpful provided that more data-model comparison will be made.

I therefore recommend this study for publication in Nature Communication after this paper undergo major revision. Here are my comments:

Major

1) I was really having problems to connect the text in the 2 and 3 paragraphs of Section 3.1 (starting with "Fig. 4 shows ...", page 7) with the Figure 4. The main problem is that the text utilizes terms "photosphere", "chromosphere", and "corona" while Fig 4 plots data in terms of "plasma depth". While for some it is straightforward to connect the terms and the plasma depth values, for majority it will be impossible to do so. For example, authors say "The temperature increase in the corona is positively correlated to the initial energy flux of the electron beam as a result of the scale and depth of energy deposition by the beam in the upper atmosphere." Its is hard to see that in the plot. I strongly recommend that authors work on that paragraph and properly describe Fig 4, since it presents one of the key results of the paper.

2) I feel that authors did not sufficiently emphasised the novelty and uniqueness of their results as well as the difference between their simulations and those that people have attempted before. These should be added to Abstract and Discussion sections.

Minor

1) The text does not read easy and is filled with technical terms and concepts that can be easily understood by few specialists. Nevertheless, the Nature Comm targets a broad scientific audience and I'd suggest that at least some parts (e.g., Abstract, Introduction and Discussion) are rewritten to adopt the results for a layperson.

2) I also note that the text and figures preparation seem a bit rushed. For example, right panel in Figure 1a has two x-axes plotted. Since these two are of different range, there are too many tick marks. Needs to be corrected. Other examples are listed below. The list is by no means complete.

3) Description of observations/events should be in past tense

4) Should be either Angstroms or nm. Not both.

5) I think that panel a in Figure 3 should precede the panel b in Figure 2. Otherwise, it is difficult to place Fig 2b into the context. Alternatively, plot a box outlining AIA FOV in the left panel of Fig 1a (instead of white arrow)

6) In Section 3.2.2 on page 9, authors say “The horizontal extent of the red boxes in the panels of Fig. 5 ...”. There are not red boxes in Fig 5.

7) Speaking of Figure 5, I do not understand the purpose of the green boxes. Why not to plot the simulated data using the same spectral range as that in observations? That would be much easier to compare. Or even better, overplot these two curves.

8) There are *numerous* problems with English grammar and (equally importantly!) Composition, which makes at times difficult to comprehend the text. (interpolating spectral point -> interpolated; appears in the location -> appeared at the location; data is -> data are, to name a few).

Reviewer #3 (Remarks to the Author):

The authors have presented observations with the Swedish Solar Telescope, in conjunction with RHESSI and AIA. The high-resolution CRISP images are shown to overlay the X-ray emission, with a jet forming near the footpoints. They focus on H-alpha, which they argue can be used as a diagnostic of chromospheric condensation in solar flares. They have synthesised H-alpha emission from a hydrodynamic simulation in support of their argument, showing that the general features are reproduced.

The paper has merit as a combination of observation and simulation, and would be well received in the community. However, there are currently a number of problems that must be resolved before I can recommend publication. I hope the authors take my comments into consideration, as I think they would greatly strengthen the results.

In the abstract, the authors write “We present the first observations with supporting simulations confirming this process.” This statement is untrue: there have been many, many studies with both observations and simulations of evaporation and condensation in flares. Recently, for example, Kennedy et al. 2015, A&A, 578, 72; Polito et al. 2016, ApJ, 816, 89; Rubio da Costa et al 2016, 827, 38.

The RHESSI data analysis is unclear, and so somewhat difficult to interpret. A few points should be clarified:

1. Which detectors were used to make the spectrum and images?
2. Which specific functions in OSPEX were used to fit the data?
3. The spectrum presented in Figure 1a looks like it could be well fit with only a thermal component. Did the authors check to see whether that was the case? The residuals of the fit would clarify, and should be included in the figure.
4. The authors state that the energy flux was 10^{10} erg/s/cm², but give no indication as to how this was determined, nor what cross-sectional area was measured (or assumed) in order to determine the flux. The results of hydrodynamic simulations depend strongly on the beam flux, so it is important to explain how the authors arrived at this number.
5. How long were the integration times used to make the RHESSI images? Which level contours are displayed?

Regarding Figure 4, the initial (t=0) values should be displayed on the plots. The authors mention in the introduction that the assumed initial atmosphere was for the quiet sun. Does this mean that VAL C or FAL C was taken as the temperature profile? Or something else? The results of the simulation depend strongly on the assumed profile, so this should be described.

Does the temperature in Fig. 4a refer to the electron or hydrogen temperature? What about the density in 4c?

I have a few queries regarding the hydrodynamic simulation.

1. If the heating lasts only 10 seconds, how is it that the flows are sustained for more than 100 seconds (Figure 4b)? The over-pressure that drives evaporation should begin to dissipate as soon as the heating ceases, as radiation and advection carry energy away.
2. The authors should comment on the assumed heating duration. 10 seconds is significantly shorter than the duration of the HXR burst.

3. Zharkova & Zharkov 2007 remark that they used coronal abundances in the radiative loss function (Equation 9 in that paper). This is incorrect for the chromosphere simulated here, and could affect the resultant evolution of the atmosphere. The authors should make sure to use photospheric abundances in this simulation.

In Figure 5, it is difficult to compare the simulations to observations since the latter plot has been normalised. It would be more helpful to see the plot in the native units of the detector, as the derived intensities are an important feature that should be reproduced by a faithful simulation.

In the fourth paragraph of the discussion section, the authors write that the discrepancy in the flows was either due to the energy flux being higher, or injected for a longer time. However, the authors have not ruled out other causes: a different low-energy cutoff or spectral index, both of which affect the depth of energy deposition; the initial temperature/density profile was significantly different to that assumed; that there may have been a different heating mechanism, such as conduction or waves; or that the line profile cannot be reproduced from a single thread, but requires a multithreaded model (compare e.g. Rubio da Costa et al 2016, ApJ, 827, 38).

Minor comments:

The introduction has neglected recent developments on the synthesis of H-alpha from radiative hydrodynamic simulations, which have made significant strides over the work done during the 80s referenced in the intro. For example, the recent work by Rubio da Costa et al 2015, ApJ, 804, 56; Rubio da Costa et al 2016, ApJ, 827, 38; Kowalski et al. 2016, ApJ, in press (arXiv: 1609.07390). The H-alpha profiles of the last one, in particular, seem in close agreement to the observations presented in this paper.

In the last paragraph of Section 3.1, the limit to up-flow speeds was derived by Fisher et al 1984, ApJL, 281, 79, and can readily exceed 1000 km/s, confirmed by many later papers. Whether the flows are explosive or not is not solely a function of energy flux, either: see Reep et al. 2015, ApJ, 808, 177.

In the second to last paragraph, there are many previous studies that the authors should not overlook, studying both hot upflows and chromospheric downflows: e.g. Del Zanna 2008, A&A, 481, 49; Milligan & Dennis 2009, ApJ, 699, 968; Graham & Cauzzi 2015, ApJL, 807, 22; Li et al 2015, ApJ, 813, 59; Tian et al 2015, ApJ, 811, 139.

Reply
to the reviewers' comments
by Druett et al.

Reviewer #1

We appreciate the reviewer's largely supportive comments very much. We revised the paper with respect to all the comments and explained the corrections, as outlined below. We hope the referee will support the revised paper for publication in Nature Communications.

C1. This is a very interesting work that reports on high-resolution multiband observations of a solar flare, in particular in the H-alpha line, and explain them in detail with specific hydrodynamic and radiation modeling. The work shows that the modeling is able to explain the detailed line shapes at very high time resolution, and in particular those at early times, with the impact of high speed electron beams onto the thick chromosphere (hammer effect), as never done before.

A1. We appreciate the appraisal of our paper by the reviewer, many thanks.

C2. The modeling and interpretation looks very strong and solid on the side of visible band and chromospheric description, which is also the central goal in this work. However, the work involves also the higher energy bands and the coronal part of the flare and, in this case, the analysis, modeling and interpretation does not look at the same level. One important issue is that in their modeling the density does not appear to increase in the corona as a consequence of the upflows (chromospheric evaporation). This looks inconsistent with the emission peak in the high energy bands (e.g. Fig.1), and with the results of several other models that include electron beam heating, even at moderate beam fluxes, e.g. Fisher et al. (1985), Peres et al. (1987), Allred et al. (2005). In order to obtain a work with a uniform quality, the authors are requested to take care of this issue.

A2. We appreciate the points raised by the reviewer. We have identified a number of responses below, which should satisfy the referee's comment. In short, during the very early phase of hydrodynamic response to electron beam heating (<100s) that we simulate in the current paper starting from the quiet sun (QS) chromosphere, it is normal to expect the QS chromospheric density to decrease to coronal density, to form a flaring corona. Much later (3-5 minutes after the beam is off) the density in flaring corona will become increasing due to continuing evaporation from the flaring chromosphere. This can last up to an hour or so, until the flare is over and the QS chromospheric density is returned to the pre-flaring conditions. This is expected in the hydrodynamic model as outlined in detail below.

The main points raised by the reviewer are answered as follows:

- a) The hydrodynamic (HD) model of Zharkova and Zharkov (2007) (or Somov et al., 1981, Solar Phys., 73, 145) uses **the quiet sun chromosphere** (from the column depth of 10^{17} cm^{-2} to 10^{22} cm^{-2}) as **the initial condition**. Further initial condition is a constant temperature of 6700K and exponential hydrostatic density distribution indicated by the straight line in our Fig. 4. This QS chromosphere is converted by a HD response to beam injection into a flaring atmosphere (with its own corona, transition region (just above $2 \times 10^{19} \text{ cm}^{-2}$) and chromosphere (below it)). While the other models (Fisher et al., 1985, ApJ, 289, 414 or Allred et al. 2005, ApJ, 630, 573) use the pre-heated flaring atmosphere (the well-known VAL F model derived semi-empirically for the chromosphere) and 'attach to it' (as stated in their own words by Fisher et al.) the QS corona.

In general, in many flares there is pre-flare heating and, thus, the models by Fisher et al or Allred et al are more applicable, than the one we use. In the case of the current flare there was no pre-heating observed in the H-alpha ribbons of event 2. Thus, our HD model is more applicable for the interpretation of coronal and H-alpha emission in this flaring atmosphere created by beam injection.

- b) The initial atmosphere used by Fisher et al requires them to amend their **heating function** (Emslie, 1978; Nagai and Emslie, 1984) by truncating it from the top (to keep the corona from overheating). Also they need to truncate this heating function from the bottom (in the upper chromosphere) to avoid a stopping depth of electrons with the lower cutoff energy where the heating function becomes infinity because of use of flux conservation approach for beam density (Mauas and Gomez, 1997, ApJ, 483, 496; Kontar et al, 2011, SSRv, 159, 301, page 6, footnote). Our heating function does not have this deficiency as it uses continuity equation (Syrovatskii and Shmeleva, 1972) for beam kinetics versus flux conservation equation used by Nagai and Emslie (1984).

- c) **Coronal density and evaporation rates.** As explained above, the precipitating beam generates the corona of a flaring atmosphere from the QS chromosphere. This naturally causes a decrease of the density in upper flaring atmosphere (corona) reported by us in Fig.4 because this is a part of the process of converting this QS chromosphere into the corona of a flare. This process lasts for about 100-300 seconds depending on the beam parameters (Somov et al, 1981) eventually forming a corona with the density of about 10^9 cm^{-3} . This density in the corona of a flaring atmosphere in our (ZZ07 and S81) model coincides with the density of the corona used in the papers by Fisher et al, 1985, or Allred et al, 2005.

During this time while the corona is establishing, the plasma from lower levels evaporates upwards, and this evaporation occurs simultaneously with sweeping. It lasts for thousand seconds (or longer) increasing coronal densities in the flaring atmosphere, exactly as the referee suggested.

From this moment, the hydrodynamic models used by Somov et al (1981) (and Zharkova and Zharkov, 2007) are consistent, by the own admission of the authors, with that of Fisher et al, 1985 and later models by Allred et al, 2005. This evaporation will certainly be increasing the coronal density that coincides with the AIA observations reported in our paper Fig.2.

Although, Fisher et al., 1985, ApJ, 289, 414, Peres et al., Allred et al, 2005 and the other papers on hydrodynamic simulations reported (Somov et al, 1981, Nagai and Emslie, 1984) did not present any increase of the coronal density in the first 50 seconds of their times (which by definition must be later times (>100s) compared to our model as discussed above. We checked all the papers (see for example Figs. 2-6 in Fisher et al, 1985a, 289, 414) or Allred et al, 2005, Figs, (Figs.3 and 4 where a small density increase seen after >75 s).

Moreover, the observations of a density increase in flares were only reported 3-10 minutes after a flare onset (Duijveman et al, 1983) that is much later than 100 s considered here. Similarly, the time delays between the peak flare temperature and peak emission measure are well documented from observations (notably, *Raftery et al., 2009, A&A 494, 1127–1136*) reporting 5-6 mins delay until the density peak after temperature peak; or *Ryan et al., 2014; Scullion et al., 2016, ApJ, in press, arXiv:1610.09255* reporting ~3 mins delay until the density peak for a C-class flare after the temperature peak). However, these time delays of a hundred or thousand seconds are not a part of the current model, which considers only the first 100 seconds after the flare onset.

- d) The two peaks in SXR emission in Fig.1 are likely related to the occurrence of three separate flaring events 1-3 in HXR (and SXR emission) for this flare (as explained in Fig. 1 and section 2.1) occurring in sequence at different times in different parts of the loop system of the flare. This means the flare starts from the event 1 (missed by H-alpha observations), which coincides with the original SXR curve increase, then continues with the event 2 (considered in this paper), which is close to the first maximum in the SXR GOES curve and then extends to the event 3, which is accountable for the second maximum in the GOES curve. Then the SXR emission naturally decreases owing to radiative cooling, thermo-conductivity and plasma motion (Moore and Datlaw, 1975; Antiochos, 1982).

Corrections in the revised paper:

- a) Added in section 1 a description of three types of hydrodynamic models currently in use by researchers. We also added the Methods section 5.2

where we placed a clear description of the current HD model and the initial conditions used: the QS chromosphere and its conversion into the corona of a flaring atmosphere with a discussion of simulated temperature, density and macrovelocity profile versus other HD models and their fit to the AIA observations. The initial exponential hydrostatic density is now shown by a solid line on Fig. 4.

- b) Explained in section 1 and 5.2 the differences in the heating functions used by our HD model versus the models by Fisher et al, 1985 and Allred et al., 2005.
- c) Discussed in sections 1 and 5.2 the general variations of density, temperature and macrovelocities simulated in the HD model by ZZ07 and decrease of temperature and density during the first 100 s. We also mention that after 3-5 minutes the density in the flaring corona starts to increase, and will do so until it returns back to the pre-flare QS chromosphere.
- d) Discussed in section 2.1 the three events observed in HXR for this flare, which contributed the SXR GOES light curve. We also briefly discussed possible options, which can extend the SXR light curve for longer times beyond the primary heating by electron beams: repeated injection of beams (Duijveman et al, 1983); secondary acceleration of ambient electrons (Petrosian et al, 2012); or strong non-thermal ionisation of the coronal ions by beam electrons (Porquet et al., 2001, A&A, Kawate et al, 2016, ApJ).

C3. Sec.2

- Fig.1: two light curves are shown (the caption counts one). What is the gray horizontal line?

A3. This is a valid comment, thanks very much. We have added the information on both GOES wavelength bands to the text of section 2.1 describing figure 1, and the explanation of the horizontal grey line, which indicates the GOES flare classification of the event, namely a C1.5 class flare.

C4. - Fig.2: the velocity of 93 km/s is affected by an uncertainty that is probably not small. The authors should quantify and discuss this error.

A4. The measurement of velocities in the coronal jet is derived from the apparent motion of the jet within the AIA image sequence in the 171 Å channel. The error in the measurement is sensitive to the pixel size of 0.6 arcsec and we can expect an error of +/-30 km/s in the time frame within which we measure the jet protrusion. This error estimation is accounted for already within Fig. 4 where we present box #1 within the macrovelocity light-curve within a range of 117-57 km/s centred at 93km/s. This velocity range remains acceptable given the model predictions for that temperature band indicated in Fig. 4 (box1), confirming

Reply to the referee by Druett et al.

other observations of the upflows of 100 km/s for the beams with the similar initial energy flux (Milligan et al., 2005; Milligan et al., 2006, ApJ, 642:L169-L171).

In the revised paper the point about the error bars is clarified in section 3.1.2 and provided references to the papers above reporting the similar velocities for the flares with the similar parameters.

C5. - *Fig.3: please, explain the normalization and provide a typical intensity.*

A5. This is a valid comment, many thanks. We have expanded our explanation of the normalisation in Methods section 5.1, which now include intensity in the data number units. We have also described the normalisation process in the caption of Fig. 3.

C6. *Sec.3.1*

- *The authors should provide some basic information about the parameters of the 1D hydrodynamic model, initial and boundary conditions, geometry, resolution.*

A6. We appreciate this comment.

In the revised paper the initial conditions and heating functions are now described in the paper sections 1 and 5.2 (Methods) as we explained in the answer A2. These initial conditions are similar to the approach by Somov et al, 1981, Sermulinya et al, 1982, Duijveman et al, 1983 and Nagai and Emslie, 1984, who also used a hydrostatic atmosphere as the initial condition.

C7. - *The coronal response looks puzzling and inconsistent with the light curves in Fig.1a.*

Upflows from the chromosphere involve that the density increases in the corona, but this does not seem the case in Fig. 4c, where instead it looks like the density decreases for small column depth. 1D hydrodynamic simulations with similar beam parameters typically obtain much more significant evaporation (see above). The authors should clarify.

A7. We understand that we needed to explain better the key points of the HD model used. Now it is done as per A2 and A6. We did not do it in the original version because the results of hydrodynamic simulations were already published 9 (Zharkova and Zharkov, 2007) or 35 years ago (Somov et al., 1981).

In the revised paper we changed accordingly sections 1 and 5.2 as per answer A2, to explain the differences between the HD models currently in use and the key points of the model we used here to explain the observations presented in this paper. What has been presented here was mainly dedicated to the interpretation of H-alpha profile using NLTE radiative model.

We would like to refer the referee specifically to the item c in the answer A2 about the coronal density variations originating from conversion of the QS chromosphere into a flaring corona, transition region, and chromosphere. This is caused by the injection of an electron beam, which sweeps the QS plasma, thus, decreasing the QS chromospheric density to form the corona of a flaring atmosphere (see for more details answer A2, item a). This process happens simultaneously with evaporation of the swept plasma back to the newly formed corona from chromospheric levels of a flaring atmosphere (see Fig. 4, b and c). This sweeping plus evaporation process lasts from 100 s (for weaker beams, considered here) to 300s (for more intense beams, considered by Somov et al., 1981 (see their Fig.3 and 4).

After this time the flaring atmosphere is formed and its coronal density has exactly of the same magnitudes (10^9 - 10^8 cm⁻³) as accepted in the other HD models (Fisher et al, 1985 Allred et al, 2005. The rates of evaporation, or macrovelocities, derived by S81 (approaching 1600 km/s for intense beam) or ZZ07 (a few hundred km/s for a weaker beam) are close (and even higher) to those reported by Fisher et al, 1985 (1000 km/s) as noted in the introduction by Fisher et al, 1985. Therefore, the evaporation rates in flaring atmospheres (after they become the ones) reported by us (and Somov et al, 1981) are in agreement with the other models.

The AIA observations (plotted in Fig.2) at the earliest stage of the flaring event 2 considered in this paper fully support our HD model predictions for the event from ZZ07 (plotted in Fig.4, box 1), in terms of temperature and macrovelocities. This implies that the hydrodynamic simulations of density for these 100 s are also accurate for this event, because only a decrease of the density from a chromospheric to the coronal one to ensure the plasma DEM reaches the observed magnitudes, which fit the AIA window for 171 Å.

Therefore, we have shown that our HD model explains rather well how the flaring atmosphere (with its own corona, transition region and chromosphere) is created from the QS chromosphere. In addition, the same model (S81) shows at the later stage the density in a flaring corona becomes increased by chromospheric evaporation, as the referee suggests, (similar to all other models and observations) when this flaring corona is trying to return back to the density of the QS chromosphere. This density increase in the flaring corona is indeed observed 3-10 minutes after a flare onset as reported in A2 item c.

In the revised version of the paper we have added a discussion of this point (coronal response in the first 100 seconds) explaining that there is no increase of coronal density at the early tens of seconds because it is being formed from the QS chromosphere, thus requiring a decrease of the chromospheric density to the coronal one, in order to form a flaring corona. The increase of density in this flaring corona caused by chromospheric evaporation is expected at later stages (3-10 min after the beam onset), when this flaring corona is trying to return back to the density of the QS chromosphere.

C8. - *"This is too high to be detected in the 17.1 nm AIA passband, but too cool to be detected in the 9.4 nm passband." 1 MK plasma is easily and customarily detected in the 17.1 nm AIA band. The 9.4 nm band has also a secondary sensitivity peak just below 1 MK and may detect such plasma as well. However, the 9.4 nm band is typically much less sensitive than the 17.1 nm one. The authors are requested to be very careful when they interpret the data without a detailed model, and modify the text accordingly.*

A8. Indeed, the 9.4 nm channel is generally more responsive to flaring conditions and, indeed, it is expected that there is secondary sensitivity just below 1 MK. From Fig. 2 (top row), we cannot rule out that there is some (if minimal) signal in the protrusion (blue arrow feature) that is most clear in 17.1 nm images and, therefore, that may be some signature of the secondary peak in 9.4 nm which is much weaker. However, when we examine the temperature profile evolution from the model in Fig. 4, within box1 between 5 – 100 s, it is expected that the plasma will be detectable in the range of $\text{Log } T=5.2$ to $\text{Log } T=6.05$ at the depths of the low corona, not specifically at the peak temperature sensitivity of the 17.1 nm channel. The 17.1nm channel is the most sensitive to this range in comparison with any other available AIA channel and we are, therefore, inclined to highlight this correspondence given the limited range of temperatures that AIA is uniquely sensitive to.

In the revised paper we modified the text of section 3.1.2 to account for the limitations in interpreting the temperature sensitivity with AIA such as to not rule out a plausible signal due to the secondary peak in 9.4 nm.

C9. Sec.3.2.1

- *"Thermal Doppler broadening, turbulent motion Doppler broadening and collisional broadening from free electrons are included." Please provide some details about each of the effects, e.g. typical values, reference for turbulent motion, etc.*

A9. We thank the reviewer for this comment. In fact, the thermal collisional broadening (together with natural broadening) is what defines Doppler broadening, or Doppler half widths, which in turn, define the absorption coefficient and the shape of spectral lines that is caused by collisions of hydrogen atoms at a given temperature.

In the revised paper we explained in section 5.3 of Methods how we calculated the effective Doppler broadening for spectral lines weighed by a length of the layer with a given temperature. Also we indicated in the revised version that micro-turbulent velocities (reference is given) are very small compared to the Doppler thermal velocities. We also included the references to our previous paper by Zharkova and Kobylinskij, 1991 where all these values are described. We also explained that we use of Voigt profiles for emission in all lines.

C10. Sec.3.2.2

- Fig.5 is very important and the authors should describe more explicitly and at once what is the physical origin of the line shapes and of the two peaks (Fig.5a, left) and make a more explicit and detailed correspondence with Fig.4.

A10. We appreciate this comment and corrected the description of this figure in the paper as follows.

- a) We have added a reference to the features in Fig. 4b, box 2 in section 3.2 when describing the effect of the hydrodynamic shock on the H-alpha line profile.
- b) We have also included a few sentences in section 5.3 and 3.2 to describe the origin of the "horns" in H-alpha line profile (mainly caused by radiative processes (lower optical thickness for the blue and red wings) and self-absorption for the central line intensity dropping below the horn intensity. We also explained that for the red wing of the line profile calculated for the time of maximum heating by the beam the downward motion of a hydrodynamic shock forms the increased intensity occurring at about 0.8 Å from the line centre. This is caused by macromotion of the whole plasma of the shock with the speed about 40-43 km/s. This is close to the velocity derived from the observations (45-50 km/s).

We have to emphasize that the line profile mainly formed by the radiative transfer effects of hydrogen transitions in H-alpha line and its interaction with optically thick continuum, as described in the Method section 5.3. We note that not all the features in the H-alpha line profile can be described by the hydrodynamic model (Fig 4). Only the big increase of the line intensity in the close red wing comes from the HD model (caused by a downward motion of the HD shock). The time of observation of such the red-shifted line profile and the distance from the central wavelength of H-alpha line where the increase of red wing intensity is measured closely match the simulated profile. Once again, only our radiative hydrodynamic model of type 1 is able to account for this excessive red-shift of the spectral line in the very early phase of the flaring which is the highlight of our results.

In the revised paper we emphasized that the maximum velocity (50 km/s) simulated from the hydrodynamic model for 5s after a beam onset shown in box 1 closely match the velocity (45-50 km/s) measured from H-alpha line profile taken by SST.

C11. - *"this is compared to a one order of magnitude change in intensity in the observations." Where can we read this change in Fig.5b? What is the normalization?*

A11. This is a valid comment, thanks.

In the revised paper we corrected the text to indicate correctly, which boxes are employed for the comparison. Additionally, the normalisation procedure is now described in response A5 to C5 and referenced in the text.

C12. Sec.4

- The authors claim that the AIA upflow "fits the hydrodynamic simulations remarkably well". This might be true for the speed, but it does not seem so for the density, as mentioned above. The authors should patch and comment this issue.

A12. We believe, this referee comment is clarified above in the answers A2 and A7. This point is also covered now the discussion of the density variations in a flaring atmosphere at later times of a gradual phase of the flare.

In the current paper we show that the model by ZZ07 (Somov et al, 1981) presents the correct speed of this chromospheric evaporation measured by AIA/SDO at the given time after the event 2 onset, exactly as reported from a comparison of Fig. 2 and Fig. 4. We also discussed the decrease of the coronal temperature and differential emission measure (DEM) by radiative cooling, thermal conduction and macromotion that is proportional to plasma density (with references to the Moore and Datlowe, 1975; Antiochos, 1982), after the beam is switched off. From these estimates, we deduce that this cooling in our HD model is likely to be caused by a decrease not only a flaring coronal temperature, but also of a flaring coronal density (from the QS chromosphere density to the coronal density) as per answer A2 (items a-c). For example, Antiochos (1982) estimated that for plasma density 10^{10} cm^{-3} the cooling time of flaring plasma from ten million kelvin to normal coronal temperatures (million K), if caused by radiative cooling, thermal conduction and plasma motion, can be about a few hundred up to thousand seconds, while for the density of 10^9 cm^{-3} this cooling time is reduced to a few tens of seconds, exactly as our HD model derived and observations by AIA confirmed.

Our simulations correctly predict the evaporation velocities for a given time after a flaring event onset, which are measured at the exact time that they are predicted by HD simulations and in the exact band of 171 A in AIA, which is expected for a given temperature at that time (shown in Fig. 4 by box 1). We believe, this constitutes a remarkable fit for the whole HD simulation for this event, including the temperature, macro velocity and density.

In the revised paper we added to section 3.1.2 a discussion from A7 of the HD model explaining the observation of the coronal jet with the close to the model evaporation rates by the decrease of both temperature and density on a flaring corona, that reduced its differential emission measure allowing AIA to catch the plasma with this T and n at the exactly moment as predicted by the model.

C13. *- 20 sec is really a very short time for cooling, so short that it looks unphysical. The*

Reply to the referee by Druett et al.

authors should check and justify this time with estimates from basic physical cooling mechanisms, i.e. conduction and/or radiation cooling.

A13. We fully agree with the referee that the time for a total cooling of a flaring atmosphere to the QS chromosphere can last for a few hundred of seconds up to an hour or more (see Shmeleva and Syrovatskii, 1973, Somov et al, 1981, Antiochos, 1982). The hydrodynamic models used by us and by others show that this cooling starts immediately after the beam stopped, and lasts for the whole period of the further simulations to above 1000s (see Fig. 4 in our paper, Figs. 2-6 in Somov et al, 1981 and Figs. 2-6 with the temperature/density variations in Fisher et al, 1985a,b).

However, for interpretation of observations of AIA we do not speak about total cooling of the plasma to the chromospheric temperatures, but only about some intermediate cooling in a flaring corona from tens million kelvin temperature to million kelvin temperature. This cooling, as we indicated in the A12, is caused not only by radiative losses, but mainly by thermal conduction and macromotion (evaporation) (Moore and Datlowe, 1975; Antiochos, 1982). This is exactly what is happening in our HD model and what observations of AIA report.

Somov et al. 1981 (their Fig.6) showed that after beam is stopped, the differential emission measure in the flaring corona quickly (within 10-20 seconds) is decreased by the orders of magnitude because of radiative losses and decrease of the density (see A2 and A12). Hence, the observed DEM integrated over the depths will be also quickly decreased within this time. This decrease of the DEM in this flare reaches the range that matches the AIA window range in 171 A allowing the AIA to observe the jet emission and see its evaporation from the flaring chromosphere into its corona.

Hence, what we captured with AIA at 29 seconds after the event 2 onset is a partial cooling of the corona, which started immediately after the beam heating stopped. It could have continued for another 20-40 minutes (not seconds) after our AIA observations were made. But we did not have H-alpha observations for the times later than 100s, thus, we did not extend the HD simulations for this later stage.

In the revised paper we added a small discussion in section 3.1.2 stating that a flaring corona cooling starts immediately after the beam is off, owing mainly to macromotion (evaporation) and thermal conduction, in addition to radiative cooling, with reference to the papers by Moore and Darlowe, 1975 and Antiochos, 1982 where the flaring cooling times were estimated.

Reviewer #2 (Remarks to the Author):

We appreciate very much the referee comments, revised the paper accordingly and explained the corrections as outlined below. We hope the referee will support the revised paper publication in the Nature Communications.

C1. *This study represents an excellent example how cutting edge observed data combined with state-of-the-art simulations advance our understanding of physical processes on the Sun. The manuscript represents first observations of the pre-flare shock. Shocks observed after the flare onset have been observed for long time, however, pre-flare shocks were never observed. Nevertheless, they have been predicted by hydrodynamic models since 80-s. The problem was that these shocks are short lived, and they manifest themselves as red-wing intensity enhancement combined with a red shift. In moderate to strong flares the intensity enhancements may be so strong that the pre-flare red-wing activation may not be readily detectable. That is where high spatial, spectral and temporal resolution observations are vital for these kind of study.*

Earlier observations by Wulser and Marti (1989) did detect red-shifted profiles but their interpretation of the data were limited to qualitative comparison with the then available simulations (e.g., Canfield, Gunkler and Ricchiazzi 1984; Canfield and Gayley 1987), which were not able to reproduce the observed intensity distribution in the H-alpha wings.

Here authors consider radiative transfer in a single plane flaring atmosphere along with non-thermal excitation and ionisation and were able to successfully match the observed line profile. This proves that the used hydrodynamic models correctly reproduces macro-velocities and the NLTE model correctly calculates the distribution of hydrogen excitation and ionisation levels as well as the emergent spectral line and continuum emission. The confirmation of the hammer effect on chromospheric plasma and validation of the models discussed here is significant on several levels. First, the model can be instrumental for understanding such active phenomena as white light flares, IRIS explosive events and Ellerman bombs. Research on these events has gained importance lately, so the study under discussion is both timely and valuable.

A1. We appreciate very much the high appraisal of our work by the referee and fully agree with the referee statement.

C2. *Although there is extensive observational data on these highly energetic events, we are still not able to accurately describe and pinpoint the emission mechanism. How does the continuum enhancement occurs in solar flares? Several different approaches are discussed but so far the conclusions are not definite. It seems that the present approach could be tremendously helpful provided that more data-model comparison will be made.*

A2. We fully agree with the referee about the understanding of the importance of the current approach.

Reply to the referee by Druett et al.

C3. *I therefore recommend this study for publication in Nature Communication after this paper undergo major revision.*

A3. We appreciate this very much.

In the revised paper we have done major revisions of all the sections and also created the section “Methods”, in order to explain the applied models outside the main body of the paper. The corrections of the paper are outlined in the answers below.

C4. Here are my comments:

Major

C4.1. *I was really having problems to connect the text in the 2 and 3 paragraphs of Section 3.1 (starting with "Fig. 4 shows ...", page 7) with the Figure 4. The main problem is that the text utilizes terms “photosphere”, “chromosphere”, and “corona” while Fig 4 plots data in terms of “plasma depth”. While for some it is straightforward to connect the terms and the plasma depth values, for majority it will be impossible to do so.*

A4.1. We appreciate this comment by the referee.

In the revised paper we added a description to the Figure 4 indicating where the new corona, chromosphere and photosphere are in the flaring atmosphere, formed by the HD response. This description is also added to the Figure 4 caption with the note that these locations of a flaring atmosphere, which are different from those in the quiet Sun.

C4.2. *For example, authors say “The temperature increase in the corona is positively correlated to the initial energy flux of the electron beam as a result of the scale and depth of energy deposition by the beam in the upper atmosphere.” Its is hard to see that in the plot. I strongly recommend that authors work on that paragraph and properly describe Fig 4, since it presents one of the key results of the paper.*

A4.2. We agree with the referee comment about this sentence, it was out of the context presented in this paper.

In the revised paper we introduced the section Methods, where in section 5.2 we described exactly what the assumptions of the HD model presented in this paper are, what its initial and boundary conditions are and what equations are considered. We also provided more references to the papers by Zharkova and Zharkov, 2007 (ZZ07) and Somov et al, 1981 where the other outcomes of simulations are described in terms of T, n and V, in response to the electron beam heating.

C4.3. *I feel that authors did not sufficiently emphasised the novelty and uniqueness of their*

Reply to the referee by Druett et al.

results as well as the difference between their simulations and those that people have attempted before. These should be added to Abstract and Discussion sections.

A4.3. We appreciate very much this comment by the referee, that helped us to have a fresh look at these sections.

In the revised version we completely re-phrased the abstract, all the sections previously used and added the Method sections removing the description of models from the main body of the paper. Also we changed the discussion and conclusions in sections 3 and 4 to reflect the novelty of the approach allowing us to interpret the results of simultaneous observations in the flaring corona and chromosphere.

Minor

C5.1. *The text does not read easy and is filled with technical terms and concepts that can be easily understood by few specialists. Nevertheless, the Nature Comm targets a broad scientific audience and I'd suggest that at least some parts (e.g., Abstract, Introduction and Discussion) are rewritten to adopt the results for a layperson.*

A5.1. We appreciate the point raised by the referee.

In the revised paper we have moved the model description for H-alpha line reduction, hydrodynamics and for radiative transfer to the Method sections 5.1, 5.2 and 5.3 respectively. This allowed us to simplify a description of the results in section 3 as much as possible without compromising the scientific consistency of the paper. Now we believe the main text of the paper can be easily read by a non-expert reader.

We also made a more lucid description of the models in the section 1, the results of hydrodynamic and radiative simulations in section 3 and some comparison with other models.

C5.2. *I also note that the text and figures preparation seem a bit rushed. For example, right panel in Figure 1a has two x-axes plotted. Since these two are of different range, there are too many tick marks. Needs to be corrected. Other examples are listed below. The list is by no means complete.*

A5.2. We appreciate this comment.

In the revised paper we carefully revised all the figures, their captions and the text describing them. Given the fact that these Figures are over-plotted and de-rotated, this was not a trivial task but it is accomplished in the revised paper.

C5.3. *Description of observations/events should be in past tense.*

A5.3. This is done with thanks.

Reply to the referee by Druett et al.

C5.4. *Should be either Angstroms or nm. Not both.*

A5.4. We agree with this comment. In the revised paper we have changed the quoted measurements of wavelengths in Angstroms.

C5.5. *I think that panel a in Figure 3 should precede the panel b in Figure 2. Otherwise, it is difficult to place Fig 2b into the context. Alternatively, plot a box outlining AIA FOV in the left panel of Fig 1a (instead of white arrow).*

A5.5. We have now implemented the 2nd (alternative) suggestion by the referee, i.e. we have plotted a box outlining the AIA FOV in the left panel of Fig. 1a.

C5.6. *In Section 3.2.2 on page 9, authors say “The horizontal extent of the red boxes in the panels of Fig. 5 ...”. There are not red boxes in Fig 5.*

A5.6. We thank the referee for this comment.

In the revised paper we have adjusted the text referring to the green boxes included in the final version, rather than the red boxes used in the previous version of the paper.

C5.7. *Speaking of Figure 5, I do not understand the purpose of the green boxes. Why not to plot the simulated data using the same spectral range as that in observations? That would be much easier to compare. Or even better, overplot these two curves.*

A5.7. The green boxes are designed to show the equivalent regions from the simulation (left) with the observation (right). We decided to plot a broader spectral range in the simulated data (than what is present for the observations) because the emission profile in H-alpha actually extends into the red wing slightly beyond the range of the spectral filter (3A) used in the observation. This component is otherwise unobserved but present in the model. This was used to emphasize the observations of H-alpha emission profiles in flares require a much broader spectral coverage than (1-2 A) than what is typically available from many current ground-based telescopes (such as at the SST, DST and NST). This should serve as a call for the development of Fabry Perot Interferometers that are capable of scanning into the far-wing positions relative to the H-alpha line core. A note about the effect is now added to the discussion of the purpose of the green boxes.

C5.8. *There are *numerous* problems with English grammar and (equally importantly!) Composition, which makes at times difficult to comprehend the text. (interpolating spectral point -> interpolated; appears in the location -> appeared at the location; data is -> data are, to name a few).*

A5.8. We appreciate this comment by the referee.

In the revised paper we worked throughout the text to correct these points and some others. Regarding “data”, in both singular instances we are using it as a non-count noun, and so the singular form is valid. However, we will agree with the language Editor of the journal with the preferred option. Also our 3 native English authors made the careful reading of the text and corrected all the possible small misspellings etc.

Reviewer #3 (Remarks to the Author):

We appreciate very much the referee comments. We revised the paper accordingly and explained the corrections as outlined below. We hope the referee will support the publication of the revised paper in Nature Communications.

***C1.** The authors have presented observations with the Swedish Solar Telescope, in conjunction with RHESSI and AIA. The high-resolution CRISP images are shown to overlay the X-ray emission, with a jet forming near the footpoints. They focus on H-alpha, which they argue can be used as a diagnostic of chromospheric condensation in solar flares. They have synthesised H-alpha emission from a hydrodynamic simulation in support of their argument, showing that the general features are reproduced. The paper has merit as a combination of observation and simulation, and would be well received in the community.*

A1. We appreciate very much the referee appraisal of our paper.

***C2.** However, there are currently a number of problems that must be resolved before I can recommend publication. I hope the authors take my comments into consideration, as I think they would greatly strengthen the results.*

A2. We fully agree with the referee and considered all these comments below, which significantly improved our paper and presentation of the results.

***C3.** In the abstract, the authors write “We present the first observations with supporting simulations confirming this process.” This statement is untrue: there have been many, many studies with both observations and simulations of evaporation and condensation in flares. Recently, for example, Kennedy et al. 2015, A&A, 578, 72; Polito et al. 2016, ApJ, 816, 89; Rubio da Costa et al 2016, 827, 38.*

A3. We agree with the referee comments, the abstract phrase was misleading.

In the revised paper we revised the abstract to reflect the precise points, which are new in this paper – it presents ‘the first close match between observed and simulated of the H-alpha profiles with large red-shifts taken in the very first seconds after a flare onset’. This is a highly significant finding. However, this finding is not in contention with other models of a flare heating, but it rather adds a new crucial component in the overall understanding of the fundamental role of electron beams in the earliest phase of the flare onset and H-alpha ribbon formation. This beam energy is transported during the first tens of seconds to the

Reply to the referee by Druett et al.

lower chromosphere / photosphere in the form of the strong shock combined with a strong evaporation of the chromospheric plasma to the flaring corona. This model prediction combined with radiative model can explain the H-alpha line and AIA observables in this time range. This is what we reveal in this study.

We also added the papers mentioned by the referee to the section 1 and mentioned the results by Rubio Da Costa et al. (2016) in section 3.2 and Kennedy et al in section 3.1.1. Because Polito et al. (2016) consider a much later phase of the flare (>1000s), their results are not directly comparable to ours. For the results by Kennedy et al. (2015), as they did not calculate any H-alpha profiles at all, just a contribution function (Fig.7). Moreover, their downward shocks (Fig. 5) are formed well above the region where H-alpha line is formed from their own contribution function, and much later (50s after beam onset) than the early red shifts reported by us and Wuelser and Marti, 1989, 1990. Therefore, these shocks in the corona could not be seen by the radiative model used for H-alpha in the chromosphere. For this reason we only mentioned their results in section 1 when describing HD models and in section 3.1.1 when presenting our hydrodynamic simulations (see the elaborated text below).

We fully respect and acknowledge the results obtained by the other authors. However, we wish to emphasize that for the first time we managed to reproduce with a single slab (thread) model the entire H-alpha line profiles with large red shifts observed at the very start of the flaring event 2, *i.e with the emission line red peak shifted entirely to +0.8 Angstrom relative to the 6562.8 Angstrom rest wavelength*. This strong red shift occurring very early in the flare formation was observed earlier by *Wuelser and Marti, 1985* and indeed now by the SST. The H-alpha profile is simulated at the maximum flux of the injected beam (5 s after its onset) with a NLTE model including non-thermal ionisation and excitation rates by beam electrons calculated analytically and for the beam densities found from continuity equation in a single slab atmosphere.

Below we elaborate on the points above.

Since 80s, when the first hydrodynamic simulations took place, there were ample simulations and observations of evaporation and condensations in flares. However, they by no means explained all the features reported by observations. Because, as we stated in the section 1, the observations of H-alpha line profiles with strong red shifts at the very first seconds after a flare onset by *Wuelser and Marti (1989, 1991)* and now by SST are still unexplained with the models by *Fisher et al, 1985*, or *Allred et al., 2005* or by any other models.

Hence, it was logical to review some other models available, this what *Zharkova and Zharkov (2007, ZZ07 thereafter)* did. Their approach is similar to that by *Somov et al., 1981 (S81)* and it seems it worked, as we show in the current paper.

This could have a number of explanations including:

a) using different initial conditions to produce the hydrodynamic responses at various times explained in Methods, section 5.2.

- 1) SS07 use the heating starting from the chromosphere of the quiet sun (QS) (column depth of 10^{17} cm^2), just below the QS transition region;
- 2) SS07 use as initial condition hydrostatic density distribution in the chromosphere defined by the exponential law,
- 3) ZZ07 use a constant temperature of 6700K to account for Balmer emission;
- 4) ZZ07 consider no motion before beam onset (hydrostatic atmosphere);
- 5) The other initial and boundary conditions are described by eq. 9-13 in Somov et al.

This means in the model by ZZ07 (as in S81) the injection of electron beam sweeps the QS chromosphere to lower depths and converts it into a flaring atmosphere's corona, chromosphere and photosphere with the TR shifted from the column depth of 10^{17} (the initial atmosphere) to 10^{19} cm^{-2} . This is contrary to Fisher et al, who simply attached (by their own description in the Fisher et al, 1985) the QS corona to the preheated flaring atmosphere of VAL F (semi-empirical model) used as the initial condition by Fisher et al., 1985 or Allred et al, 2005. According to Fisher et al, 1985, the beam injection creates some sweeping with evaporation of the chromospheric plasma to the corona and a low temperature condensation in the upper chromosphere moving as a shock to the photosphere. However, the shock in our model (ZZ07) appears at much deeper chromospheric depths compared to Fisher et al, 1985, which happened to be exactly the depths where H-alpha line core is formed. Also evaporation rates predicted by S81 model are higher than by Fisher et al, 1985 (see A12).

b) Limitations of heating functions used by Fisher et al and Allred et al. The heating functions used by Somov et al and ZZ07 are smooth functions calculated by Syrovatskij and Shmeleva, 1972 solving the continuity equation with a maximum heating at the upper chromosphere. This is contrary to the heating functions of Emslie (1978) and Nagai and Emslie (1984) used by Fisher et al, which are truncated (by own admission by Fisher et al, 1985) from the top in the corona (to reduce the heating in the attached corona) and in the upper chromosphere, to avoid the stopping depth of electrons with lower cutoff energy where the heating function becomes infinity because of the use of the flux conservation approach for beam density (Mauas and Gomez, 1997, ApJ,; Kontar et al, 2011, SSRv, 159, 301, page 6, footnote).

This leads in the models by Fisher et al or Allred et al, to the formation of a low temperature condensation higher in the chromosphere compared to the simulations by

Somov et al, 1981 and ZZ07 presented here, where the shocks are formed in the middle chromosphere.

c) Inclusion of the updated non-thermal ionisation and excitation rates for hydrogen atoms.

In the current paper we use these rates calculated analytically by Zharkova and Kobylinskij, (1993, Sol. Phys., 143) from the hydrogen cross-sections (Johnson, 1972) and beam electron numbers taken from the continuity equation. While others use semi-analytical rates and the beam densities derived from the flux conservation equation (Emslie, 1978, Nagai and Emslie, 1984), which applies significantly reduced electron numbers in the chromosphere, in order to avoid an infinity in the heating rates (Mauas and Gomez, 1997, ApJ)

This is why we revisited the interpretation of hydrogen emission with the updated radiative model of hydrogen emission for the hydrodynamic model by ZZ07.

Now we wish to comment further on the recommended papers, in order to distinguish our results from those mentioned by the referee:

In *Rubio da Costa et al 2016, 827, 38. A3* multi-threads model is assumed, but they still did not get H-alpha profiles with strong red-shifts. While we use a simpler single thread approach which clearly managed to reproduce the line profile features and evolution of the H-alpha line. We also seem to reproduce reasonably well the temperature, density and velocity response for the coronal plasma, TR plasma, and the observation of the coronal jet by AIA.

While in *Rubio da Costa et al 2016, 827, 38. A3*, the H-alpha line profile shows a blue-shifted asymmetry after 52 seconds: the blue shifted peak of the line becomes stronger and the line is shifted to higher wavelengths. We also note that from Fig. 15 in *Rubio da Costa et al 2016, 827, 38. A3* it is clear that the non-equidistant spectral line scan in H-alpha from IBIS means that from 0.5 Angstrom to 1 Angstrom in the red and blue wings the line is significantly under-sampled.

While with SST we have equidistant sampling out to 1.4 Angstroms and we find that this is one of the main reasons why we have detected this signature of the shock front so clearly in our observations. We observe far enough out to the continuum with CRISP and we have sampled sufficiently to recover this signature. A combination of observing H-alpha in a deep scan (33 spectral positions) and catching a flare in good seeing close to disk centre (to avoid projection effects in the observed spectra) is not common with the most powerful telescopes at this high spatial and temporal resolution (~7.25s) in order to catch the shock signature. Indeed, we also capture similar asymmetry effects after 30s in our model from our Fig. 5 (lower curves) in support of the registered asymmetries that *Rubio da Costa et al 2016* and *Kuridze et al., 2015* also model and observe. In *Rubio da Costa et al 2016, 827, 38. A3* from their Fig 11 they show a noticeable red peak due to downflows of almost 15

km/s. However, this is much weaker than the early strong red-shift we detect of ~45-50 km/s in H-alpha, leading to continuum brightening. We mentioned this result in our paper.

Also in regard to *Kuridze et al., 2015* who measured notable red-shifts in the H-alpha emission line formation after the flare onset and resulting red/blue line asymmetries. In our case the predominant red shift signature is detected within the first 5-10 s after the flare onset. For stronger beams this hammer effect, reported in our paper, can shift the entire line profile into the red shift, because of large macrovelocities (up to 100 km/s) of the plasma motion downward to the low chromosphere if the initial beam flux is much higher, as reported by Zharkova and Zharkov, 2015. This can last for a few tens of seconds before the atmosphere thermalizes and the shifted profile returns to the rest wavelength position leading to red-blue asymmetries. Our model presented here does not refute the arguments made in *Rubio da Costa et al 2016, 827, 38. A3* or *Kuridze et al., 2015* but rather supports their results, which appear at a later stage in a flare evolution. Instead, our model actually prepends those results with the critically important phase of this evolution in the earliest times (i.e. in <10s from the start of flare heating).

Kennedy et al 2015 simulated the evaporation phase through injecting a beam heating over 100s and let the atmosphere evolve until 300 s. They use RADYN HD code (Allred et al, 2005), which utilise the pre-heated flaring atmosphere VAL F as the initial condition merged with the quiet sun corona. Their HD results (type 2) are different from ours (type 1), as described in section 1. *Kennedy et al, 2015* argue that the compression of the chromosphere in their model is achievable only when continuous heating of the atmosphere is applied over 100s.

The authors applied this model to produce the light curves of helium and metals and the contribution function for H-alpha emission (their Fig.7). However, they do not reproduce any of H-alpha line profiles at all. Moreover, it is evident for their HD model type 2 that the HD shock moving downwards is formed too high in the atmosphere (in the lower corona, between 1 and 2 Mm) and only after 50 seconds of heating (see their Fig.5). While their H-alpha is formed between 0 and 1 Mm as seen from Fig. 7, so that a simulated line profile will not see this shock as it is outside its formation region.

Thus, the simulations by Kennedy et al explain very well the coronal emission in terms of light curves. However, they are not relevant to the H-alpha observations we reported where the red shift (and shock) is clearly observed 7 s after the event onset, similar to the observations by Wuelsi and Marti, 1989, Wuelsi et al, 1990. Hence, Kennedy et al paper is not comparable directly with ours for radiative signatures. Although, we mentioned this paper when describing hydrodynamic model types in section 1 and in section 3.1.1 when describe HD model calculated by us.

The reason why our model type 1 demonstrates this effect in the H-alpha line with red-shifts at the very start of the flare, in comparison with type 2 model of RADYN, is because we

adopt a quiet sun atmosphere in the initial conditions for the hydrodynamic response (described in section 5.2). Whereas RADYN adopts an initial atmosphere of a flare (model VAL F) (in contrast to ours) and adds the QS corona to make this flaring atmosphere extended from the corona to chromosphere. Hence, the formation of the shock front in our model will be different and more amplified due to the development of larger temperature and pressure gradients upon the beam heating (aka the “hammer effect”). Moreover, they use linear depths instead of Lagrangian coordinates (column depth), which reduces their simulations region to a smaller region, a sub-part of the region simulated in type 1 HD models (covering 5 orders of magnitude of column depths) or even in type 2 models developed for Lagrangian coordinates (see Fisher et al, 1985).

Also RADYN uses the truncated heating function that reduces the heating by beam electrons in the upper chromosphere, while our model uses the full power of electron beams (see also the discussion in section 1 of the paper and the text above in this answer). In our model the heating function has a maximum at the stopping depth of the lower energy electrons (In the upper chromosphere), while heating function in the model of Kennedy et al. is truncated (see section 1 in the paper and the description above). This is why the beam heating of ten seconds in the type 1 HD model we use, is sufficient to produce a strong shock in the chromosphere exactly at the time reported by H-alpha observations. The paper by Kennedy et al, 2015 is now mentioned in section 1 with a general discussion of type 1 and type 2 HD models.

In summary, given the close correlation between H-alpha signatures from our model and the observations at the same time after a flaring event onset, we can conclude that our HD atmosphere (type 1) and radiative model are more applicable to the flaring events, which start without preheating (or flare pre-cursors). The strong shocks formed at the early times of flaring event onset are shown in our paper to be very important evidences that electron beams are the agents transporting energy to much deeper atmospheric levels via a hydrodynamic response. We show from observations that these shocks are formed much earlier than previously thought and simulated by other types of HD models. We also have shown earlier that for very intense electron or mixed (electron and proton) beams these shocks, actually, can even travel beneath the photosphere as reported by Zharkova and Zharkov, 2007, ApJ, and Zharkova and Zharkov, 2015, Solar Physics.

Hence, the main advance of the current paper is the reproduction of these highly red-shifted H-alpha profiles with 1D radiative and hydrodynamic models in a single slab (or thread), which still stands. As an extra bonus, we also explained correctly, in terms of timing and magnitudes, the velocities of chromospheric evaporation measured with AIA by the velocity predicted by our HD model.

C4. *The RHESSI data analysis is unclear, and so somewhat difficult to interpret. A few points should be clarified:*

C4.1. *Which detectors were used to make the spectrum and images?*

A4.1. The images were made using the CLEAN algorithm and detectors 3-8. The spectra were made from detectors 4, 5 and 9.

C4.2. *Which specific functions in OSPEX were used to fit the data?*

A4.2. As indicated in Figure 1, the spectrum was fitted with thermal + thick target with single power law components. We fitted over the energy range 7 - 40 keV.

C4.3. *The spectrum presented in Figure 1a looks like it could be well fit with only a thermal component. Did the authors check to see whether that was the case? The residuals of the fit would clarify, and should be included in the figure.*

A4.3 This is a fair comment.

In the revised paper we described that we did also fit the spectrum with a) a thermal plus thick target components and b) a single thermal component, and found that the chi2 and residuals were comparable for both cases. The residuals are now added to the figure 1.

But because the H-alpha emission occurred very quickly and simultaneously with HXR emission, we found that an electron beam is a more suitable agent for delivering non-thermal excitation and ionisation in this flare than thermal electrons with higher temperature, because only sub-relativistic electrons can reach the chromosphere in under 1 s timescale and excite/ionize hydrogen within seconds to account for H-alpha emission appearing simultaneously with HXR emission. This is now explained in the revised paper, section 5.2.

C4.4. *The authors state that the energy flux was 10^{10} erg/s/cm², but give no indication as to how this was determined, nor what cross-sectional area was measured (or assumed) in order to determine the flux. The results of hydrodynamic simulations depend strongly on the beam flux, so it is important to explain how the authors arrived at this number.*

A4.4. We agree, and thank the reviewer for this comment.

In the revised paper we included the calculation of the flaring event 2 area in section 2.1, and the initial energy flux of beam electrons based on the total electron flux and the area where it was deposited. We used the area of the H-alpha kernel where the line profile was obtained.

C4.5. *How long were the integration times used to make the RHESSI images? Which level contours are displayed?*

A4.5. The integration times were 20s as the photon numbers are very low. Contour levels are 30%, 50% and 70% of the maximum intensity. This is now indicated in the text of section 2.1.

C5. *Regarding Figure 4, the initial (t=0) values should be displayed on the plots. The authors mention in the introduction that the assumed initial atmosphere was for the quiet sun. Does*

this mean that VAL C or FAL C was taken as the temperature profile? Or something else? The results of the simulation depend strongly on the assumed profile, so this should be described.

A5. This is a valid point. We have to indicate that we did not use either VAL C or VAL F models for the flaring chromosphere. We also did not ‘attach’ the QS corona’ to the flaring model VAL F as Fisher et al, 1985a-c or Allred et al, 2005 did.

In our model the corona in a flaring atmosphere is obtained directly from the quiet sun chromosphere heated by electron beam which causes a hydrodynamic response (hammer effect) by sweeping the plasma to the lower atmospheric levels. This sweeping creates flaring atmosphere with its own corona, chromosphere and photosphere. This makes our HD simulations self-consistent. This approach even worked well to interpret a seismic response of the solar interior generated by such the hydrodynamic shocks and production of sunquakes (Zharkova and Zharkov, 2015, Sol. Phys.).

In the revised paper we explained in the revised paper, section 5.2 Method, that the hydrodynamic models used are taken from the paper by Zharkova and Zharkov, 2007, ApJ, and added the details about the initial conditions. These conditions are described in the answer A3 above. This is similar to the approach by Somov et al, 1981, Sermulinya et al, 1982, Duijveman et al, 1983 and Nagai and Emslie, 1984, who also used a hydrostatic atmosphere as the initial condition.

Also in the revised paper the initial atmosphere is displayed on Fig.4c by the straight line presenting a hydrostatic density prior a flare onset. This is now included this information into the caption of Fig.4 and section 5.2. The constant temperature and zero macrovelocities are not required to be presented as they are constant.

C7. *Does the temperature in Fig. 4a refer to the electron or hydrogen temperature? What about the density in 4c?*

A7. This is the kinetic temperature of electrons and density is the total plasma density. The ion density is also calculated but not presented here as at the first seconds their differences are negligible. This is now clarified in the caption.

C8. *I have a few queries regarding the hydrodynamic simulation.*

C8.1. *If the heating lasts only 10 seconds, how is it that the flows are sustained for more than 100 seconds (Figure 4b)? The over-pressure that drives evaporation should begin to dissipate as soon as the heating ceases, as radiation and advection carry energy away.*

A8.1. No, the heating does not stop immediately after the beam is switched off, because the hydrodynamic approach includes conductive processes (which are diffusive by nature and work on large scale column depths used in our model), in addition to that heating by beam electrons. As it was shown independently by Shmeleva and Syrovatskii, 1973, 33, 341; Kostyuk and Pikelner, 1975, Sv. AJ, 18, 590; Somov et al, 1981 and McClymont and Canfield,

1983a, ApJ, 265, 483 (section ii) Hydrodynamics, 2nd par, 3rd line), the characteristic times of the ambient plasma heating via a thermal conduction and radiative cooling vary from 10 to 100 seconds depending on the temperature in the corona and chromosphere, with the average time of 30s (Shmeleva and Syrovatskii, 1973; McClymont and Canfield, 1983).

The vast majority of hydrodynamic papers from 70s and 80s and the modern papers show the hydrodynamic simulations of flaring atmosphere comprising of the impulsive heating by beams or heat fluxes (called the impulsive phase) lasting 5 (Nagai and Emslie, 1984, Fisher et al, 1985a-c, Peres et al, 1987 and many others) or 10s (Somov et al, 1981, Sermulinya et al, 1982, Solar Phys.; Zharkova and Zharkov, 2007, ApJ) with the cooling period after the beam is switched off (called the gradual phase of a flare) lasting for a few thousand seconds and longer (see also reviews from 80s, book by Somov, 2000 or Priest, 2000).

In the revised paper: We explained the hydrodynamic processes more clearly in sections 1 and 5.2 as well as included characteristic times of hydrodynamic heating with the references to the pioneering papers by Shmeleva and Syrovatskij, 1973, MacClymont and Canfield, 1983, Somov, 2000 (Monograph).

C8.2. *The authors should comment on the assumed heating duration. 10 seconds is significantly shorter than the duration of the HXR burst.*

A8.2. The duration of hard X-ray emission varies from milliseconds (Kiplinger et al, 1983, ApJL,265, L99 (SMM); Kundu et al, A&A, 420, 351 (Yohkoh); Charikov et al, Proc. IAU Symposium, Vol. 223. Cambridge University Press, 2004. 429 (CORONAS)) to five-ten or even a hundred seconds (see for example, De Jager and Boelee, 1984, Solar Physics. 92, 227).

Nagai and Emslie used two timescales in their heating timescales: a 2 s preheating duration and a 60 s duration of the main beam heating. Although, their heating function was truncated in the upper chromosphere to avoid infinity at the stopping depth of electrons with lower cutoff energy (see Mauas and Gomez, 1997, ApJ, 483, 496). Fisher et al. 1985 and Allred et al, 2005 used 5 second beam spikes, Somov et al., 1981, Mariska et al, 1989, Zharkova and Zharkov, 2007 use 10 seconds with the smooth heating function derived by Syrovatskii and Shmeleva, 1972, and so do Pikel'ner and Kostyuk, 1976, who considered 100 s of beam injection. They are all valid for different events as discussed in paragraph above.

Our model uses 10 s as it provides the reasonable explanation of the observed H-alpha profiles and AIA jet. In this event 2 we had a very precise location where H-alpha kernel was detected within the first 5-7 seconds after the flare onset, whose emission coincided temporarily with the HXR emission. This defined the time of 10 s chosen by us for the duration of the electron beam impulse in this flare.

Reply to the referee by Druett et al.

For other flares we can use other durations, or a set of repeated short bursts of beam injection to cover the HXR duration for up an hour or so (as suggested by Somov et al, 1981 and by Nagai and Emslie 1984).

C8.3. *Zharkova & Zharkov 2007 remark that they used coronal abundances in the radiative loss function (Equation 9 in that paper). This is incorrect for the chromosphere simulated here, and could affect the resultant evolution of the atmosphere. The authors should make sure to use photospheric abundances in this simulation.*

A8.3. ZZ07, similar to all the other papers (Somov et al, 1981; McClymont and Canfield, 1983b, ApJ, 265, 497; Nagai and Emslie, 1984, ApJ, 279, 896; Fisher et al, 1985a-c; Allred et al, 2005), use the standard radiative cooling curves in the corona calculated by Cox and Tucker (1969) or McWhirter et al (1976) for the optically thin coronal emission for the solar (which reasonably called coronal) abundances. The use of the coronal emission is defined by the fact that it has higher frequencies, and thus has the highest energies compared to any lower energy emission from the lower atmosphere. The contribution of the lower energy hydrogen emission from the lower atmosphere was included by ZZ07, similarly to McClymont and Canfield, 1983b, who expanded the radiative losses curve to the lower temperature emission (<20000 K) with hydrogen (Ly, Ba, Pa and Br) lines and continua emission to account for radiative losses at the chromosphere and photosphere. This is described in the paper by Zharkova and Zharkov (2007, ApJ).

Following the referee comment, this point is now clarified in the revised paper in the section 5.2, we called these solar abundances.

C9. *In Figure 5, it is difficult to compare the simulations to observations since the latter plot has been normalised. It would be more helpful to see the plot in the native units of the detector, as the derived intensities are an important feature that should be reproduced by a faithful simulation.*

A9. We accept the comment by the referee and corrected the paper accordingly.

In the revised paper Methods section 5.1 now contains a discussion of the normalisation process, which describes the observations in the native data unit counts.

C10. *In the fourth paragraph of the discussion section, the authors write that the discrepancy in the flows was either due to the energy flux being higher, or injected for a longer time. However, the authors have not ruled out other causes: a different low-energy cutoff or spectral index, both of which affect the depth of energy deposition; the initial temperature/density profile was significantly different to that assumed; that there may have been a different heating mechanism, such as conduction or waves; or that the line profile cannot be reproduced from a single thread, but requires a multithreaded model (compare e.g. Rubio da Costa et al 2016, ApJ, 827, 38).*

A10. We can only partially agree with this statement. The heating by thermo-conduction via a hydrodynamic response of the ambient plasma, as well as by Coulomb collisions with beam electrons, is already included in our hydrodynamic approach (see A8.1) as it clearly described all the papers considering hydrodynamics (Somov et al, 1981, Nagai and Emsle, 1984, Fisher et al, 1985a-c, Allred et al, 2005, Zharkova and Zharkov 2007). In our model, similar to Somov et al, 1981, we consider separately the heating of the ambient electrons and ambient atoms / ions with the two energy equations (see the original paper ZZ2007 for these equations). This is why the heating does not stop immediately if one switches off the injection of beam electrons (see also the answer A8.1).

The parameters of HXR emission for the flare are defined by the RHESSI energy spectra, which give us a spectral index, the lower cutoff energy and the number of electron flux per second (10^{26} erg/s) from which we can calculate the initial energy flux $F_0 \sim 10^{10}$ erg/cm²/s used in hydrodynamic and radiative simulations for the flare area ($1'' \times 2'' = 750$ km x 2×750 km) measured from H-alpha line ($\sim 10^{16}$ cm²).

Although, these parameters have some uncertainties, they cannot dramatically change the outcome of the HD simulations because (a) the initial cutoff energy cannot be lower than 10 keV (b) the spectral index can only be reduced to 2.82, a change that is virtually undetectable in the HD model. The only parameter which can affect the HD model is the initial energy flux, but given the low HXR photon counts this flux cannot be from the electron beam. Hence, there could be a mixed beam (indicated by Gordovskyy et al., 2005, JASR, 35, 1743; Zharkova and Zharkov, 2007, 2015), which can contribute to the energy flux for HD without showing in HXR emission.

In the revised paper we discuss possible reasons (flux, index or duration of injection), which can affect resulting H-alpha line profile from the HD model used are discussed in the paper, section 3.2.

The current model uses heating by beam electrons and thermal conduction is a part of it (See Methods section 5.2, and eq. 3-6 in Zharkova and Zharkov, 2007).

The temperatures and densities in the flaring atmosphere are uniquely defined by its hydrodynamic response to the injection of beam electrons with these parameters, and this response is presented in Fig. 4. For each second of hydrodynamic response we calculated the radiative response of hydrogen atoms, in general, and H-alpha line profiles, in particular, for these temperatures T , densities n and macrovelocities V . We then compared our simulations with those observed for similar timings. Hence, the produced profiles are unique and appropriate only for this time of HD response in the flare, e.g. for each particular model for a given second after the beam (or flare) onset.

Waves in the atmosphere including Alfvén waves quickly dissipate (even if their coronal speed of 2000 km/s can be increased by some mechanism by the order of magnitude to

20000 km/s as suggested by Fletcher and Hudson, 2008, ApJ). Hence, during their propagation through flaring atmosphere, waves cannot produce such the quick changes in HXR and H-alpha profiles as observed in flares. Also the time of flight calculated from observations for the agents producing observed HXR emission, as shown by Aschwanden et al (1996,1997, 1998) clearly show that only relativistic electrons with $V \sim 1/3$ of speed of light can comply with the time delays measured from observations (Aschwanden et al., 1996, ApJ, 470, 1198; Aschwanden and Schwarz, 1996, ApJ, 464, 974). Moreover, if HXR and optical emission is produced by waves, it would result in the halo emission and not in so well directed and correlated HXR and H-alpha emission in the locations where it is observed in flares. Although, for the sake of clarity in the future it is possible to consider heating by waves if the heating function is produced (no such heating function exists).

As far as the simulated H-alpha profiles from *Rubio da Costa et al, 2016*, they have more notable similarities of the profiles observed when the atmosphere relaxes and the H-alpha shock signature returns to equilibrium (>30s in our case). Their H-alpha profiles show a good self-absorption in the emission line core with symmetric horns, which are a result of radiative transfer effects with large optical thickness.

However, these profiles by *Rubio da Costa et al, 2016* do not demonstrate any strong dominant red shifts (exhibiting peaks out to greater than or equal to +1 Angstrom) reported by Wuelser and Marti (1989) from Locarno telescope and definitely not such the big red shifts as we report here.

In our flaring event we consistently see the same profile in many pixels in the locations of the observed H-alpha ribbon source (extracted from the red box in Fig. 3). In fact, we consistently see the similar H-alpha profiles with large red shifts in a few different pixels of the H-alpha ribbon that does not agree with the idea of multiple threads (or slabs) having different horizontal or vertical velocities of slabs.

And we reproduce these H-alpha line profiles using a simple 1D radiative model in a single 1D flaring atmosphere with the physical conditions produced by a hydrodynamic response. Clearly, we do not require a complex multi-thread model, in order to explain the observed H-alpha line profiles with strong red shifts observed at the flare onset, and this is a significant finding of the current paper.

We added some discussion of these topics into the section 3.2 and 4 of the revised paper.

C11. Minor comments:

The introduction has neglected recent developments on the synthesis of H-alpha from radiative hydrodynamic simulations, which have made significant strides over the work done during the 80s referenced in the intro. For example, the recent work by Rubio da Costa et al 2015, ApJ, 804, 56; Rubio da Costa et al 2016, ApJ, 827, 38; Kowalski et al. 2016, ApJ, in

press (arXiv: 1609.07390). The H-alpha profiles of the last one, in particular, seem in close agreement to the observations presented in this paper.

A11. We appreciate this comment and looked with thanks at all the papers recommended by the referee. In the revised paper we added these citations and discussion of relevant points from these papers.

C12. *In the last paragraph of Section 3.1, the limit to up-flow speeds was derived by Fisher et al 1984, ApJL, 281, 79, and can readily exceed 1000 km/s, confirmed by many later papers. Whether the flows are explosive or not is not solely a function of energy flux, either: see Reep et al. 2015, ApJ, 808, 177.*

A12. We partially agree with this comment and refer to the paper by Somov et al, 1981 (their Fig. 4) for the beams with the energy flux of beam electron of 10^{11} erg/cm²/s where the evaporation velocities approaching 1600 km/s at the maximum of the hydrodynamic response at 100 seconds before the evaporation velocity starts slowing down. Thus, our statement does not contradict to the conclusion by Fisher et al, 1985, who also stated that the evaporation velocities are directly proportional the beam initial energy fluxes. Although, Zharkova and Zharkov 2007 shown that the hydrodynamic heating depends not only of electron flux, but also on spectral index of electron beams, making more explosive evaporation for beams with higher indices.

In the current paper we use the HD model from ZZ2007 for the initial energy flux $F_0=10^{10}$ erg/cm²/s and spectral index of 3 for electron beam energy spectrum. For this HD model the evaporation velocities reach up to 500km/s at 100 s that seems reasonable to be considered as moderate evaporation as defined by Fisher et al. 1985. We added the citation to this paper in the discussion.

The paper by Reep et al, 2015 used a very simplified (by their own admission) heating rate by isothermal electrons following the model by Klimchuk et al, 1987. This model was designed to explain coronal heating (by the own words of Klimchuk in the first paper), but never intended to explain a full complexity of flaring atmospheres. Their hydrodynamic equations are similar to Somov et al, 1981 but written for a linear depth, thus missing on large scale conductivity effects. Then their numeric scheme is significantly simplified to considering a linear scale instead of Lagrangian coordinate used in more accurate simulations by the other authors (Fisher or Somov). Moreover, since their heating function is also over-simplified (assuming the equal heating per volume) for the purpose of demonstration to mono-energetic electrons, the conclusions of simulations by Reep et al. 2015 are not directly comparable with those by Fisher et al or by Somov et al, 1981 and ours (Zharkova and Zharkov, 2007).

Moreover, our simulations by ZZ07 have shown the velocities of evaporation depend not only on a beam flux but also on their spectral index, increasing the evaporation speeds but decreasing the speeds of hydrodynamic shocks moving downwards for softer beams. For

Reply to the referee by Druett et al.

larger HD shocks harder electron beams are required as demonstrated by Zharkova and Zharkov, 2015 in relation to formation of sunquakes by hydrodynamic shocks. These effects of spectral index are not present for the heating by mono-energetic electron beams used by Reep et al, 2015; or by Bradshaw and Cargill, 2013.

Reep et al compared their heating functions with the ones introduced by Emslie (1978) and Nagai and Emslie (1984) used by Fisher et al, 1985. But these functions are known to have a large deficiency in the electron numbers overestimating them at deeper layers because they use a flux conservation instead of continuity equation (Mauas and Gomez, 1997, ApJ, 483, 496; Kontar et al, 2011, 2011, SSRv, 159, 157 (p.6, footnote)). Hence, we believe the conclusions by Reep et al, 2015 can be accepted for this heating function but may be not applicable for the heating function used by Somov et al, 1981 or Zharkova and Zharkov, 2007. On this reason we felt that for the paper clarity it is better to omit this paper from reference list.

In the revised paper we added a comparison of the evaporation rates obtained by us with those by Fisher et al, 1985 to reflect that velocities of evaporation depend not only on the beam flux but also on their spectral index, increasing the evaporation speeds but decreasing the speeds of hydrodynamic shocks for softer beams. We also mentioned the difference in heating functions and thus, the difference in the evaporation and shock velocities obtained in these two types of models as the most appropriate for a comparison.

C13. *In the second to last paragraph, there are many previous studies that the authors should not overlook, studying both hot upflows and chromospheric downflows: e.g. Del Zanna 2008, A&A, 481, 49; Milligan & Dennis 2009, ApJ, 699, 968; Graham & Cauzzi 2015, ApJL, 807, 22; Li et al 2015, ApJ, 813, 59; Tian et al 2015, ApJ, 811, 139.*

A13. Many thanks for these observational papers.

In the revised version we included these paper citations into the Introduction and discussion sections.

Reviewer #1 (Remarks to the Author):

The authors did not solve the major issue I raised.

Fig.4 unequivocally shows that in the corona the temperature drops continuously from TWO (not twenty) to ~ 0.1 million degrees K, and there is no plasma evaporation from the chromosphere to the corona. So, the model is unable to explain the soft X-ray emission. This might be either because the beam pulse is too short or weak, or because of model limitations, or both.

Contrary to what the authors say, this is not in agreement with other models which instead show a density increase in the corona in the first 50 s and no such temperature drop (Fig.3 in Allred et al. 2005).

In the end, the model explains ONLY the H-alpha emission in the low atmosphere, and is inaccurate for the corona. In my opinion, this is not a sufficient improvement to previous works (e.g., Allred et al. 2005) to justify the publication in Nature Communications.

Reviewer #2 (Remarks to the Author):

This is my second review of the manuscript. I am quite satisfied with the revised version. The authors did a very good job on improving the paper. The answers to all comments provided by the three referees are comprehensive and convincing. Many figures were significantly modified and are now at the level acceptable for publication. I therefore have no further comments and recommend this manuscript for publication.

Reviewer #3 (Remarks to the Author):

The authors have significantly clarified much of their work, and given thorough explanations where appropriate. The paper as it stands has considerably improved, both in terms of the text and the figures. It read much more clearly now. I have two concerns left, though, that should be addressed.

The first regards the RHESSI analysis, and the parameters derived from it and then used in the simulation. It still needs more detail to properly interpret, and importantly to reproduce. I attained

rather different fits from the spectrum when I attempted it myself. Figure 1a, right, shows the X-ray spectrum during event 2 (UT 09:15:54), integrated over 20 seconds according to the text, using detectors 4, 5 and 9, and fit in the range 7-40 keV according to the response.

- What time period was used for the background?
- Why did the fit extend to 40 keV? There is only noise above ~ 20 keV.
- The plot indicates that only detector 5 was used in the spectrum, and it's not clear if this is a simple typo. It also indicates the interval as being integrated over 2 minutes.
- The derived parameters from the thick2 component indicate that this component was fit assuming a double power law. This is in contrast to the simulation, which assumes a single power law as explained in Section 5.2. (The six parameters listed at the bottom of the plot, respectively: electron number flux in 10^{35} electrons/s, spectral index of the electron distribution below the break energy, break energy in keV, spectral index of the electron distribution above the break energy, low energy cut-off and high energy cut-off in keV).
- Using the numbers given from the thick2 fit, the non-thermal power would be closer to $\sim 6 \times 10^{26}$ erg/s (Holman et al 2011), which would raise the energy flux in the simulation by a factor of 6 and drastically affect the results.

The second major concern regards the comparison presented in Figure 5. It would be much easier to compare if the two plots were on the same intensity scale. I again request that either the observations should be left in their native units (not normalised), or the simulation should be normalised by the same scaling factor as the observations. It is impossible to compare the absolute intensity without doing so.

Further, the claim that the "simulation profiles closely match with the observed profiles" or that the match is "near perfect" is dubious at best. In 5a, for example, the ratio of the red to blue wing peak intensities is a factor of 2 in the observation and a factor of 4 in the simulation. The observed red shift peak is 50 km/s, while the simulated one is closer to 30 km/s. The absorption feature at line center is not shifted in the observations, but blue shifted by about 10 km/s in the simulation. 5b has similar problems. The authors should temper their conclusions appropriately.

Authors' reply to the comments by the reviewers

Reviewer #1

C1. Fig.4 unequivocally shows that in the corona the temperature drops continuously from TWO (not twenty) to ~ 0.1 million degrees K.

A1. We thank the reviewer for the feedback. Indeed, the upper temperature produced for the *flaring* corona heated by an electron beam was 2MK, which was reduced to 7×10^5 K at 10s and 2×10^5 K at 20s. We have adjusted the text of the paper accordingly. For comparison of these temperature variations to the similar variations at the similar times in A2005 model see A4 below.

C2. There is no plasma evaporation from the chromosphere to the corona.

A2. Indeed our previous version of the paper was not clear on this point. To clarify this misunderstanding, we now explicitly outlined the points in our model in section 3.1.1 and section 3.1.2, which clearly demonstrate that there is plasma evaporation present in our model as follows:

In section 3.1.1 we now explicitly emphasize the presence of evaporation of chromospheric plasma into the corona and indicate that with the velocities of evaporation shown in Fig.4b, middle plot, box 1 (upflows, with negative macrovelocities). The critical point of our model is that it initially evolves from the quiet sun chromosphere into the newly formed flaring atmosphere with its corona and chromosphere. As a result, the plasma material from this new flaring chromosphere immediately evaporates to the flaring corona with velocities up to 400 km/s, as shown in our Fig.4b (or in the middle plot). We now described more clearly how this evaporated upflow (caused by beam heating) starts from the very first seconds and then continues, after the beam heating ends, for the remainder of our simulations (100s). The evaporation velocities, reported by Somov et al. (1981), Sermulinya et al. (1982), Zharkova and Zharkov (2007) are similar to the results shown later by Fisher et al. (1985a-c) and reported in our paper.

For further clarity on this point, we emphasize again that the temperature and density changes during the first seconds of the hydrodynamic response reflects the process of a beam sweeping the quiet chromosphere plasma to deeper layers and converting this initial quiet Sun chromosphere into a flaring atmosphere with its own corona, transition region and chromosphere. We then emphasize that, as a consequence of this sweeping to deeper layers, at later times, when the sweeping is completed, the evaporation of the material from the flaring chromosphere will increase the coronal densities until the final relaxation of this flaring atmosphere back to a quiet Sun chromosphere.

In section 3.1.2, we report that the simulated velocities of plasma evaporation in a flare are confirmed with the observations from AIA, showing a very good match in timing and temperature range. We hope that this revised version of the paper undoubtedly demonstrates the plasma evaporation present in our model to the reviewer's satisfaction.

C3. So, the model is unable to explain the soft X-ray emission. This might be either because the beam pulse is too short or weak, or because of model limitations, or both.

A3. The referee raises an important point with regard to SXR emission, which we appreciate. However, in this paper we do not attempt to explain the soft x-ray emission. In this paper we explain the EUV emission measured by AIA together with optical H-alpha emission observed by

SST. As we show in the paper, our radiative model for hydrogen atoms combined with the HD model by Somov et al 1981 and Zharkova and Zharkov, 2007 shows a very good agreement with the observed H-alpha profiles and, now, the EUV emission at the very first seconds after a flare onset.

In the forum of scientific discussion, for the purpose of considering SXR emission in a flaring atmosphere, we can again outline plausible explanations for the prolonged SXR emission in flares (i.e. see the previous reply to reviewer 1, A2 and section 1, 1st paragraph of the paper). In general, this can result by assuming either: *a*) a continuous density increase in the corona as pursued by Allred et al, 2005 or Kennedy et al, 2015, or *b*) non-thermal ionisation of the coronal ions by energetic particles combined with their slower recombination rates as we explained in section 1 (Porquet et al, 2001, A&A, 373, 1110; Kawate et al, 2016, ApJ, 826, 3). These options could be explored in the future using our model and that will be the scope of the future paper. In summary, we would like to thank the referee for raising this important point with regard to SXR emission but this will be the scope for future research.

C4. Contrary to what the authors say, this is not in agreement with other models, which instead show a density increase in the corona in the first 50 s and no such temperature drop (Fig.3 in Allred et al. 2005).

A4. We agree with the reviewer that our model differs from the results of Allred et al., 2005 (A2005) in several key aspects, and next we will expand on the reasons for both the similarities and differences. In general, we wish to say that there is nothing wrong for different models to produce different results because they explain different scenarios or different aspects of these events that can be beneficial for the overall investigation of solar flares.

We thank the reviewer for highlighting the references made in our paper. The text and references cited in section 3.1.1 have been amended to reflect the differences and similarities of HD models used by different authors.

Below we elaborate further on the reviewers point relating to the agreement of our model with the other models:

A4.a. Comparison with Fisher et al.1985 model. Let us first compare our results with the results of Fisher, Canfield, and McClymont (1985) (FCM85) model, who found HD response in Lagrangian coordinates using a short (5s) duration of intense heating by electron beams, similar to our model (10s). The simulations of FCM85 (Figs. 4, & 5) are shown below for the initial fluxes of beams that are close to our model.

From their Figures 4 and 5 we can observe the following similarities to our simulations:

- The coronal temperatures by FCM85 (top rows in both Figures below) are very similar to those in our model F10 for the flux of 3×10^{10} erg/cm²/s (compare our Fig. 4 (top plot) and FCM85, Fig. 4 below) and in model F11 (flux of 10^{11}) by Somov et al, 1981 (their Fig. 2 close to Fig. 5 below by FCM for 5×10^{10} erg/cm²/s).
- The speeds of chromospheric evaporation (FCM85 Fig.4, 5, bottom row of plots) generated in the flaring corona by both models heated by electron beams are also similar to those simulated by us. In Fig.4 max evaporation speed at the column depth of 10^{18} cm⁻² approach 350 km/s in ours and theirs (column for 4s) or in Fig. 5 the speeds approach 400 km/s (at 4s) in Fisher and 400 km/s (4s) in Somov et al, 1981 (their Fig. 4).

- Coronal densities in FCM85 (second row of plots). The plots from left ($t=0s$) to right ($t=7s$), show (as highlighted by the red circles) that there is a reduction of the chromospheric densities to the coronal ones at 4s for the Fig. 4 (FCM85) that is similar to our model shown in our Fig.4.
- The magnitudes of the density decrease calculated for the same time (4 s) in Fig.4 of FCM85 model are similar to those in our F10 model (Fig.4), or in Fig. 5 of FCM85 to Fig.3 in Somov et al for F11 model.

FIG. 4.—Same as Fig. 2, but with $F_{20} = 3 \times 10^{10} \text{ ergs cm}^{-2} \text{ s}^{-1}$

FIG. 5.—Same as Fig. 2, but with $F_{20} = 5 \times 10^{10} \text{ ergs cm}^{-2} \text{ s}^{-1}$

These similarities between our model and others also continue after the beam is switched off:

- Coronal densities (in FCM85 Figs.4, 5 second row panels at 7s, red circles) continue to decrease because of the plasma sweeping by a beam towards the chromosphere. This is similar to our Fig. 4c (bottom plot) and to Fig. 3 in Somov et al, 1981.
- Also chromospheric evaporation at 7s becomes much stronger that is evident in the plots of macrovelocity (bottom row of plots) in both Fisher's models (fig4 and Fig.5 below) and our HD model F10 (see our Fig. 4b in the paper) and in S81 model F11 (their Fig5).

We hope that our answer provides further clarity to the reviewer concerning the origins of the initial quiet sun density decrease in our model while forming a flaring atmosphere followed by a standard increase of coronal density in this flaring atmosphere which is similar to other HD models in the similar conditions. For example, Fisher et al. 1985 show similar density decreases in the first 50 s when the beam reaches the quiet sun chromosphere and converts the quiet Sun chromospheric densities into coronal densities of a flaring atmosphere.

A4.b. Comparison with A2005 model. There are some differences and some similarities with this model. A2005 consider a pre-heated flaring atmosphere softly heated afterwards for 3-4 minutes

(226s) by an electron beam, in contrast, we start from the quiet sun chromosphere that is sharply heated (within 10s) by bursts of electrons. These two types of hydrodynamic models explain the two fundamentally different scenarios of flare heating and we trust that the reviewer accepts they are both acceptable for different events (i.e. pre-heated flaring events for flares with pre-cursors described by model A2005 and the initial impulsive heating described by our model).

The differences are explained in our previous answer A2 to the reviewer 1 and section 1 of our paper. For these reasons, the one-to-one agreement between our (S81, ZZ07) model and A2005 is not expected to be a pre-requisite for consideration as they can account for different types of events. We hope, the reviewer kindly agrees with the principle behind this statement.

Despite these differences, there are a number of similarities between S81, ZZ07 and A2005 HD models outlined below:

1) Coronal temperature decrease. Fig.3 from A2005 is presented below.

FIG. 3.—Solar atmosphere at four times during the gentle phase of the F10 flare. The top row shows the logarithm of the temperature T as a function of height compared with the preflare state (PFS). In the middle row, the electron density n_e (left axis) and beam heating rate Q_e (right axis) are plotted. The bottom row shows the mass density ρ (left axis) and hydrogen ionization fraction X (right axis). Note the change in scale of the horizontal axis in the last column.

It can be seen from Fig.3 of A2005 that there is in fact a reduction in the coronal temperature (marked by the red circles we have overlaid) at the times before 50s (see A2005 Fig. 3 (above), panels 3 (3s) and 4 (50s)). This point is in contrast to the statement made by the reviewer in their report. This reduction is similar to the reduction reported in our HD model (Fig.4a) for the same time range 30-70s) calculated for the model F10. The range of the temperature decrease in A2005 (from 800 000 K to 100 000K) is the same as in our model shown in Fig.4a.

However, the difference is that in our model the temperature is reduced (our Fig.4a) because the beam is switched off after 10 seconds, whereas in A2005 Fig.3 the temperature is reduced (compare the last two top panels in Fig.3 above from A2005) due to the co-temporal density increase (shown in the bottom panels in their Fig.3) during the prolonged heating by a mild beam. We hope very much that we have now explained clearly the reasons for a density decrease in our model (from the quiet Sun chromospheric density to the coronal density of a flaring

atmosphere). The model of A2005 also shows the similar density reductions at the times when the beam heating reaches the chromosphere (see the point 2 below). In this sense, the both models (ours and A2005) are consistent for the scenarios of the flares they model, as we stated earlier in A2.

An important new finding in our paper is that this reduction of the coronal temperature allowed us to capture the coronal jet (or chromospheric evaporation) with the EUV observations by AIA (reported in section 3.1) and to equate the simulated and observed velocities of this evaporation. At the same time, these blue shifts in the EUV line were co-temporal and co-spatial with the unique strong red shifts in H-alpha reported by us in sections 2.2 with co-observations from CRISP and simulated with our radiative and HD models in section 3.2. This represents a strong case for the support for a scenario of fast impulsive heating by an electron beam in this flaring event, as described by our HD and radiative models.

2) Coronal density decrease. In the HD model of A2005 for F11 beam, A2005 report (A2005 Fig.5 below) a rapid density decrease and increase of the temperature (at 6-15s) occurs when the heating agents reach at the depths of the quiet Sun chromosphere, i.e. between 1 and 1.5 Mm, as shown in the vertical panels 3 and 4 (bottom row, marked by red circles). This is, again, in agreement with the results of HD model F11 simulated by Somov et al, 1981 (their Fig. 3).

This shows there is also sweeping by a beam of some coronal plasma towards the chromosphere in A2005 model (their Fig.5 below) that agrees with our model F11 from S81 (Fig.3).

FIG. 5.—Solar atmosphere at four times during the F11 flare. The quantities plotted are identical to Fig. 3, except that the He III fraction is plotted in the bottom row rather the hydrogen ionization fraction.

We hope that we have clearly explained in the answers above the similarities and reasonable disagreements between our and Allred et al 2005's HD models for the full satisfaction of our respected reviewer and the Editor.

In summary, in the revised paper the following corrections are made:

- In section 3.1.1 the first reference to A2005 (regarding coronal densities) is removed. The reference to Kennedy et al. is also removed, as this simulation has heating for 110s and is based on the model of A2005, additionally the paper by Kennedy et al. does not display densities in the figures.
- The second reference to A2005 in par 1 on page 8 is regarding the differences in the HD shocks in the chromosphere. This reference is retained as well as the other references following the answers A4 above.
- The statement regarding “beam heating over 100 seconds” at the end of section 3.1.1 is clarified.

C5. In the end, the model explains ONLY the H-alpha emission in the low atmosphere, and is inaccurate for the corona. In my opinion, this is not a sufficient improvement to previous works (e.g., Allred et al. 2005) to justify the publication in Nature Communications.

A5. We thank the reviewer for highlighting the need to emphasize the key advances presented in our paper, which is a very essential improvement on the interpretation of H-alpha observations, which was overdue for approximately 30 years, since observations by Ichimoto and Kurokawa, 1984; Wuelser and Marti (1989), Wuelser et al, 1991, 1993.

The changes made to section 3.2 and 4 now state clearly these key advances, which we summarize below for the reviewer’s consideration:

In summary, our results uniquely show the following:

- a) The close correspondence between observed (by the new high-resolution instrument SST) and simulated spectral line profiles with our NLTE model for H-alpha with large (~1A) red shifts caused by strong hydrodynamic shocks at the first few seconds after a flare onset has never been reported before;
- b) The simultaneous observation in EUV emission by the state-of-art AIA instrument of a coronal jet confirming evaporation rates for this event at the correct times derived from HD simulations in the first 100 seconds after a flare onset. This is also predicted very accurately by our model with respect to the time of appearance of the jet, temperature, velocity and lifetime.
- c) The speeds and depths of the hydrodynamic shocks in flaring atmospheres simulated with our hydrodynamic model previously allowed Zharkova and Zharkov (2015, Solar Phys, 290, 3963) to explain the helioseismic waves induced by flares (sunquakes, Kosovichev and Zharkova, 1998, Nature, 393, 317) by deposition of the momentum delivered by these HD shocks to the photosphere and beneath. This opens an exciting perspective for the further research combining hydrodynamic, radiative and helioseismic approaches.

The HD shocks with the momentum and energy required for detectable sunquakes cannot be generated in the HD model by A2005, because of the limit of 40 km/s for the shock velocities and higher depths in the atmosphere where they are deposited in the A2005 model, as elaborated below.

What is essential in the current paper that the strong increased of Doppler downward velocities and the red-shifter H-alpha line profiles of the flaring event reported by us are not accounted for by any models either by A2005 or Fisher et al., 1985.

Further elaboration on a comparison with the A2005 model:

1) In the A2005 HD model F11 (energy flux of 10^{11} erg/cm²/s), the maximal downward velocity of the shock in the chromosphere, where the H-alpha line is formed, is 40 km/s while the F10 model by A2005 does not show any significant shocks, or red-shifted H-alpha line profiles (see Fig.8 in A2005).

While in our model the downward velocities of up to 50 km/s are easily achieved for a moderate F10 model at the very first 5 seconds after a beam onset (our Fig.4b) with the energy flux 10 times smaller than in A2005. Moreover, the velocities of this downward motion can reach up to 150 km/s for our F11 model (Somov et al (1981, their Fig.4) or up to 250 km/s for our F12 model (Zharkova and Zharkov, 2015, their Fig. 1), similar to those reported by Ichimoto and Kurokawa, 1984 or Wuelser and Marti, 1989.

FIG. 9.—Line profiles for Ly α , H α , He II λ 304, and Ca II K at four times during the F11 flare (times are indicated at the top of each column). The dotted lines indicate the level of the continuum and the line center, while the dashed line is the preflare line profile. The emission in panels (1) and (9) is too small to be seen on this scale.

2) The A2005 paper's claim to have the consistency with the Doppler velocities measured in H-alpha line by Ichimoto and Kurokawa (1984SoPh...93..105I) (IK84) is not exactly substantiated. The only model by A2005, which reports the H-alpha line profiles with maximum Doppler shift velocities of 40 km/s is the model F11 (shown in Fig.9 by A2005 and reproduced by us here). Its maximal red shift occurs at t=6 s after the beam onset (see the red circles in Fig.9 above from A2005), but very shortly (at 11s, or 5 s later) the red shift is replaced with a large blue shift (see the blue circle in Fig. 9 from A2005).

However, IK84 (their Fig. 4 and 5) report the observations of Doppler velocities ranging from 40 to 100 km/s resulting the red shifts 1-2A in H-alpha line. Moreover, these red-shifts appear at the first

5-15 s after flare onsets and last for a minute or two. The similar or even larger red shifts during flare onsets are reported by Wulser and Marti (1989), Wuelser et al (1993) and by us in the current paper.

It is evident that the greater Doppler shifts (up to 3 Å), which are reported in H-alpha profiles by many authors (IK84, Wuelser and Marti 1989, 1993) at the flare onsets cannot be reproduced by any of A2005 models, including Kennedy et al, 2015.

3) In contrast, in our paper we reproduce these observed H-alpha line profiles with the 1Å red shift using the radiative and hydrodynamic models reported here for a moderate F10 beam. Our H-alpha line profiles show the red shifts about 1Å (or downward velocity of 45 km/s) being in extremely close agreement with the observation from the SST, as discussed in section 3.2.

Furthermore, the combined HD and radiative model used here can explain even more powerful flaring events with the downward velocities up to 200 km/s (or red shifts up to 3-4 Å as reported by some observations) caused by heating with a more powerful beam model (i.e. F11-F12 models) (Zharkova and Zharkov, 2015).

Therefore, the H-alpha profiles generated from our HD and radiative models support the previous and recent H-alpha line observations at the flare onset, representing highly significant results in their own rights.

5) Further to the points 1-4 above, our model also predicts correctly the timing, temperature and velocities (our Fig.4a b, box 1) of the EUV jet observed by AIA (our Fig.2b and our Fig.4). This corroborates the predicted upflows in the hydrodynamic model, and indicates the onset of chromospheric evaporation, as predicted (see also answer A1 to comment C1).

This makes the current paper extremely timely to be published in Nature Communications, because its findings resolve the long standing puzzle of red-shifted H-alpha line profiles at the onset of flares by emphasizing the key role of energetic particles in producing optical ribbons in solar flares. This paper will be of substantial interest to many researchers working in the field of solar and stellar flare evolution.

Reviewer #2 (Remarks to the Author):

C1. This is my second review of the manuscript. I am quite satisfied with the revised version. The authors did a very good job on improving the paper. The answers to all comments provided by the three referees are comprehensive and convincing. Many figures were significantly modified and are now at the level acceptable for publication. I therefore have no further comments and recommend this manuscript for publication.

A1. We thank the reviewer very much for their assistance in improving this paper.

Reviewer #3 (Remarks to the Author):

C1. The authors have significantly clarified much of their work, and given thorough explanations where appropriate. The paper as it stands has considerably improved, both in terms of the text and the figures. It read much more clearly now. I have two concerns left, though, that should be addressed.

A1. We thank the reviewer for their invaluable assistance in bringing about this improvement, and will endeavour to expound and amend our paper in order to resolve the remaining concerns of the reviewer.

C2a. The first regards the RHESSI analysis, and the parameters derived from it and then used in the simulation. It still needs more detail to properly interpret, and importantly to reproduce. I attained rather different fits from the spectrum when I attempted it myself. Figure 1a, right, shows the X-ray spectrum during event 2 (UT 09:15:54), integrated over 20 seconds according to the text, using detectors 4, 5 and 9, and fit in the range 7-40 keV according to the response.

A2a. We appreciate the referee comment highlighting this problem. We have re-fitted the spectrum over the period 09:15:54 – 09:16:14 UT, using the detectors 4,5 and 9 and a thermal + single power law fit and 0.33 keV binning over the range 7-21 keV. As you can see from the revised figure, the chi2 is much reduced and the residuals show less structure. The power-law exponent is still 3.82. The upper energy cutoff was reduced to 21 keV in line with the referee comment about the noise in HXR photons, in this event above 20 keV. This reinforces our initial beam parameters discussed in the paper.

In the revised paper, this updated spectrum was used in the Fig. 1 instead of the previous one.

C2b. - What time period was used for the background?

A2b. The background period was chosen to be 09:38:40 – 09:40:56 UT. This is now added to the paper text.

C2c. - Why did the fit extend to 40 keV? There is only noise above ~ 20 keV.

A2c. The reviewer is quite correct that there are few counts above 20 keV. We only extended the over-plotted model spectrum to 40 keV for a display purposes.

In the revised paper, the new figure has been generated to only show the fit to ~21 keV as the referee recommends (see A2a).

C2d. - The plot indicates that only detector 5 was used in the spectrum, and it's not clear if this is a simple typo. It also indicates the interval as being integrated over 2 minutes.

A2d. This is a valid comment, thanks. It was an illustrative plot which is now removed. In the revised paper, we have replaced this illustrative plot with the more suitable one using the detectors mentioned (see answer A2a).

C2e. - The derived parameters from the thick2 component indicate that this component was fit assuming a double power law. This is in contrast to the simulation, which assumes a single power law as explained in Section 5.2. (The six parameters listed at the bottom of the plot, respectively: electron number flux in 10^{35} electrons/s, spectral index of the electron distribution below the break energy, break energy in keV, spectral index of the electron distribution above the break energy, low energy cut-off and high energy cut-off in keV).

A2e. We thank the reviewer 3 for pointing out this inconsistency. We have re-fitted the spectra as described in A2a that eliminate this problem, the spectrum has now a single power law.

C2f. - Using the numbers given from the thick2 fit, the non-thermal power would be closer to $\sim 6 \times 10^{26}$ erg/s (Holman et al 2011), which would raise the energy flux in the simulation by a factor of 6 and drastically affect the results.

A2f. We accept this comment with thanks. This problem is now rectified, there is no double power law spectrum generated for this event 2, only a single power law one as per answers A2a – A2e.

However, a slight increase of the initial energy flux of electron beam heating the atmosphere, in general, could be one of the options which can lead to a better fit of simulations to observations of H-alpha red-shifted profile, as we discussed in the second last paragraph of section 3.2.

C3. The second major concern regards the comparison presented in Figure 5. It would be much easier to compare if the two plots were on the same intensity scale. I again request that either the observations should be left in their native units (not normalised), or the simulation should be normalised by the same scaling factor as the observations. It is impossible to compare the absolute intensity without doing so.

A3. We accept this point with thanks. We have now adjusted the simulated profiles to the relative units of the observational profiles as the reviewer 3 advised. The observed and simulated profiles can now be more easily compared on the same x-axis and normalised y-axis.

We added a sentence in the text of the paper explaining possible reasons for the differences between the simulated and observed profile intensities in the red and blue wings, while explaining the location of red-shift at 1A, similar to the SST observations (1.1A).

C4a. Further, the following claims are not correct:

a) the “simulation profiles closely match with the observed profiles” or that the match is “near perfect” is dubious at best... The authors should temper their conclusions appropriately.

b) In 5a, for example, the ratio of the red to blue wing peak intensities is a factor of 2 in the observation and a factor of 4 in the simulation.

c) The observed red shift peak is 50 km/s, while the simulated one is closer to 30 km/s.

A4a. We acknowledge these comments and made the following corrections:

- a) We amended our discussion in section 3.2 to reflect some imperfections of our model fit to the observation. We also discussed in section 3.2, last par. the possible options for a better fit of the simulations to the observations, which forms part of a future work.

We agree with the reviewer that the simulated ratio of the red-to blue wing peak intensities in Fig.5a (left) is still higher than in the observations (by a factor of 1.5-2). This comes from a higher peak of the red wing induced by the downward motion of the hydrodynamic shock formed in the upper chromosphere.

These differences can be corrected with a more energetic beam than we used in the current model. More energetic beam will amend the red-to-blue intensity ratio and reduce the intensity of the red wing. One reason why the electron flux could be more intense than that used in the current model comes from the possibility that the flaring ribbon emission has a much smaller source size than the SST diffraction-limited resolution of ~ 100 km in H-alpha. For example, if the emitting area has a cross-section of 50 km, while the pixel size covers 100 km, this will result in a smoothing / averaging of the signal of the observed intensity of the H-alpha profile, thereby, reducing the maximal intensity about a factor 2 for this example.

A smaller emitting area will equate to an increase of the initial flux of beam electrons, in-turn, producing a hydrodynamic shock deeper in the chromosphere in the simulation with larger downward velocities. Such the shift of a beam heating to deeper layers would mean a reduction of the intensity at red-shift wavelength, given that the contribution function of the intensity of line formation is now shifted to deeper layers with larger optical thickness, and thus have smaller intensities. This new scenario would not only result in a reduced intensity in the simulated red wing, but also could remove the slight blue-shift of the central reversal intensity in the model profile, while increasing the intensity of a blue wing, and, therefore, decreasing the red-to-blue intensity ratio in the simulated profile. That would make it much more in agreement with the observed intensity ratio of the red and blue wings.

However, while this scenario is plausible it still assumes that the observed intensity ratio is perfectly accurate, which it may not be, given that we do not have infinite spatial resolution in the observations. That will then have a feedback / influence what the exact values of the initial beam flux should be accepted (within a factor or two). This outstanding issue therefore cannot be fully reconciled beyond the limits of the current state-of-the-art observations and we propose this as a future work with supporting observations from next generation instruments with much higher resolution, such as DKIST.

In summary, given the fact that we do not have higher resolution data, than that provided by SST, we can only present a discussion on these differences in the current paper which we do.

This discussion is added to section 3.2 marked in bold.

- b) We appreciate very much this comment by the reviewer 3 and apologise that our H-alpha profile in the previous version of the paper was a bit confusing.

Actually, the simulated value of the red-shifted peak in the profile data was about 40 km/s and not to 30km/s as it appeared for the reviewer. However, while checking the simulations' results to prepare the feedback to the reviewer's comment we realised that we can improve the simulated profile and that this would improve its correspondence to observations. And for this we are very grateful to the reviewer 3.

Thus, we have conducted the improvements in the calculation of the H-alpha line profile:

- 1) We used a few additional depth points in the radiative transfer model in the locations around the maximum macrovelocities where previously we had only two points between 38 and 55 km/s).
- 2) For calculation of the line emission at a given wavelengths, we calculated synthetic profiles for H-alpha emission at each given layer for all depths, where the line is formed, using the real Doppler half-widths $\Delta\lambda_D$ for the layer (dependent on a temperature). Then for the resulting line profile we added these synthetic profiles for the same wavelengths and plotted the resulting H-alpha profiles over the wavelengths with the effective Doppler half-width $\Delta\lambda_D^{Eff}$ weighted by each layer's optical thicknesses.

Note: Doppler half-width $\Delta\lambda_D$ characterises the line profile width caused by thermal Doppler broadening (thermal motion of hydrogen atoms at given temperature). The Doppler half-width is the width of this thermal profile at half the maximum intensity.

These two corrections above 1) and 2) improved the accuracy of the simulated H-alpha line profile, which now allowed us to distil more accurately the red-shifted peak at about 1 Å, corresponding to the macro-velocity of 45 km/s (see the updated Fig.5a, left), instead of 40 km/s shown in the previous version of the paper. This shift is defined by the macro-velocities in the HD model (up to 55 km/s) shown in Fig. 4c for the chromosphere, or H-alpha core formation region.

This H-alpha red-shift in the model became now really close to the observed red shift just above 1Å corresponding to 45-50 km/s reported from the SST observations (Fig.5a, right).

We hope the referee would agree with us, that there is a close correspondence between the simulated and observed H-alpha profiles in the red shift size (~1Å, or 45km/s). We owe this improved accuracy to the referee comment, which motivated us to improve the model results, thank you very much.

- 3) Updated with this additional resolution the shapes of H-alpha profiles at later times and re-plotted them in Fig.5b, left).

These new points 1-3 are reflected in section 3.2, and the conclusions.

C4b. The absorption feature at line centre is not shifted in the observations, but blue shifted by about 10 km/s in the simulation. 5b has similar problems.

A4b. We agree with this statement with thanks. The similar blue or red shifts of central reversal wavelengths were also reported before in the simulations and observations.

1) Examples of red shifts

- In our observations there is a red shift in the central reversal at 29 s (Our Fig.5b, below left).
- Kuridze et al. (2015) also report red shifts of central reversal of their models after the flare maximum (see the plot below on the right from their Fig.4, below right, green line (red line is a double Gaussian fit)).

2) Examples of Blue Shifts

- Ichimoto and Kurokawa show a small shift in central reversal during some frames (00:19:59 UT below)

Fig. 4a.

3) Simulated shifts:

- Kuridze et al. also note that they were unable to avoid a blue shift in central frequency, or central reversal in simulations where there is a red excess ($t=2s$, their Fig.7 below). They also report a red shift in the profiles with a blue excess. ($t=10-14 s$, their Fig.7 below)

Figure 7. The temporal evolution of the synthesised $H\alpha$ profiles during an F11 flare. The vertical dotted lines indicate the line center. The right panel shows the evolution of line profile asymmetry, I_B/I_R , calculated with the method described in Section 3.

- Heinzel et al 1994 So. Ph. Reports red shifts of central reversals in the H-alpha profiles with blue shifts, similar to Kuridze et al., 2015.

Fig. 5. (a) Synthetic $H\alpha$ profiles (see the text), T disk-center quiet-Sun profile and to the enhanced e flare kernels. (b) The velocity field vs temperature u

Summary: It appears that small shifts of the central intensity in the simulated H-alpha line profiles occur: a) to the blue side for the lines with strong red-shifts and b) to the red side for the lines with strong blue shifts. They are also occasionally observed (e.g. Ichimoto and Kurokawa, 1984 above, or Wuelser and Marti, 1989).

In the revised paper we discussed this point in section 3.2.

In summary, we wish to thank very much the reviewer 3 for his/her comments that allowed us to significantly improve the fit of the H-alpha model profile to observations.

We hope that the reviewer 3 would support now our conclusion that the proposed radiative model in HD atmosphere heated by electron beam can explain some key features of the H-alpha line profiles, which could not be explained for the past 30 years, since the observations by Wuelser and Marti (1989) or Ichimoto and Kurokawa (1984).

Reviewer #1 (Remarks to the Author):

The authors show that their model is in agreement with the others regarding the chromosphere and the base of the corona. They also give up and do not make any attempt to improve their model and account also for the flaring upper corona. As a consequence, they accurately and wisely avoid to discuss the evolution of the upper corona. One is then left with the doubt whether the flaring corona is simply missed by the model or it is ruled by a different heating mechanism, as often invoked in the past (e.g., Peres et al. 1987, Brosius 2012, Battaglia et al. 2015). I also cannot help wondering about what causes the soft X-ray emission shown in Fig.1 (left), which the authors do not explain and which represents a standard marker of a solar flare. I am still puzzled by a model that is unable to reproduce flare temperatures (>10 MK), while others do. The authors should at least mention that the model does not address this issue, which deserves further investigation.

Reviewer #3 (Remarks to the Author):

The authors have improved the paper somewhat, in particular the simulated H-alpha profile looks better. I still think there are outstanding problems, which need to be remedied.

The abstract, Sections 3.2 and 4 once again state that simulations, plural, were performed. This is not true of what's presented in the paper.

I'm in agreement with Reviewer #1 that the model would not reproduce the SXR emission, and contrary to the authors' statement in their response, would not even reproduce the AIA emission, particularly the brightening in the 94 A channel, indicative of high density plasma at a temperature close to 10 MK. The authors could, of course, synthesise AIA emission from their model to validate their claims, as done by many other authors.

The RHESSI analysis has some problems now that there's enough information to try to reproduce it.

1. The background period 09:38:40 - 09:40:56 UT is not a solar background, but something closer to a flat field. RHESSI was entering its nighttime period then, as should be clear from the light curves. The sudden drop at 09:36 in the RHESSI light curves does not occur in the GOES data, showing that it does not represent solar data.

The normal procedure for RHESSI background subtraction is to average the time periods from +/- 15 orbits (approximately 1 day), assuming no flares occurred at those times.

2. The RHESSI spectrum as presented appears to have pulse pile-up, evident from the spectrum at ~ 13-14 keV. The count rates exceed 1000 counts/s in the SXR bands, used as a test for pulse pile-up. The authors are interpreting the excess emission as non-thermal emission from an electron beam, in which case they need to explain how they've ruled out pulse pile-up.

When I use a pulse pile-up correction, the fit finds no evidence for non-thermal emissions (whether done with the 1pow or thick2 functions in OSPEX).

3. I would recommend that the authors continue to use the thick2 function for attempts at fitting, rather than 1pow, which does not fit to an actual bremsstrahlung function from an electron distribution. 1pow cannot give a low energy cutoff E_c , for example, so it's then incorrect to state that the simulation values are based off the observed values.

Thick2 can be fit with a single power-law by setting the break energy greater than the high-energy cut-off parameter, and not allowing them to vary. See the OSPEX documentation here:

https://hesperia.gsfc.nasa.gov/ssw/packages/spex/idl/object_spex/fit_model_components.txt

4. The authors have not explained how they have now obtained their new energy flux or low energy cutoff. The values surely have changed if the fit was re-done. If the differences are significant, the authors ought to consider re-doing the simulation.

The authors also explain a number of differences between the simulated and observed H-alpha profiles by saying that the "differences can be corrected with a more energetic beam." The authors have a model at their disposal to actually test this, and should do so!

Authors' replies #4 to the comments by the reviewers

Reviewer #1

C1. The authors show that their model is in agreement with the others regarding the chromosphere and the base of the corona. They also give up and do not make any attempt to improve their model and account also for the flaring upper corona. As a consequence, they accurately and wisely avoid to discuss the evolution of the upper corona. One is then left with the doubt whether the flaring corona is simply missed by the model or it is ruled by a different heating mechanism, as often invoked in the past (e.g., Peres et al. 1987, Brosius 2012, Battaglia et al. 2015).

A1. We thank the reviewer for their analysis of our previous responses, and the agreement reached regarding the important distinctions between flaring chromosphere and the flaring corona. Our model includes a flaring corona that is converted from a quiet Sun chromosphere with a *transient* coming from the flaring chromosphere, rather than having an *inherent* corona initially. Hence, we do not attempt to explain the formation of a *persistent* temperature of the overlying or neighbouring corona, which becomes an issue with RHESSI resolution discussed in A2 below.

For this particular flaring event 2, heated by a weak beam with the initial energy flux of about 10^{10} erg/cm²/s (with variations of factor 0.7 to 3), it means that the corona does not become hotter than 2 MK and the AIA spectral line synthesis (outlined herein) confirms this with the supporting observations of the ribbon jet into the corona.

However, the hydrodynamic code used in this paper is indeed able to account for higher temperatures up to 20-30 MK, which occur when the corona, formed from the flaring chromosphere, is heated by the beam with the initial fluxes of 10^{11} or 10^{12} (see the answer A3 below) as published in papers by Somov et al, 1981, 1982, Sermulinia et al, 1983, Zharkova and Zharkov, 2007, 2015. Our model comparison with the detailed H-alpha spectral line adds further confirmation that the fluxes of the beam cannot be greater than 10^{10} erg/cm²/s, further ruling out corona with temperatures greater than 2MK (see the answer A2 below).

C2. I also cannot help wondering about what causes the soft X-ray emission shown in Fig.1 (left), which the authors do not explain and which represents a standard marker of a solar flare.

A2. We wish to reiterate our replies to referee 1 from the previous two reviews. The SXR emission Fig.1 (left) is shown in the GOES light curves, which are taken from full disk imaging of the Sun, and therefore include the contributions from many beam injection sites over the large flaring regions (see Figs.1b, 2a and 3a left panel), described in section 2.1, as events 1,2, and 3. We do not attempt to explain the emission in the surrounding corona as we indicated in answer A1, but just in a flaring event with H-alpha emission observed by SST.

In considering the SXR detected by RHESSI, let us compare the resolution per pixel provided by RHESSI (~2"), AIA (0.6") and SST (~0.06"). Following the RHESSI website <https://hesperia.gsfc.nasa.gov/rhessi2/mission/mission-facts/index.html>. and the original paper by Hurford et al. 2002 <http://adsabs.harvard.edu/abs/2002SoPh..210...61H> stating this resolution to be at best 2.3" (rounded to 2" by us) we can evaluate that in one pixel of HXR emission by RHESSI

(2"x2") we have about 3x3 (9 in total) pixels of AIA emission and about 33x33 (~1100 in total) pixels of SST emission. For the H-alpha flaring kernel considered we observed 5x5 (25 in total) pixels of the SST event 2, where H-alpha line emission was recorded, only these pixels were affected by an electron beam modelled in our HD and NLTE models.

One pixel of AIA emission of size (0.6"x0.6") would contain 10x10 (100 in total) pixels of SST, from which only 25 are affected by a beam, as SST observations show. Hence, the contribution due to flaring emission, which supports our model comes only from the area covered by these 25 SST pixels. The area in the remaining 75 pixels of SST (out of total 100) covered by a single pixel of AIA would provide the emission from the neighbouring corona of this active region, which account for its thermal emission.

The situation is even more convoluted for the RHESSI emission for this kernel, where a single RHESSI pixel contains 1100 pixels of SST, from which only 25 have the flaring event with the injected beam, while the area covered by the other 1075 pixels (or 98% of RHESSI data) is the neighbouring corona of this active region, which does not have any effect from this particular electron beam. This explains the occurrence of thermal emission in the RHESSI data and the reason why we should not account for it because this is likely due to the coronal heating problem in the active region and not in the flaring atmosphere considered, since the latter constitutes only 2% of the emission area registered by RHESSI.

In summary, our model is the one of a flaring atmosphere affected by a short injection of a single beam (labelled as event 2 in section 2.1), which is appropriate for comparison with the high cadence and high resolution imaging and spectra of the selected flare kernel observed using SST (Fig.3a, right panel). Therefore, our model should not and, as the reviewer correctly states, does not, reproduce the SXR emission of the whole flaring region in the remaining area equivalent to 1075 pixels of SST measured by one RHESSI pixel, or the full disk emission measured in the GOES light curve. It rather models a single beam injection event, which contributes, to the full disk SXR curve at the time of the beam injection as reproduced in the paper in Fig.1 (left).

In the revised paper this resolution issue is mentioned in section 2.1 and explained in section 5.1.

C3. I am still puzzled by a model that is unable to reproduce flare temperatures (>10 MK), while others do. The authors should at least mention that the model does not address this issue, which deserves further investigation.

A3. Firstly, we would like to issue the point that there should be no *expectation / criteria* that a flare must have a temperature of 10MK. To support this point, we would like to draw the reviewers attention to the paper *Ryan, D. et al., 2012, "The Thermal Properties of Solar Flares over Three Solar Cycles Using GOES X-Ray Observations", ApJ, 202, 11*. In that paper, see in particular Fig. 8 (top right panel), there is a broad distribution of peak flaring temperatures that does extend well below 10MK. Secondly, we fully agree that this particular beam model (F10: Initial flux = 10^{10} erg cm⁻² s⁻¹) does not produce flare temperatures >10 MK. We also agree that our models do not include contributions from the persistent, overlying and neighbouring corona (see A2 above). Thanks to comments by the reviewers, **changes to our text reflect this fact, and mention that the HD models with stronger initial fluxes do indeed produce 10MK coronal temperatures in the flaring corona, converted from the quiet Sun chromosphere.**

In order to assuage the concerns of the reviewer regarding our models, we include examples of the models of beams with higher initial fluxes (See the F11 F12 model profiles below for spectral index 3) which do reproduce the temperatures $>10\text{MK}$ discussed by the reviewer (Somov, Syrovatskii and Spektor 1981, Zharkova and Zharkov 2007, Zharkova and Zharkov 2015). The coronal temperature can approach 50 MK or higher for spectral indices higher than 5.

For the particular beam injection site studied here, the weaker F10 beam is the appropriate model, when compared with the observations for the weak flare studied. Evidence for this includes (a) the HD shock velocities and Doppler shift of the H-alpha line during the impulsive phase, which is a much closer match to that of the F10 model, but not for the F11 and F12 models (b) the material in the jet that emerges from the H-alpha kernel location (Fig.2b, blue arrow) is visible in AIA 171 and only very faintly in AIA 94, and with a timescale that agrees with the flow speeds predicted by the F10 model (see section 5.3 in the revise paper and answer A4 for reviewer 3 for more details on the AIA line synthesis comparison with the observations in AIA 171 and AIA 94).

Therefore, the measured coronal temperature ($<2\text{MK}$) and upward speed in the corona (93 km/s), as well as, the measured chromospheric downward velocity (up to 50 km/s), all conclusively support the low beam flux of the F10 model used in our simulations. This beam injection happens to be in the 2% of the area registered as the ball-park figure provided by RHESSI analysis of the flaring region (covering 98% of the surrounding area) without this beam injection site.

The authors wish to thank the reviewer 1 for valuable comments from which the paper strongly benefited.

Reviewer #3 (Remarks to the Author):

C1. The authors have improved the paper somewhat, in particular the simulated H-alpha profile looks better. I still think there are outstanding problems, which need to be remedied.

A1. We thank the reviewer for their valuable feedback and will endeavour to address the points raised.

C2. The abstract, Sections 3.2 and 4 once again state that simulations, plural, were performed. This is not true of what's presented in the paper.

A2. We partially agree with this comment. Although a number of simulations were performed to reconstruct the observed H-alpha profile, it is absolutely correct that only the most appropriate simulation of F10 model for this particular event was presented in the previous version of the paper.

In the revised paper we present these additional simulations, which include: (a) 2 additional HD models for other fluxes (7×10^9 , and 3×10^{10} erg cm⁻² s⁻¹), besides the model F10 already presented. This covers the range of fluxes obtained from the RHESSI data, if the H-alpha and AIA observations are also considered (see section 5.1 explaining the resolution issue affecting the area of flaring event covered by RHESSI and H-alpha kernels) and (b) more mesh points throughout the H-alpha formation region to obtain even closer resolution of the H-alpha blue and red horns. The application of these new HD models for NLTE simulation of H-alpha emission shows that only the model with the flux 10^{10} erg/cm²/s can closely account for the H-alpha profile observed, while the models with lower (or higher) energy flux produce much smaller (or larger) red shifts of H-alpha emission than those recorded by SST (see Fig. 4 and 5). Hence, we demonstrate that the RHESSI data can only be used for estimating the order of magnitude of the beam flux. See also the answer A3 below regarding the difference in resolution of RHESSI pixel (2"), AIA pixel (0.6") and SST pixel (0.06") and their implication on the measured results.

In the revised version we include plots for two HD models simulated (see Fig.4) and the results of three NLTE simulations for 3 HD models (see fig 5a, left). Hence, now we believe it is appropriate to keep the phrases mentioned in a plural form in the abstract and section 3.2.

C3. I'm in agreement with Reviewer #1 that the model would not reproduce the SXR emission,

A3. We are also in agreement with the both reviewers. However, we wish to re-iterate the point from the previous replies to the reviewers that the model presented by us does not, and should not reproduce the full SXR emission presented in Fig.1 (left), which is the GOES light curve taken from full disk images of the Sun. This includes the contributions from the whole flaring regions (see Figs.1b, 2a and 3a left panel) for events 1, 2, and 3 as described in section 2.1 and given the arguments mentioned previously. Our model includes a flaring corona that is converted from a quiet Sun chromosphere with a *transient* coming from the flaring chromosphere, rather than having an *inherent* corona initially. Hence, we do not attempt to explain the formation of a

persistent temperature of the overlying or neighbouring corona, which becomes an issue with RHESSI resolution issue discussed below.

Following the RHESSI website <https://hesperia.gsfc.nasa.gov/rhessi2/mission/mission-facts/index.html>. and the original paper by Hurtford et al. 2002 <http://adsabs.harvard.edu/abs/2002SoPh..210...61H> stating this resolution to be not worse than 2.3" (rounded to 2" by us) we can evaluate the resolution given by one pixel by RHESSI (2"x2").

To visualise a response for the emission detected by RHESSI, let us compare this resolution per pixel provided by RHESSI (~2") and by AIA (0.6") and SST (0.06"). In one pixel (2"x2") of high energy emission by RHESSI we have about 3x3 (9 in total) pixels of AIA emission and about 33x33 (~1100 in total) pixels of SST emission. For the flaring kernel considered only 5x5 (25 in total) pixels of the SST event 2 were used to measure H-alpha line emission, and only these pixels were affected by the local heating due to electron beam.

Then one pixel of AIA emission of size (0.6"x0.6") would contain 10x10 (100 in total) pixels of SST, from which only 25 are affected by the beam. Hence, the flaring H-alpha emission comes only from the area covered by these 25 SST pixels. The area equivalent to the remaining 75 pixels of SST (out of total 100) covering a single pixel of AIA would provide the emission from the neighbouring corona of this active region, which account for its thermal emission.

The situation is even more straight for the RHESSI emission, where a single RHESSI pixel contains 1100 pixels of SST, from which only 25 have the flaring event with the injected beam, while the area covered by the other 1075 pixels (or 98% of RHESSI data) is the neighbouring corona of this active region, which does not have any effect from this particular electron beam. This explains the equal correspondence of the RHESSI data to thermal emission in the whole area or to non-thermal emission imposed by a small area observed by RHESSI. This explains the reason why we should not account for this thermal emission in this observation because this is 98% the emission from the neighbouring corona meaning a need to explain the coronal heating problem for the quiet sun in the active region and not in a flaring atmosphere, which constitutes only 2% of the emission area registered by RHESSI.

In summary, since our model is for a hydrodynamic and radiative response of a flaring atmosphere by brief injection of electron beam (labelled as event 2 in section 2.1), then it is more appropriate using the observed high cadence and high resolution H-alpha images and spectra SST spectra of the selected flare kernel (Fig.3a, right panel) for comparison with simulated H-alpha profiles. This is what we did now and explicitly indicated in section 5.1.

However, our model should not and, as the reviewers correctly state, does not reproduce the SXR emission of the whole flaring region in the remaining area measured by many RHESSI pixels recorded, because even a single pixel of RHESSI is equivalent of 1075 pixels of SST, or the full disk emission measured in the GOES light curve. Our simulation rather models a single beam injection event in a small area, which will contribute, at the time of the beam injection, to the full disk SXR curve reproduced in the paper in Fig.1 (left) but cannot be responsible for the whole GOES curve.

In the revised paper we explained this issue in sections 2.1, 5.1 and 5.2.

C4. ...and contrary to the authors' statement in their response, would not even reproduce the AIA emission, particularly the brightening in the 94 A channel, indicative of high density plasma at a temperature close to 10 MK. The authors could, of course, synthesise AIA emission from their model to validate their claims, as done by many other authors.

A4. We followed the reviewer's advice to investigate the AIA response more fully. The response functions of the AIA channels are shown below (Fig. 1) plotted against \log_{10} of T (temperature) for the spectral lines of interest in this paper (94A, 171A and 304A).

Fig. 1. AIA response functions versus temperatures.

Indeed, the AIA 94A channel has its largest sensitivity peak close to 10MK but it is not limited in sensitivity to that specific temperature. It is clearly shown above that AIA 94A has a secondary peak with the maximum at the temperature of 1MK (see green line in Fig.1 above), i.e. it is sensitive to the flaring corona temperature peak that we model (1-2 MK). Therefore, we fully expect to observe, and indeed we do observe, emission in this 94A filter (as shown in Fig. 2 in the paper), slightly enhanced 94A emission from the source of the beam injection, in addition to the 171A emission with higher intensity.

We accept that it is important to account for the AIA synthesized intensity to compare with the observations. However, we wish to emphasize again that our model includes a flaring corona that is converted from a quiet Sun chromosphere with a *transient* coming from the flaring chromosphere, rather than having an *inherent* corona initially. Hence, we do not attempt to explain the formation of a *persistent* temperature of the overlying or neighbouring corona (note that the AIA pixel has a resolution of 0.6'' versus the SST pixel resolution of 0.06'' meaning that minimum area covering 75 of total 100 H-alpha pixels would be, in fact, the neighbouring corona) (see the answer A3 above). We do not intend to explain the contributions of these emissions to the emission of a flaring corona (in one particular loop observed by SST in 5x5 pixels), because our model certainly does not solve the coronal heating problem. However, what we are able to account for is a small AIA intensity in the 94

channel *above* an observed background level, due to the secondary sensitivity peak at temperatures around 1MK (discussed above).

Fig. 2. AIA spectral line synthesis (lines) and observed intensities (symbols) overlaid

Fig.2 (above) shows the simulated (green line) and observed (green crosses) light curves for AIA 94A channel, and likewise in yellow the light curves for the AIA 171A channel in the area of 3x3 AIA pixels. There is a small increase above the background levels in the observed AIA 94A during the first 20-30 seconds caused by heating of flaring atmosphere by an electron beam. Emission due to the

secondary sensitivity peak in AIA 94A, is shown to be capable of producing such an enhancement. This is in agreement with the temperature of 1-2 MK, which is predicted in our HD models heated by an electron beam with lower energy flux of 10^{10} erg/cm²/s.

Most importantly, the F10 HD model leads to a much greater excess of intensity in the AIA 171A channel for some time during and after the beam injection (Fig.7a, yellow line), as observed (Fig.7a, yellow crosses). This AIA 171A enhancement should remain visible during the outflow (jet) process, and is indeed observed (see Fig.2b, blue arrow). The simulated jet travels at ~ 90 km s⁻¹ and the observed jet traverses 3 pixels in the image space corresponding to ~ 1500 km. Hence, the simulated jet would take ~ 15 -20 s to appear 3 pixels from the H-alpha kernel location in the observations (or later if travelling at some angle out of the plane of observation). So the simulation predicts that we should only expect to see the displacement of the jet after this time, in agreement with observations (Fig.2b, blue arrow). The simulated light curves in the two channels were normalised to unity at their peak values (Fig.7b) and subtracted, in order to analyse the excess of the AIA171A enhancement relative to that in AIA 94A. This excess is plotted as a fraction of the enhancement in the AIA171A channel in Fig.7c. The enhancement in AIA 171A peaks during the beam injection phase (0-10 s) and decreases afterwards (Fig.7a). After 50 s the response in this channel has returned to background level (Fig.7a). Because the jet is observed away from the H-alpha kernel location after 20 s and the AIA cadence is 12 s, the jet should only be visible in AIA 171A for 1-2 time frames according to the F10 model, which is indeed the case (Fig.2b).

Fig.7c and shows that at 30 s our model predicts a much greater enhancement over the AIA171A background than over the AIA94A background. Therefore, the F10 model predicts the presence of a jet, outflowing from the chromospheric source of beam heating, that is visible in AIA 171A and not AIA 94A at around 30 s. The fact that a jet is not observed in AIA 94A but only in AIA 171A (Fig.2b), adds further evidence to support the value of the flux used in the F10 model, because it places an upper boundary on the temperature, T, of the outflow, i.e. T is much less than 10 MK, and limited to 1-2 MK. At the same time, the jet is not observed in AIA 304A (Fig.2b) resulting in similar implications for the lower temperature. One can conclude that the jet must be also much hotter than 100,000 K (the sensitivity peak for the AIA 304A channel). This is why this jet is clearly observed in the AIA 171A channel.

This interpretation is completely supporting our F10 model predictions of the temperature and density properties of the flaring outflow. **A discussion of these points is now included in a new methods section 5.3 of the revised paper.**

C5. The RHESSI analysis has some problems now that there's enough information to try to reproduce it.

A5. We will endeavour to answer these questions below. However before we begin this, it is worth to emphasize the resolution issues for the RHESSI, AIA and SST pixels discussed above in A3.

This difference in the spatial resolutions of different instruments used in this study leads to the fact that the RHESSI data covering 98% for the neighbouring emission and only 2% for the flaring kernel emission, can be only used to produce a 'ball-park' flux number, or a range, of initial energy fluxes of electron beam causing this HXR and H-alpha emission. The theoretical H-alpha line profiles with red

shifts caused by hydrodynamic shocks with the downward shock velocities produced by injection of electron beams with given energy fluxes can be compared then with the observed H-alpha profiles with Doppler shifts measured by SST and the outflow velocities of the jet measured by AIA 171.

Since the RHESSI data cannot be resolved in a flaring loop with H-alpha emission observed in 5x5 SST pixels, we have to conclude that the RHESSI data alone should not be used for determining the beam flux of a small, individual injection site. In this case, only the 'ball-park' flux range of fluxes can be provided through RHESSI analysis that led us to a relevant set of beam fluxes for this event. The most appropriate beam flux model was tuned more precisely by comparison of the H-alpha line profile with the relevant observations. The updated version of the paper now reflects this fact (see sections 2.1, 3.2 and 5.1).

C5.1. The background period 09:38:40 - 09:40:56 UT is not a solar background, but something closer to a flat field. RHESSI was entering its nighttime period then, as should be clear from the light curves. The sudden drop at 09:36 in the RHESSI light curves does not occur in the GOES data, showing that it does not represent solar data.

A5.1. - The drop at 09:36 is due to the onset of spacecraft night. The background is comprised of instrumental, non-solar and solar contributions. While subtracting the background outside of the flare time in the same attenuator state is the ideal case as suggested by the referee (and agreed by us), it is not unreasonable to use a period of night for this when such an interval is not available (see for example Krucker et al., 2002, Sol. Phys. 210, 445 and Iain Hannah's tutorial:

http://www.astro.gla.ac.uk/solaire/RHESSI_MATERIAL/rhessi_spectra_hannah.pdf). Background taken during night includes the instrumental and non-solar components. In this case we subtract a period of night following the flare, when the solar background should be low, as can be seen from the 0.5 -4.0 A GOES curve. As noted in A5.2 below, the background subtraction is most critical for high energies, which this flare does not exhibit.

Given the spatial resolution issue between RHESSI and H-alpha observations discussed in A3, we cannot fit this emission exactly to the flaring event, which mainly has the contributions of neighbouring corona of the active region. Thus, we do not think it can be productive to spend much time trying to improve the RHESSI analysis for this event which 98% has the background emission and only 2% of the selected flaring kernel.

C5.2. The normal procedure for RHESSI background subtraction is to average the time periods from +/- 15 orbits (approximately 1 day), assuming no flares occurred at those times.

A5.2. Inaccurate background is an important issue at high energies, e.g. 15 orbit averaging is appropriate for higher energies, and, especially, gamma-ray spectroscopy (e.g. Share et al., 2004, ApJL, 615,169). But in this flare the energy range is only 7-21 keV. Inspection of the background spectrum shows it looks entirely reasonable for these energies. Furthermore, following Kim Tolbert's guide to object based spectroscopy it is acceptable to select background levels at time intervals closer in proximity to the flaring event (not necessarily 1 day difference).

C5.3. The RHESSI spectrum as presented appears to have pulse pile-up, evident from the spectrum at ~ 13-14 keV. The count rates exceed 1000 counts/s in the SXR bands, used as a test for pulse pile-up.

The authors are interpreting the excess emission as non-thermal emission from an electron beam, in which case they need to explain how they've ruled out pulse pile-up. When I use a pulse pile-up correction, the fit finds no evidence for non-thermal emissions (whether done with the 1pow or thick2 functions in OSPEX).

A5.3. This is a repeated comment from the review in the round 1 (see Appendix 1 below, round 1, comment 4.3 and answer to it).

Pulse pileup has always been corrected in our RHESSI fittings via OSPEX.

We agree and have explicitly acknowledged in sections 2 and 5.2 (see Appendix 1, round 1, comment 4.3 and answer A4.3) that for this flare, a thermal fit of HXR emission is as good as a non-thermal power law. So in reality, we cannot distinguish between these two functional fits and draw precise conclusions from the fits because of the RHESSI and H-alpha resolution issue discussed in A3.

Therefore, the fitted RHESSI data, specifically the estimate of the electron flux parameter, cannot be *solely used* as justification for the chosen initial electron flux of the beam in our model, leading to the H-alpha emission. Additional justification for the choice of initial electron flux is achieved in our paper through constraining the simulated profile of H-alpha emission by that observed with SST and to fit the required red shift, and by the velocities of the hydrodynamic shocks produced in the chromosphere. This means that the RHESSI-defined electron flux does not define our conclusions but merely provide a suitable *context* of parameters (i.e. in general agreement as a ball-park number) for our conclusions derived through H-alpha profile fit.

Fig. 3. Updated H-alpha line profiles calculated for 3 HD models (7F9, F10, 3F10).

In the revised paper, we demonstrate that the simulated H-alpha profile fits better the observed H-alpha profile for the model with the electron flux of 10^{10} erg/cm²/s (purple) than for 3×10^{10} erg/cm²/s (blue) or 7×10^9 erg/cm²/s (yellow). The new version of Fig.5 containing the spectral profiles for these electron fluxes (that now appears in the new version of the manuscript), is again presented next for your information.

These simulated H-alpha profiles for beams with different energy fluxes (yellow, purple, blue) after 5 seconds of beam injection, cover the upper and lower limits of the initial energy flux, as derived with either the pow or thick2 fit from the RHESSI spectrum. Given uncertainty of the RHESSI data and larger area covered by HXR images compared to H-alpha or AIA, this is the best outcome that can be extracted from the RHESSI data. We hope, this explanation clarifies that the RHESSI spectrum gives us a good “ball-park” figure for what we expect the electron fluxes to be (as observed in this flare). However, a more precise determination of the electron flux is achieved from the model comparison with observed H-alpha profiles. This has been the essence of our results all along as we do not try to model the SXR emission of RHESSI.

C5.4. I would recommend that the authors continue to use the thick2 function for attempts at fitting, rather than 1pow, which does not fit to an actual bremsstrahlung function from an electron distribution. 1pow cannot give a low energy cutoff E_c , for example, so it's then incorrect to state that the simulation values are based off the observed values.

A5.4. This is the repeated comment from the round 2 (see Appendix 1, round 2 comment C2a and answer A2a). We already used the thick2 function and referee asked us to change it then.

In Appendix 1 we show the images produced for paper and confirm that all RHESSI fitting give consistently the same spectral index ~ 3.8 for HXR emission. We consider this to be a verified spectral index for HXR emission used by us in further simulations. This index is converted to the electron beam spectral index of 4 (+0.05), following the Fokker-Planck simulation of electron beam kinetics and resulting HXR emission by Zharkova and Gordovskyy (2005, A&A, Fig.11). This is now clearly stated in the paper (sections 5.1 and 5.2).

Also in the past two decades Zharkova et al has shown in many papers (Zharkova et al, 1995, A&A; Zharkova and Gordovskyy, 2005, A&A; Siversky and Zharkova, 2009, A&A; Zharkova et al., 2010, A&A; 2010, ApJ, Zharkova et al., 2011, A&A) that a single electron beam can produce the HXR spectrum with double power law because of a return current effect, leading to soft-hard-soft pattern of HXR spectra shown in the RHESSI nuggets:

(http://sprg.ssl.berkeley.edu/~tohban/nuggets/?page=article&article_id=25 and http://sprg.ssl.berkeley.edu/~tohban/wiki/index.php/Return-current_Model_Spectra_and_Enhanced_Plasma_Resistivity).

This is well accepted now (see for example, Zharkova et al, 2011, SSRv; Holman et al, 2011, SSRv and RHESSI book). For the purpose of our simulation we always use the spectral index of higher energy part, which is not affected by return current and reflects correctly spectral index of electron beam producing this HXR emission.

Given the narrow range of energies 7-21 keV available for this event, it is rather difficult to vary significantly the lower cutoff energy either above or below 7 keV used, because below 7 keV HXR is mainly thermal emission, while any move above 10 keV will reduce the total flux to negligible values.

Let us demonstrate this point in simple calculations of the energy flux F_0 for a power law electrons in pure collisional approach with a distribution function $\sim KE^{-g}$ (where in our case $g = 4$ following Fig.11 in Zharkova and Godrovskyy, 2005, A&A cited in the paper). Then the energy flux delivered by these electrons can be found by integration over energy of the product of energy and distribution function:

$$F_0 = K \int_{E_{low}}^{E_{high}} E E^{-4} dE = -\frac{K E^{-2}}{2} \Big|_{E_{low}}^{E_{high}} = \frac{K[E_{low}^{-2} - E_{high}^{-2}]}{2}.$$

For $E_{low}=7$ (squared gives 49) it gives the flux $F_0=0.02\frac{K}{2}$ and for $E_{low}=10\text{keV}$ (squared gives 100)= $F_0=0.01\frac{K}{2}$. The difference between these fluxes is a factor of 2.

While the uncertainty of the areas of a flaring atmosphere observed by RHESSI and SST caused by a difference in the spatial resolution of RHESSI and SST is even greater. In the revised paper for simulations we used the fluxes between $F1=7 \times 10^9$ and $F2=3 \times 10^{10}$ erg/cm²/s, covering the range of factor 4 between $F1$ and $4F1$ that covers the variations of E_{low} and areas.

As explained in A3, for this observation of H-alpha emission with high-resolution telescope, we can only use RHESSI for the estimation of the flux limits.

Hence, in this revision we conclude now that the main fit of the beam parameters (initial energy flux) was tuned using H-alpha observations rather than HXR observations.

C5.5. Thick2 can be fit with a single power-law by setting the break energy greater than the high-energy cut-off parameter, and not allowing them to vary. See the OSPEX documentation here: https://hesperia.gsfc.nasa.gov/ssw/packages/spex/idl/object_spex/fit_model_components.txt.

A5.5. Yes, we are aware of this, thank you. This comment is answered in A5.5 above (see also Appendix 1, round 2, comment C2e and answer A2e). Again, given the spatial resolution issues of RHESSI and SST pixels discussed in A3 above, the parameters derived from the RHESSI data have only the 'ball-park' approach value to estimate the range of parameters. Therefore, we did not perform any further fitting for the RHESSI data.

In the revised paper in the caption in Fig. 1 we have now explicitly acknowledged that HXR is mostly of thermal nature and with only a small non-thermal component. We refer the readers to sections 2.1 and 5.1 for further discussion on this. For demonstration of this point we kept a single power law spectrum in Fig. 1 as in the previous version and we hope very much that the reviewer 3 would agree with this presentation.

C5.6. The authors have not explained how they have now obtained their new energy flux or low

energy cutoff. The values surely have changed if the fit was re-done. If the differences are significant, the authors ought to consider re-doing the simulation.

A5.6. This comment is answered in A5.5 above. Given the spatial resolution issue between RHESSI and SST pixels discussed in A3 above, the parameters derived from the RHESSI data have only the ball-park approach value to estimate the range of beam parameters. The tuning of fitting parameters for the electron beam was done using the H-alpha observations and fitting them with simulated H-alpha profiles for initial fluxes within the limits derived from HXR observations (Fig.5 in the paper or Fig.3 in this report).

C5.7. The authors also explain a number of differences between the simulated and observed H-alpha profiles by saying that the “differences can be corrected with a more energetic beam.” The authors have a model at their disposal to actually test this, and should do so!

A5.7. We appreciate this comment by the referee and took it into account.

In the revised paper, we have corrected the difference between observed and simulated H-alpha profiles by: a) adding more depth modelling points in HD modelling for the chromospheric region where H-alpha emission is formed compared to the number used in the previous version of the paper; b) simulating hydrodynamic models for 3 initial energy fluxes covering the limits derived from HXR emission, e.g. for the initial energy flux 7×10^9 , 10^{10} , 3×10^{10} erg cm⁻² s⁻¹.

This allowed us to reproduce more closely the observed H-alpha profile shown in the revised Fig. 5 (or in Fig.3 above), which shows that only the 10^{10} model fits very closely the H-alpha line profile observed.

Also in the updated paper, our H-alpha profile simulated for a beam with initial flux 10^{10} erg cm⁻² s⁻¹ now has the red-to-blue wing peak intensity ratio of 2.9 close to the observed 2.1 and the blue horn is located at -0.5A from the line centre, also in agreement with the observations. The shift of the central reversal slightly to the blue wing could be explained by many other variations and uncertainties, such as the precise macrovelocities of the shock just above the horn formation region or contributions to the observed profile from pixels which are not fully covering the beam injection site etc. We hope that these improvements add further weight to the comparison. However, we retained the criticisms of this profile in the paper, as suggested previously by the reviewer.

We wish to thank the reviewers for enabling this further clarity in the presentation of our results from all of their suggestions in this review process. Consequently, we wish to acknowledge the reviewers in the acknowledgement section of the paper, given their input, specifically with regard to the content of the methods section, with their approval.

Appendix 1

Extracts of the comments on RHESSI analysis by the referee 3

Round 1

C4. The RHESSI data analysis is unclear, and so somewhat difficult to interpret. A few points should be clarified:

C4.1. Which detectors were used to make the spectrum and images?

A4.1. The images were made using the CLEAN algorithm and detectors 3-8. The spectra were made from detectors 4, 5 and 9.

C4.2. Which specific functions in OSPEX were used to fit the data?

A4.2. As indicated in Figure 1, the spectrum was fitted with thermal + thick target with single power law components. We fitted over the energy range 7 - 40 keV.

C4.3. The spectrum presented in Figure 1a looks like it could be well fit with only a thermal component. Did the authors check to see whether that was the case? The residuals of the fit would clarify, and should be included in the figure.

A4.3 This is a fair comment.

In the revised paper we described that we did also fit the spectrum with a) a thermal plus thick target components and b) a single thermal component, and found that the chi2 and residuals were comparable for both cases. The residuals are now added to the figure 1.

But because the H-alpha emission occurred very quickly and simultaneously with HXR emission, we found that an electron beam is a more suitable agent for delivering non-thermal excitation and ionisation in this flare than thermal electrons with higher temperature, because only sub-relativistic electrons can reach the chromosphere in under 1 s timescale and excite/ionize hydrogen within seconds to account for H-alpha emission appearing simultaneously with HXR emission. This is now explained in the revised paper, section 5.2.

C4.4. The authors state that the energy flux was 10^{10} erg/s/cm², but give no indication as to how this was determined, nor what cross-sectional area was measured (or assumed) in order to determine the flux. The results of hydrodynamic simulations depend strongly on the beam flux, so it is important to explain how the authors arrived at this number.

A4.4. We agree, and thank the reviewer for this comment.

In the revised paper we included the calculation of the flaring event 2 area in section 2.1, and the initial energy flux of beam electrons based on the total electron flux and the area where it was deposited. We used the area of the H-alpha kernel where the line profile was obtained.

C4.5. How long were the integration times used to make the RHESSI images? Which level contours are displayed?

A4.5. The integration times were 20s as the photon numbers are very low. Contour levels are 30%, 50% and 70% of the maximum intensity. This is now indicated in the text of section 2.1.

Round 2

C2a. The first regards the RHESSI analysis, and the parameters derived from it and then used in the simulation. It still needs more detail to properly interpret, and importantly to reproduce. I attained rather different fits from the spectrum when I attempted it myself. Figure 1a, right, shows the X-ray spectrum during event 2 (UT 09:15:54), integrated over 20 seconds according to the text, using detectors 4, 5 and 9, and fit in the range 7-40 keV according to the response.

A2a. We appreciate the referee comment highlighting this problem. We have re-fitted the spectrum over the period 09:15:54 – 09:16:14 UT, using the detectors 4,5 and 9 and a thermal + single power law fit and 0.33 keV binning over the range 7-21 keV. As you can see from the revised figure, the χ^2 is much reduced and the residuals show less structure. The power-law exponent is still 3.82. Although the upper energy cutoff was reduced to 21 keV that is in line with the referee comment about the noise in HXR photons in this event above 20 keV. This reinforces our initial beam parameters discussed in the paper.

In the revised paper this updated spectrum was used in the Fig. 1 instead of the previous one.

C2b. - What time period was used for the background?

A2b. The background period was chosen to be 09:38:40 – 09:40:56 UT. This is now added to the paper text.

C2e. - Why did the fit extend to 40 keV? There is only noise above ~ 20 keV.

A2c. The reviewer is quite correct that there are few counts above 20 keV. We only extended the over-plotted model spectrum to 40 keV for display purposes.

In the revised paper the new figure has been generated to only show the fit to ~20 keV as the referee recommends (see A2a).

C2d. - The plot indicates that only detector 5 was used in the spectrum, and it's not clear if this is a simple typo. It also indicates the interval as being integrated over 2 minutes.

A2d. This is a valid comment, thanks. It was an illustrative plot which is now removed. In the revised paper we have replaced this illustrative plot with the more suitable one using the detectors mentioned (see answer A2a).

C2e. - The derived parameters from the thick2 component indicate that this component was fit assuming a double power law. This is in contrast to the simulation, which assumes a single power law as explained in Section 5.2. (The six parameters listed at the bottom of the plot, respectively: electron number flux in 10^{35} electrons/s, spectral index of the electron distribution below the break energy, break energy in keV, spectral index of the electron distribution above the break energy, low energy cut-off and high energy cut-off in keV).

A2e. We thank the reviewer for pointing out this inconsistency. We have re-fitted the spectra as described in A2a that eliminate this problem, the spectrum has now single power law.

C2f. - Using the numbers given from the thick2 fit, the non-thermal power would be closer to $\sim 6 \times 10^{26}$ erg/s (Holman et al 2011), which would raise the energy flux in the simulation by a factor of 6 and drastically affect the results.

A2f. We accept this comment with thanks. This problem is now rectified, there is no double power law spectrum generated for this event 2, only a single power law one as per answers A2a –A2e.

Although, the increase of the beam energy by factor 6 suggested by the reviewer 3 from our illustrative spectrum with detector 5 would improve the fit of H-alpha line profile with red shift allowing to obtain the exact velocities of the red shift of about 50 km/s as reported by the observations. However, since the HXR photon statistics above 20 keV is poor, as the reviewer 3 also noted, so we cannot make this claim with confidence that the energy flux of electron beam was higher, but can only speculate about it.

This is what we discussed in the paper text by looking at the possible ways how to improve the fit between the observed and simulated profiles.

Round 3.

Current version of the plot – used for demonstration only.

Fitting with the thick target showing pure thermal HXR emission.

R3 comments:

The RHESSI analysis has some problems now that there's enough information to try to reproduce it.

1. The background period 09:38:40 - 09:40:56 UT is not a solar background, but something closer to a flat field. RHESSI was entering its nighttime period then, as should be clear from the light curves. The sudden drop at 09:36 in the RHESSI light curves does not occur in the GOES data, showing that it does not represent solar data.

The normal procedure for RHESSI background subtraction is to average the time periods from +/- 15 orbits (approximately 1 day), assuming no flares occurred at those times.

2. The RHESSI spectrum as presented appears to have pulse pile-up, evident from the spectrum at ~ 13-14 keV. The count rates exceed 1000 counts/s in the SXR bands, used as a test for pulse pile-up. The authors are interpreting the excess emission as non-thermal emission from an electron beam, in which case they need to explain how they've ruled out pulse pile-up.

When I use a pulse pile-up correction, the fit finds no evidence for non-thermal emissions (whether done with the 1pow or thick2 functions in OSPEX).

3. I would recommend that the authors continue to use the thick2 function for attempts at fitting, rather than 1pow, which does not fit to an actual bremsstrahlung function from an electron

distribution. 1pow cannot give a low energy cutoff E_c , for example, so it's then incorrect to state that the simulation values are based off the observed values.

Thick2 can be fit with a single power-law by setting the break energy greater than the high-energy cut-off parameter, and not allowing them to vary. See the OSPEX documentation here:

https://hesperia.gsfc.nasa.gov/ssw/packages/spex/idl/object_spex/fit_model_components.txt

4. The authors have not explained how they have now obtained their new energy flux or low energy cutoff. The values surely have changed if the fit was re-done. If the differences are significant, the authors ought to consider re-doing the simulation.

REVIEWERS' COMMENTS:

Reviewer #3 (Remarks to the Author):

The authors have satisfied all of my concerns, and I would now recommend publication. The authors have shown a good faith and earnest effort to satisfy my comments. There are a few minor points I am still skeptical of, but I do not believe they would alter any conclusions.

I thank the authors for their patience, and look forward to seeing further developments!

Reviewer #3 (Remarks to the Author):

The authors have satisfied all of my concerns, and I would now recommend publication. The authors have shown a good faith and earnest effort to satisfy my comments. There are a few minor points I am still skeptical of, but I do not believe they would alter any conclusions.

I thank the authors for their patience, and look forward to seeing further developments!

Response to reviewer

We thank the reviewer for their time, feedback and assistance in improving the paper throughout the peer review process.